# Olmix: A Framework for Data Mixing Throughout LM Development

**Mayee F. Chen** [1 2 *]  **Tyler Murray** [1]  **David Heineman** [1]  **Matt Jordan** [1]
**Hannaneh Hajishirzi** [1 3]  **Christopher Ré** [2]  **Luca Soldaini** [1]  **Kyle Lo** [1 3]

## Abstract

Data mixing—determining the ratios of data from different domains—is a first-order concern for training language models (LMs), but existing mixing methods have poorly understood design choices and assume that the set of domains remains fixed throughout development. We present OLMIX, a framework that addresses two challenges encountered during LM development. First, the configuration space for developing a mixing method is not well understood—design choices across existing methods lack justification or consensus and overlook practical issues like data constraints. We conduct a comprehensive empirical study of this space, identifying which design choices lead to a strong mixing method. Second, the domain set evolves throughout LM development as datasets are revised and expanded—a problem setting largely unaddressed by existing works. We study how to efficiently recompute the mixture after the domain set is updated, given an existing mix from before the update. We introduce mixture reuse, a mechanism that reuses existing relative ratios and recomputes ratios only for domains affected by an update. Over a sequence of five domain-set updates mirroring real-world LM development, mixture reuse matches the performance of fully recomputing the mix after each update with 74% less compute and improves over training without mixing by 11.6% on downstream tasks.

## 1. Introduction

Modern language models (LMs) are trained on datasets composed of many domains, such as web text, code, and PDFs. The composition of these domains is crucial for strong downstream performance, making data mixing a first-order component of LM development (Grattafiori et al., 2024; Qwen et al., 2024; Olmo et al., 2025). However, naively searching for a good mix is computationally inefficient, requiring many training runs—possibly thousands of GPU hours—to assess performance. This has resulted in a growing literature on data mixing methods that aim to find strong mixtures systematically with less compute (Xie et al., 2023; Fan et al., 2024; Chen et al., 2025a), *inter alia*. Many mixing methods that achieve promising results follow a common offline schema (Liu et al., 2025b;a; Ye et al., 2025): 1. train a set of smaller proxy models on different mixtures (a "swarm"), 2. fit a regression model on this swarm to predict performance from mixture weights, and 3. select a final mix that optimizes predicted performance.

However, translating these methods to real-world LM development reveals two challenges that existing work does not address: configuring a mixing method at the *start of* development and maintaining strong mixtures *throughout* development as domains evolve (Figure 1). We present OLMIX, a framework for addressing these motivating problems:

**P1: The mixing configuration problem (§3).** At the start of LM development, practitioners must configure a mixing method on an initial domain set. However, the configuration space of existing methods is poorly understood: design choices are often unjustified, contradict across methods, or fail to address practical issues like data constraints. As a result, practitioners have conflicting or little guidance on which choices lead to a strong mixing method.

The first component of OLMIX addresses P1 through a comprehensive empirical study of the offline schema's configuration space. Specifically, we resolve conflicting guidance in prior work by answering the following research questions:

RQ1 *What is the smallest proxy model size that generalizes to larger target models?*

RQ2 *What is the minimum number of proxy runs (swarm size) needed to learn an effective mix for a given domain set?*

*Internship at the Allen Institute for AI. [1]Allen Institute for AI, Seattle, WA, USA [2]Department of Computer Science, Stanford University, Stanford, CA, USA [3]Paul G. Allen School of Computer Science & Engineering, University of Washington, Seattle, WA, USA. Correspondence to: Mayee Chen <mfchen@cs.stanford.edu>, Kyle Lo <kyleclo@cs.washington.edu>.

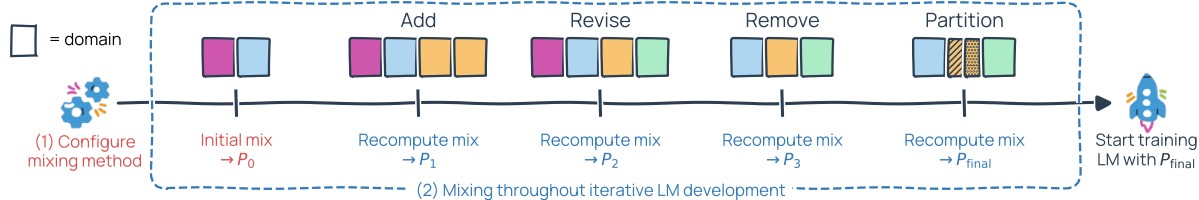

*Figure 1.* Two problems with data mixing encountered during LM development: (1) How to best configure your mixing method? (2) How to efficiently mix under evolving domain sets, such as during iterative LM development?

RQ3 *Is there an optimal family of regression models for predicting mix performance?*

We further address an unexplored but practical setting:

RQ4 *How can we perform data mixing under finite data constraints?*

These findings inform our base mixing method, OLMIXBASE (Algorithm 1), which we then use to address P2.

**P2: The evolving domain problem (§4).** Existing mixing literature assumes a fixed domain set (Albalak et al., 2024), but real-world LM development is iterative: datasets are repeatedly added, removed, revised, and partitioned (Olmo et al., 2025; Bakouch et al., 2025; Wettig et al., 2025). As a result, mixtures must repeatedly be recomputed as domain sets evolve. Since full recomputation after each update becomes increasingly expensive, we ask whether historical mixtures can be reused, motivating a new problem:

RQ5 *Is there an efficient strategy for recomputing the data mix after a domain update?*

The second component of OLMIX addresses P2 by introducing *mixture reuse*, an alternative to full recomputation that lowers cost by applying OLMIXBASE to fewer domains (Figure 7). FULLMIXTUREREUSE freezes the relative weights of domains unaffected by the update, recomputing only their total weight alongside the weights of affected domains. We analyze when this matches full recomputation, finding that the performance gap grows with the magnitude of the shift in optimal ratios caused by the update and the coupling between reused and recomputed domains. We therefore propose PARTIALMIXTUREREUSE as an extension of FULLMIXTUREREUSE: by recomputing the ratios on some unaffected domains in addition to the affected domains, it reduces coupling and improves performance despite the extra cost.

We evaluate mixture reuse in an end-to-end case study mirroring real-world LM data development, where the domain set evolves over 5 updates culminating in 64 domains. When training 1B-parameter models on 100B tokens, mixture reuse improves over the natural distribution by +11.6% and captures 95% of full recomputation's gains while requiring 67% fewer proxy runs.

## 2. Related Work

We provide an extended related work in Appendix A. A vast body of mixing methods follows the offline schema (Liu et al., 2025a; Ye et al., 2025; Ge et al., 2025b; Kang et al., 2025; Diao et al., 2025; Que et al., 2024; Wettig et al., 2025; Belenki et al., 2025; Held et al., 2025). Other mixing approaches dynamically explore mixtures during a proxy run (Fan et al., 2024; Xie et al., 2023), and online mixing methods (Chen et al., 2023; 2025a; Jiang et al., 2024; Albalak et al., 2023) dynamically adjust mixtures during final training. Given the prevalence of the offline schema, our work provides a study of its key design choices and extends it to handle evolving domain sets.

Real-world LM development involves iterative refinement of training data. Works like DCLM (Li et al., 2024a), Dolma (Soldaini et al., 2024), and FineWeb (Penedo et al., 2024) document domain updates through filtering, deduplication, and dataset additions. Projects like OLMo 1–3 (Olmo et al., 2025) and SmolLM 1–3 (Bakouch et al., 2025) showcase how training data evolves over time through these updates. Motivated by these works, OLMIX views data mixing as an ongoing process throughout LM development.

## 3. The Mixing Configuration Problem

We present the first component of OLMIX: a comprehensive empirical study of the configuration space of mixing methods. In §3.1, we formalize the data mixing problem that mixing methods aim to solve and describe the offline mixing schema. Then, we present our empirical study of key design choices and our findings (§3.2). These findings inform OLMIXBASE (§3.3), the mixing method we used throughout development.

### 3.1. Background

#### 3.1.1. THE DATA MIXING PROBLEM

Our goal is to determine ratios over training data domains that result in strong downstream performance of the LM.

**Domain set and mixes.** Let $\mathcal{D} = \{D_1, D_2, ..., D_m\}$ be a set of $m$ domains, where each domain $D_i$ has a training dataset of size $N_i$ tokens. A domain is a group of data, ranging

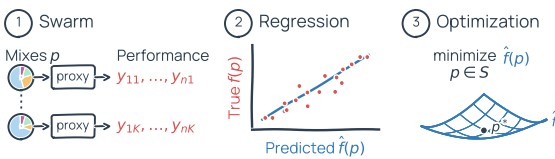

*Figure 2.* The offline mixing schema used by many existing methods (§3.1.2). OLMIX studies the design choices needed to configure each step into a strong mixing method (§3.2-§3.3).

from coarse-grained **sources** (defined by data provenance) to fine-grained **topics** (semantically coherent partitions of a source) (Wettig et al., 2025). We specify a data mixture via a probability vector $p \in \triangle^{m-1}$, such that training an LM with $R$ total tokens according to $p$ uses a dataset with $p_i \cdot R$ tokens from $D_i$ for each domain $i$.

**Model and evaluation.** We train a target LM of $S$ parameters for $R$ tokens on $p$, denoted as $\text{LM}(S, R, p)$; We then evaluate this model on a suite of $n$ downstream tasks. We measure the performance $f_i(\text{LM}(S, R, p))$ on each task $i$ in terms of bits-per-byte (BPB), the negative log likelihood of the correct answer normalized by answer length in UTF-8 bytes. Heineman et al. (2025) showed that BPB can be used for decision-making at small model scales and Huang et al. (2024) showed that BPB scores correlate with downstream performance across capabilities and model families.

**Goal.** Given a domain set $\mathcal{D}$ and target training configuration of $S$ parameters and $R$ tokens, we aim to find a mixture $p^\star$ that minimizes the average BPB across all downstream tasks—that is, minimizing $\frac{1}{n}\sum_{i=1}^{n} f_i(\text{LM}(S, R, p))$.

### 3.1.2. THE OFFLINE MIXING SCHEMA

Many existing methods follow an offline mixing schema; these methods are also described as "function fitting-based offline methods" in Liu et al. (2025b)'s survey. We describe the three steps of this schema below and summarize them in Figure 2.

**Step 1: Swarm construction.** We sample the space of possible mixes by training a "swarm" of $K$ small proxy models of size $S_{\text{small}}$ on $R_{\text{small}}$ tokens, each with different mixture weights, $p^1, p^2, ..., p^K \in \triangle^{m-1}$ sampled from some distribution $\mathcal{P}$. We evaluate each proxy model on the downstream task suite to obtain $y_{ij} := f_i(\text{LM}(S_{\text{small}}, R_{\text{small}}, p^j))$, the performance of the $j$th proxy model on the $i$th task, for all tasks over the entire swarm. Altogether, this creates a dataset $\{(p^j, \{y_{ij}\}_{i=1}^{n})\}_{j=1}^{K}$ of mixture weights paired with their performance across all tasks.

**Step 2: Regression model.** We fit regression models using the above dataset to capture the relationship between the mixture weights and downstream performance. We learn a function $\hat{f} \in \mathcal{F}$, where $\hat{f}(p) \approx \frac{1}{n}\sum_{i=1}^{n} f_i(\text{LM}(S_{\text{small}}, R_{\text{small}}, p))$.

This enables us to predict the average downstream BPB of a proxy model trained on any candidate mix $p$.

**Step 3: Mixture optimization.** We propose a mix by solving an optimization problem that uses the regression model $\hat{f}$ as a surrogate for true performance. This optimization problem takes the form $\text{minimize}_{p \in \mathcal{S}} \hat{f}(p)$, where $\mathcal{S} \subseteq \triangle^m$ denotes the feasible set, which may be the full probability simplex or a restricted subset.

### 3.2. Study: Configuring a Mixing Method

Key design choices in the offline mixing schema are not well understood in existing literature. We conduct a comprehensive study on each design choice and use our findings to develop OLMIXBASE (Algorithm 1). In total, the full study covers seven offline-schema design choices spanning swarm construction, regression modeling, and mixture optimization; the main text focuses on the choices most central to OLMIXBASE, and Appendix B reports the remaining design choices.

### 3.2.1. EXPERIMENTAL SETUP

**Data.** We use DCLM (Li et al., 2024b) partitioned into 24 topic-based domains using WebOrganizer (Wettig et al., 2025). In Appendix C, we also provide results for mixing over data sources.

**Model.** We train 1B parameter decoder-only transformer models using Olmo 2 architecture (OLMo et al., 2024) to 100B tokens (5x Chinchilla). We use a batch size of 512, sequence length of 4096, and max learning rate of 0.0018. See Appendix F for more details.

**Evaluation.** We measure the BPB (bits-per-byte) over gold responses on 52 downstream tasks spanning math, code, and commonsense QA. For QA tasks, the gold continuation is the answer text; for math and code tasks the gold continuation is a human-written response. We treat subtasks (e.g. MMLU or coding language subsets) as standalone tasks when taking a macro-average of BPB scores. See Table 8 in Appendix F for the entire list of tasks.

### 3.2.2. RQ1: PROXY MODEL SIZE

**What is the smallest proxy model size (the number of parameters, $S_{\text{small}}$) such that decision-making generalizes to larger target models?** Proxy models must provide reliable signal that transfers to the target model's scale while ideally being small so that computational costs are low. Existing methods use sizes ranging from 1M to 400M parameters, making it unclear what size is sufficient.

We train proxy models of varying sizes (1M, 15M, 30M, 60M) with 5x Chinchilla multiplier paired with 1B target models on the same mixtures and measure rank correlation of their downstream performance. High correlation indicates

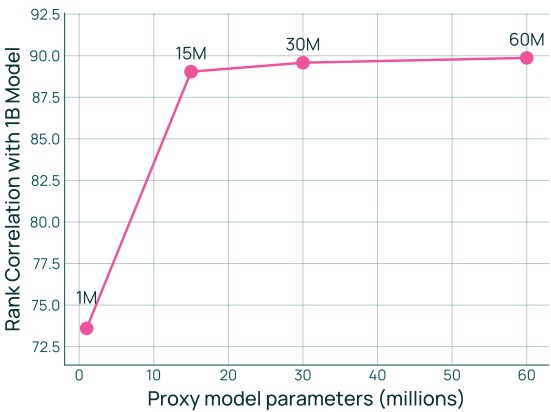

*Figure 3.* Correlation in performances between proxy models and 1B target models fixing mixture $p$. Proxy models with over 15M parameters achieve strong rank correlation (§3.2.2).

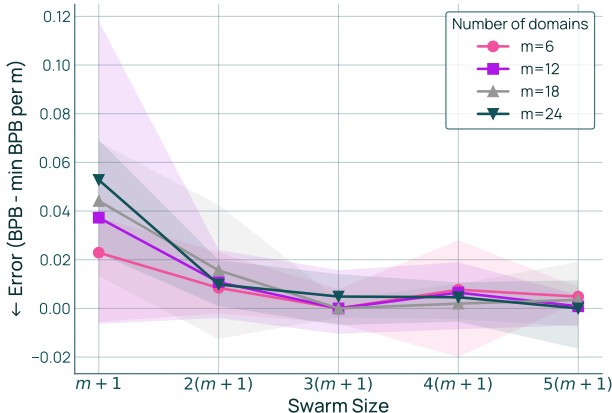

*Figure 4.* Error vs. Swarm Size. Curves collapse across different $m$, indicating $\mathcal{O}(m)$ runs are needed (§3.2.3).

that rankings at the proxy scale predict rankings at the target scale, ensuring reliability of proxy swarm decisions.

Figure 3 shows that proxy models with $> 15$M parameters achieve strong rank correlation ($\rho > 89$). Notably, while RegMix (Liu et al., 2025a) recommends using a 1M proxy configuration, our results show 1M models achieve only $\rho = 73$, suggesting that they are unreliable for guiding mixture decisions.[1] **For our setting, we use 30M proxy models trained for 3B tokens (5x Chinchilla), which achieve a Spearman correlation of 89.6 with 1B target models.**

### 3.2.3. RQ2: Swarm Size

**How many proxy runs $K$ do we need to learn a good mix on $m$ domains?** The swarm size directly controls the cost-performance tradeoff: more runs increase computational cost but improve mix quality. Existing methods use fixed swarm sizes ranging from 20 to over 500 proxies, and few relate these choices to the number of domains $m$.

For a fixed number of domains $m$, we run a sweep over swarm sizes $K = c(m+1)$ for linear multiples $c = 1,2,3,4,5$. We perform this sweep for $m = 6,12,18,24$ domains and 3 seeds. For each $m$, we define error as the difference between the BPB of a proposed mix and the best BPB achieved by any mix found in the corresponding sweep for that $m$. We used 30M proxy models for all runs here but verify that the proposed mixes from these different swarm sizes also exhibit the same trends at 1B scale (Figure 15 in Appendix C).

Figure 4 shows that sample complexity is linear in $m$: the error curves across $m$ collapse when plotted against the linear multiple $c$. This reveals that $\mathcal{O}(m)$ runs are sufficient for strong performance—a finding that contradicts prior

assumptions of quadratic scaling (Ye et al., 2025; Ge et al., 2025b) and provides practitioners with a concrete prescription for allocating compute. **We recommend using at least $K \geq 3(m+1)$ proxy runs with log-linear regression, since $c = 3,4,5$ all have close to zero error in our sweep.**

### 3.2.4. RQ3: Regression Model Family

**Is there an optimal family of regression models ($\mathcal{F}$) for predicting mix performance?** The regression model is central: proposed mixes are optimized directly using its predictions. Each existing method proposes a different model and reports strong fit, making it unclear which to adopt.

We compare five regression families: LightGBM (Liu et al., 2025a), Gaussian Process (Chen et al., 2025b), BiMix (Ge et al., 2025b), AutoScale (Kang et al., 2025), and log-linear (Ye et al., 2025). We measure the regression fit across regression models and swarm sizes $K = 25,50,75,100,125$ over 3 seeds.

We discover that swarm size is a key confounding factor: different models excel at different swarm sizes (Figure 5), potentially explaining the lack of consensus. Notably, log-linear models (Ye et al., 2025) achieve the best regression fit overall across swarm sizes. We verify this choice via downstream evaluation (training 1B models) at $K = 125$, where the log-linear model's proposed mixes again achieve the best performance (Figure 16 in Appendix C). **We recommend log-linear models when using $K \geq 3(m+1)$ proxy runs**, aligning the model-family choice with the swarm-size finding above.

### 3.2.5. RQ4: Data-Constrained Settings

**How do we mix under finite data constraints?** In practice, LM training is often data-constrained: the target model's requested training tokens $R$ exceed the available data. In this

---

[1]Investigating the public RegMix code, we found their 1M implementation is closer to 15M, which our results suggest is a good proxy size.

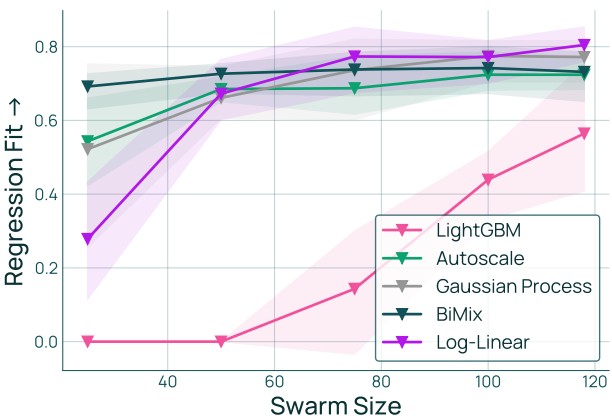

*Figure 5.* Fit versus Swarm Size across Regression Models. Different models require different amounts of data, and the log-linear model achieves the best regression fit (§3.2.4).

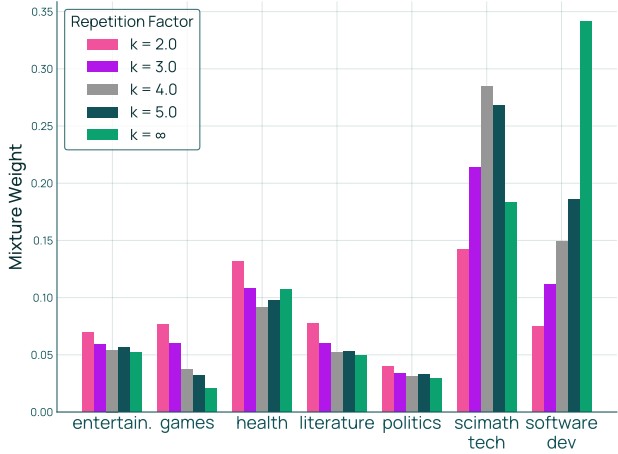

*Figure 6.* Proposed Mixture Weights versus Repetition Factor. Enforcing varying caps on repetition $k$ leads to significantly different proposed mixes (§3.2.5).

regime, some mixtures cause excessive sample repetition, degrading performance (Muennighoff et al., 2025). For example, a mixture that proposes 40% code even when code comprises only 5% of the available data would oversample code data 8 times. Existing mixing methods assume compute-constrained settings with effectively infinite data and provide no way to control repetition.

How might we adapt the offline mixing schema to a data-constrained setting? We study the effects of enforcing repetition constraints in the mixing objective (minimizing BPB); that is, recasting the third step of the offline schema as a constrained optimization problem with an additional term $p_j \leq kN_j/R$, for $j \in [m]$. $k$ is the repetition factor, the max allowable repetitions for any document.

Figure 6 shows how the proposed mix is impacted smoothly by repetition constraints during mix optimization. As $k$ increases, high-utility domains rapidly increase while low-utility domains decrease, showing that repetition constraints significantly impact proposed mixes. **We therefore recommend incorporating repetition constraints directly into mixture optimization, since accounting for the effects of finite data can substantially alter the proposed mix.** Table 5 in Appendix C.6 further shows that where these constraints are encoded (e.g., in optimization versus swarm construction) can also affect constraint satisfaction and performance.

### 3.3. Configuring OLMIXBASE

Our results in §3.1.2 let us define OLMIXBASE, a concrete configuration of the offline mixing schema (Table 2; Algorithm 1). Selected configurations resulting from our findings are highlighted in blue[2].

---

[2]The KL term in (1) is based on our study of the optimization solver in Appendix B.3.

---

**Algorithm 1** OLMIXBASE

1: **Input:** Domains $\mathcal{D} = \{D_1, \ldots, D_m\}$, swarm size $K = \mathcal{O}(m)$, repetition factor $k$, requested tokens $R$, KL penalty $\lambda$, natural distribution $p_0 \propto \{N_1, \ldots, N_m\}$.
2: Sample mixes $p^1, \ldots, p^K \sim \mathcal{P}$ on $\mathcal{D}$.
3: Train proxy models $S_{\text{small}} > 15\text{M parameters}$ on these mixes, and evaluate on downstream tasks to get a dataset of mixes and performance, $\{(p^j, \{y_{ij}\}_{i=1}^n)\}_{j=1}^K$, where $y_{ij} := f_i(\text{LM}(S_{\text{small}}, R_{\text{small}}, p^j))$.
4: **for** $i \in [n]$ **do**
5:     Use $\{(p^j, y_{ij})\}_{j=1}^K$ to fit the log-linear model $\hat{f}_i(p) = c_i + \exp(A_i^\top p)$, where $c_i \in \mathbb{R}^+$ and $A_i \in \mathbb{R}^m$.
6: **end for**
7: Solve the optimization problem:

$$\text{minimize}_{p \in \triangle^{m-1}} \frac{1}{n} \sum_{i=1}^n \hat{f}_i(p) + \lambda D_{\text{KL}}(p \| p_0) \quad (1)$$

$$\text{subject to} \quad p_j \leq \frac{kN_j}{R} \, \forall j \in [m]$$

8: **Return** $p^\star$, the solution to (1).

## 4. The Evolving Domain Problem

From §3, OLMIXBASE assumes a fixed domain set $\mathcal{D}$. Here, we study the *evolving domain problem* encountered throughout LM development. We first define the problem setting (§4.1). Then, we present our method, FULLMIXTUREREUSE (§4.2), and theoretical analysis of it (§4.3). We introduce PARTIALMIXTUREREUSE as a coupling-aware extension (§4.4), and we evaluate these methods empirically in §4.5.

### 4.1. Problem Setup

During LM development, the domain set evolves from $\mathcal{D}$ to an updated set $\mathcal{D}' = \{D_1', \dots, D_{m'}'\}$ of size $m'$, where each domain $D_i'$ now has $N_i'$ tokens. We use $q \in \triangle^{m'-1}$ to express data mixtures over $\mathcal{D}'$. In practice, domain updates are typically localized, affecting only a subset of domains while leaving others unchanged. To capture this structure, we partition the domain sets as $\mathcal{D} = [\mathcal{D}_1, \mathcal{D}_2]$ and $\mathcal{D}' = [\mathcal{D}_1, \mathcal{D}_2']$, where $\mathcal{D}_1$ is the **unaffected domain set**, and $\mathcal{D}_2$ is the **affected domain set** that is transformed into $\mathcal{D}_2'$.

In studying the data development practices in long-run LM development projects like OLMo 1–3 (Olmo et al., 2025) and SmolLM 1–3 (Bakouch et al., 2025), we identified four patterns of domain updates, which we call **domain update operators**, that describe how $\mathcal{D}_2$ transforms into $\mathcal{D}_2'$:

- `Add`: $\mathcal{D}_2 = \emptyset$ and $\mathcal{D}_2'$ contains the newly added domains. This occurs when new datasets are created or acquired.

- `Remove`: $\mathcal{D}_2$ contains one or more domains and $\mathcal{D}_2' = \emptyset$. This can occur, for instance, when domains are discarded due to low utility.

- `Partition`: $\mathcal{D}_2$ consists of one domain that is split into multiple subdomains in $\mathcal{D}_2'$ such that $\mathcal{D}_2 = \bigcup_{D_i' \in \mathcal{D}_2'} D_i'$. Partitioning is commonly used to obtain finer-grained mixtures, which can improve performance (Wettig et al., 2025; Diao et al., 2025; Ge et al., 2025a).

- `Revise`: $\mathcal{D}_2$ is one domain that is modified to produce the domain in $\mathcal{D}_2'$. This occurs when contents of the dataset are modified, such as samples being reformatted or rewritten (Team et al., 2025; Maini et al., 2024).

**Problem.** We have an existing mixture $\tilde{p} \in \triangle^{m-1}$ over the current domain set $\mathcal{D}$ proposed through a procedure like OLMIXBASE. But now, the domain set has evolved from $\mathcal{D}$ to $\mathcal{D}'$, and our goal is to compute an updated optimal $q^\star \in \triangle^{m'-1}$ on $\mathcal{D}'$:

$$\text{minimize}_{q \in \triangle^{m'-1}} \frac{1}{n} \sum_{i=1}^{n} f_i(\text{LM}(S,R,q)) \qquad (2)$$

$$\text{s.t.} \quad q_j \leq \frac{kN_j'}{R} \quad \forall j \in [m']$$

**Baseline: full recomputation.** A direct approach is to apply OLMIXBASE to $\mathcal{D}'$ after each update, requiring $\mathcal{O}(m')$ proxy runs per update (§3.2.3). Since this cost accumulates across repeated updates, we ask whether the existing mix $\tilde{p}$ can be reused to maintain performance with fewer proxy runs.

### 4.2. Mixture Reuse

Our goal is to leverage the existing mixture $\tilde{p}$ to efficiently compute a strong mixture on an updated domain set $\mathcal{D}'$ without full recomputation over all $m'$ domains. Our core idea is to freeze the relative ratios among the unaffected domains according to $\tilde{p}$, aggregate them into a single *virtual domain*, and only recompute its total weight along with the weights of the affected domains $\mathcal{D}_2'$. This reduces the dimensionality of the optimization from $m'$ to $1 + |\mathcal{D}_2'|$. This mechanism is presented in Figure 7 (right).

For example, suppose $\mathcal{D}$ contains $m = 3$ domains with mixture $\tilde{p} = [0.25, 0.25, 0.5]$, and $\mathcal{D}'$ is formed by adding one new domain. Instead of learning a 4-dimensional mixture, we learn a 2-dimensional mixture over the virtual domain and the new domain. If that mixture is $[0.4, 0.6]$, we can unravel it to induce the mixture over the $m' = 4$ domains:

$$q^* = 0.4 \cdot [0.25, 0.25, 0.5] \cup [0.6] = [0.1, 0.1, 0.2, 0.6].$$

#### 4.2.1. FORMALIZING THE MIXTURE REUSE PROBLEM

The example above provides the intuition behind mixture reuse, which we formalize in the mixture reuse problem.

First, we introduce notation to handle domains we will reuse versus recompute. Define $\mathcal{D}_{\text{fix}} := \mathcal{D}_1$ as the unaffected domains whose relative ratios we will keep fixed from $\tilde{p}$, and $\mathcal{D}_{\text{comp}} := \mathcal{D}_2'$ as the affected domains whose ratios we recompute. We remap these variables because, as seen later in §4.4, we may choose to recompute unaffected domains.

We partition any mix $q$ on $\mathcal{D}'$ as $q = [\rho q_{\mathcal{D}_{\text{fix}}}, (1-\rho) q_{\mathcal{D}_{\text{comp}}}]$, where $\rho \in [0,1]$ is the total weight on the unaffected domains, and $q_{\mathcal{D}_{\text{fix}}} \in \triangle^{|\mathcal{D}_{\text{fix}}|-1}$ and $q_{\mathcal{D}_{\text{comp}}} \in \triangle^{|\mathcal{D}_{\text{comp}}|-1}$ are the relative mixes within each domain subset. Similarly, we define $\tilde{p}_{\mathcal{D}_{\text{fix}}} \in \triangle^{|\mathcal{D}_{\text{fix}}|-1}$ as the normalized ratios from $\tilde{p}$ restricted to $\mathcal{D}_{\text{fix}}$.

With this notation, the mixture reuse problem is the following approximation of (2):

$$\text{minimize}_{q \in \triangle^{m'-1}} \frac{1}{n} \sum_{i=1}^{n} f_i(\text{LM}(S,R,q)) \qquad (3)$$

$$\text{s.t.} \quad q_j \leq \frac{kN_j'}{R} \quad \forall j \in [m']$$

$$q_{\mathcal{D}_{\text{fix}}} = \tilde{p}_{\mathcal{D}_{\text{fix}}}$$

The new constraint $q_{\mathcal{D}_{\text{fix}}} = \tilde{p}_{\mathcal{D}_{\text{fix}}}$ enforces that the relative ratios among $\mathcal{D}_{\text{fix}}$ are the same as in $\tilde{p}$. The mixture reuse problem hence presents a tradeoff, which we analyze in §4.3: it has reduced dimensionality due to the constraint but is an approximation of the original problem.

#### 4.2.2. FULLMIXTUREREUSE

We now describe FULLMIXTUREREUSE, our approach for solving the mixture reuse problem (3) using OLMIXBASE. To handle the constraint $q_{\mathcal{D}_{\text{fix}}} = \tilde{p}_{\mathcal{D}_{\text{fix}}}$, we encode it through a change of variables. We aggregate all domains in $\mathcal{D}_{\text{fix}}$ into a single virtual domain and define

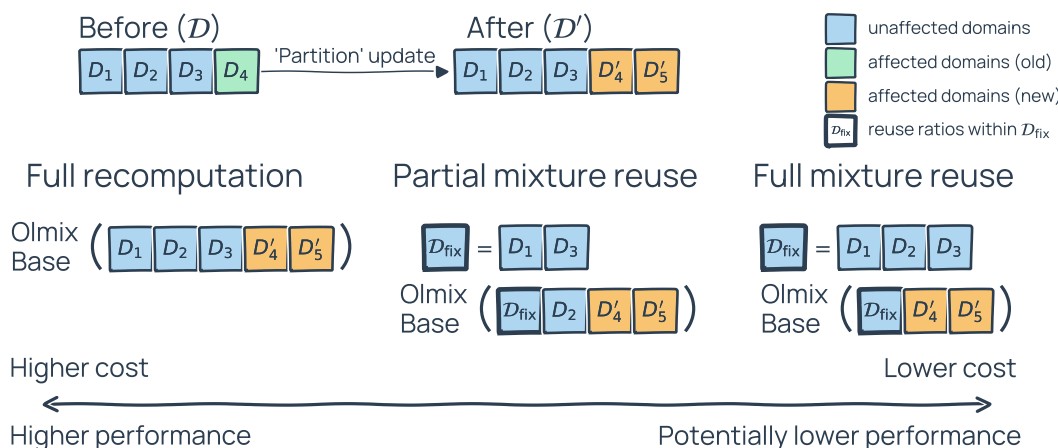

*Figure 7.* Overview of mixture recomputation strategies. After a domain update, full recomputation applies OLMIXBASE to all of $\mathcal{D}'$, while FULLMIXTUREREUSE (§4.2) and PARTIALMIXTUREREUSE (§4.4) freeze ratios for reused domains $\mathcal{D}_{\text{fix}}$ and recompute lower-dimensional mixtures, yielding a cost-performance spectrum.

$r = [\rho, (1-\rho)q_{\mathcal{D}_{\text{comp}}}] \in \triangle^{|\mathcal{D}_{\text{comp}}|}$, where $\rho$ is the weight on the whole of $\mathcal{D}_{\text{fix}}$ and the remaining entries correspond to $\mathcal{D}_{\text{comp}}$. From our earlier example, $\rho = 0.4$ and $q_{\mathcal{D}_{\text{comp}}} = [1]$. Any $r$, given $\tilde{p}_{\mathcal{D}_{\text{fix}}}$, can be expanded into a mixture $q$ over $\mathcal{D}'$ that satisfies the constraint. This transforms the mixture reuse problem into standard mixing over $1 + |\mathcal{D}_{\text{comp}}|$ domains. We can now apply OLMIXBASE (Algorithm 1) over $r$, which requires only $\mathcal{O}(|\mathcal{D}_{\text{comp}}|)$ swarm runs instead of $\mathcal{O}(m')$:

**Step 1: Swarm construction.** We sample $K$ mixture weights $r^1, \ldots, r^K \in \triangle^{|\mathcal{D}_{\text{comp}}|}$. For each $r^j$, we expand it into $q^j$, train a proxy model using $q^j$, and evaluate to obtain performance $\{y_{ij}\}_{i=1}^n$.

**Step 2: Regression model.** As in OLMIXBASE, for each task $i \in [n]$ we fit a log-linear regression model: $\hat{g}_i(r) = d_i + \exp(B_i^\top r)$, where $d_i \in \mathbb{R}^+, B_i \in \mathbb{R}^{|\mathcal{D}_{\text{comp}}|}$.

**Step 3: Mixture optimization.** Solve:

$$\text{minimize}_{r \in \triangle^{|\mathcal{D}_{\text{comp}}|}} \frac{1}{n}\sum_{i=1}^n \hat{g}_i(r) + \lambda D_{\text{KL}}(r\|r_0) \quad (4)$$

$$\text{s.t.} \quad \rho \leq \min_{j \in \mathcal{D}_{\text{fix}}}\left\{\frac{kN_j'}{R\tilde{p}_j}\right\}, r_j \leq \frac{kN_j'}{R} \quad \forall j \in \mathcal{D}_{\text{comp}}$$

where $\lambda$ is the KL penalty and $r_0$ is the natural distribution. Let the solution be $r^\star(\tilde{p}_{\mathcal{D}_{\text{fix}}})$ and expand it into the proposed mix on $\mathcal{D}'$ as $q^\star(\tilde{p}_{\mathcal{D}_{\text{fix}}})$. The full approach is in Algorithm 2.

### 4.3. Theoretical Analysis

FULLMIXTUREREUSE reduces costs by reusing existing ratios, but when does it match versus degrade compared to full recomputation? We theoretically analyze this, finding that performance depends on (1) how much the optimal mix changes due to the domain update, and (2) a coupling effect between $\mathcal{D}_{\text{fix}}$ and $\mathcal{D}_{\text{comp}}$. Importantly, we empirically validate our theory in Appendix E.4: **we measure the terms in our bounds and show they tightly track actual performance gaps across settings.**

**Assumptions.** We make the following assumptions.

1. Log-linear model holds: For each task $i \in [n]$, the performance can be expressed as $f_i(\text{LM}(S, R, q)) = c_i + \exp(A_i^\top q)$ for some $c_i \in \mathbb{R}^+, A_i \in \mathbb{R}^{m'}$.

2. Full recomputation (using OLMIXBASE) minimizes (2), and FULLMIXTUREREUSE minimizes (3).

**Definitions.** Define performance of $q$ as $F(q) := \frac{1}{n}\sum_{i=1}^n f_i(\text{LM}(S, R, q))$. $q^\star$ is the optimal solution to (2), and $q^\star(\tilde{p}_{\mathcal{D}_{\text{fix}}})$ is the solution to (3). We study the *performance gap* of FULLMIXTUREREUSE, $F(q^\star(\tilde{p}_{\mathcal{D}_{\text{fix}}})) - F(q^\star)$.

Let $q_{\mathcal{D}_{\text{fix}}}^\star$ denote the optimal mix on $\mathcal{D}_{\text{fix}}$ after the domain set update, so that $q^\star = [\rho^\star q_{\mathcal{D}_{\text{fix}}}^\star, (1 - \rho^\star)q_{\mathcal{D}_{\text{comp}}}^\star]$. Let $\kappa(\mathcal{D}_{\text{fix}}, \mathcal{D}_{\text{comp}})$ denote a *coupling term* (Definition E.1), an expression over the log-linear model's coefficients that captures task influence of $\mathcal{D}_{\text{fix}}$ versus $\mathcal{D}_{\text{comp}}$.

#### 4.3.1. PERFORMANCE CHARACTERIZATION

Our first result characterizes the performance gap in terms of $\tilde{p}_{\mathcal{D}_{\text{fix}}}$ itself. See Appendix E.1 for the proof.

**Theorem 4.1** (Performance gap bound). *There exists a finite $C_1 > 0$ such that the performance gap is bounded by*

$$F(q^\star(\tilde{p}_{\mathcal{D}_{\text{fix}}})) - F(q^\star) \leq C_1 \kappa(\mathcal{D}_{\text{fix}}, \mathcal{D}_{\text{comp}})\|\tilde{p}_{\mathcal{D}_{\text{fix}}} - q_{\mathcal{D}_{\text{fix}}}^\star\|$$

The performance gap is controlled by two factors:

- $\|\tilde{p}_{\mathcal{D}_{\text{fix}}} - q_{\mathcal{D}_{\text{fix}}}^\star\|$ ("reuse gap"): how close is the reused mix to the optimal mix after the update.

- $\kappa(\mathcal{D}_{\text{fix}},\mathcal{D}_{\text{comp}})$: the coupling term is large when $\mathcal{D}_{\text{fix}}$ and $\mathcal{D}_{\text{comp}}$ impact the same set of downstream tasks and low when they impact separate tasks. It controls the rate at which performance depends on the reuse gap.

When both terms are small, FULLMIXTUREREUSE matches full recomputation. Our empirical validation (Appendix E.4, Figure 22) confirms that the reuse gap strongly predicts the performance gap across different mixing scenarios.

### 4.3.2. REUSE GAP

Theorem 4.1 shows the performance gap is governed by the reuse gap, but when is the reuse gap small? We analyze the case where $\tilde{p}$ is optimal before the update and domains are added. Proofs and results on other operators are in Appendix E.2.

**Theorem 4.2.** *Define $\tilde{p}$ as the solution to* (2) *on $\mathcal{D}$ and suppose that new domains are added. There exists a finite $C_2 > 0$ such that the reuse gap is bounded by*

$$\|\tilde{p}_{\mathcal{D}_{\text{fix}}} - q^\star_{\mathcal{D}_{\text{fix}}}\| \le C_2 \kappa(\mathcal{D}_{\text{fix}},\mathcal{D}_{\text{comp}})(1-\rho^\star)$$

The reuse gap is controlled by two factors:

1. $\kappa(\mathcal{D}_{\text{fix}},\mathcal{D}_{\text{comp}})$: the coupling term, also in Theorem 4.1.

2. $1 - \rho^\star$: This term captures how much newly added domains move the optimum, since 1 is effectively $\rho$ induced by $\tilde{p}$ before domains are added. This term can be small when: 1) the added domains are low utility, or 2) the added domains are small, which constrains their max weight in mixture optimization.

Our empirical validation (Appendix E.4, Figure 23-26) shows that the success of FULLMIXTUREREUSE is correlated with small $1-\rho^\star$ and low coupling.

### 4.4. PARTIALMIXTUREREUSE

Our analysis (§4.3) indicates that FULLMIXTUREREUSE can underperform full recomputation when coupling between $\mathcal{D}_{\text{fix}}$ and $\mathcal{D}_{\text{comp}}$ is high, which we empirically confirm in Appendix E.4. We propose PARTIALMIXTUREREUSE (Figure 7 center) as a middle ground: rather than reusing all unaffected domains, it selectively recomputes some, reducing coupling while keeping costs below full recomputation.

**Method.** Let $\mathcal{D}_{\text{partial}} \subset \mathcal{D}_1$ be a subset of unaffected domains to reuse. We redefine the domains we reuse versus recompute as $\mathcal{D}_{\text{fix}} := \mathcal{D}_{\text{partial}}$ and $\mathcal{D}_{\text{comp}} := (\mathcal{D}_1 \setminus \mathcal{D}_{\text{partial}}) \cup \mathcal{D}'_2$, and then apply FULLMIXTUREREUSE (Algorithm 2) with this new partition. The mixing problem has dimension $1 + |\mathcal{D}'_2| + |\mathcal{D}_1 \setminus \mathcal{D}_{\text{partial}}|$, interpolating between FULLMIXTUREREUSE's $1 + |\mathcal{D}'_2|$ and full recomputation's $m'$. Since swarm cost scales linearly with dimension (RQ2), this directly interpolates computational cost.

**Interpretation and Empirical Validation** Carefully choosing $\mathcal{D}_{\text{partial}}$ can reduce the coupling $\kappa(\mathcal{D}_{\text{fix}}, \mathcal{D}_{\text{comp}})$ from Theorem 4.1. An intuitive example: when adding code data to web topics, the web source's software development topic should be recomputed alongside it since both strongly influence code evaluation tasks. Figure 25 in Appendix E.4 confirms that this reduces coupling and improves performance compared to reusing all web topics.

### 4.5. Experimental Results

We evaluate FULLMIXTUREREUSE and PARTIALMIXTUREREUSE—with recomputation done via OLMIXBASE—across a sequence of five domain updates mirroring real-world LM development. Our results show that these methods maintain strong performance while substantially reducing computational costs compared to full recomputation. See Appendix F for full experiment details.

### 4.5.1. SETUP

**Domain Updates** We simulate a real-world LM development workflow with an initial Common Crawl web corpus that undergoes 5 updates (Table 1; full details in Table 7). The final domain set contains 64 domains (token counts in Table 6).

*Table 1.* Simulated domain updates for §4.5 experiments.

| Operation | Dataset(s) | $\Delta m$ |
|---|---|---|
| Initial | WebOrganizer over Common Crawl | 24 |
| Add | New code dataset | +15 |
| Add | Six PDF and math datasets | +6 |
| Revise | Filter low quality PDF documents | 0 |
| Remove | Low quality math dataset | −1 |
| Partition | WebOrganizer over PDF dataset | +20 |

**Methods compared.** We consider:

- Natural: mix proportional to domain sizes.

- Full recomputation: apply OLMIXBASE after each update (high performance, high cost).

- Swarm reuse: reuse all accumulated proxy runs that can be represented on the current $\mathcal{D}'$ and apply OLMIXBASE on the combined swarm (see Alg. 3).

- FULLMIXTUREREUSE (our method): reuse ratios for unaffected domains, recompute affected ones.

- PARTIALMIXTUREREUSE (our method): reuse ratios within DCLM topics and within Stack-Edu languages while recomputing at the source level; recompute DCLM's software development ratio when adding Stack-Edu due to high coupling (see Appendix E.4, Figure 25 for justification).

**Experimental protocol.** For full recomputation, we use swarm size $K \approx c(m+1)$ and vary $c \in \{1,2,3\}$ to showcase

different compute regimes. For swarm reuse and mixture reuse, we set swarm sizes to roughly match that of full recomputation with $c=1$. We report results over 3 random seeds of swarms with $k=4$ and $R=1T$. Details are in Appendix F.1.

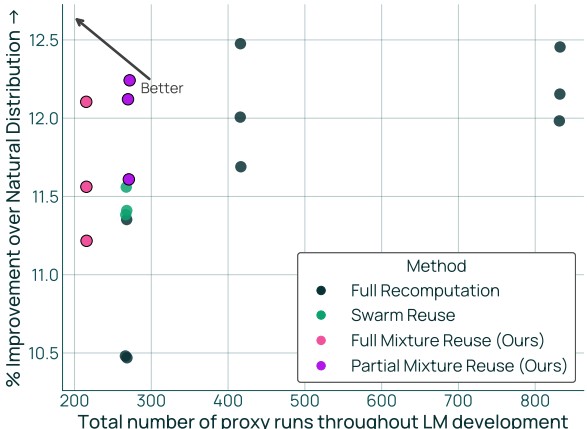

*Figure 8.* Performance improvement versus cost of mixing under evolving domains (§4.5.2).

### 4.5.2. RESULTS

See App. F.2 for further results on 1) $R=6T$, 2) smaller proxy run budgets, and 3) individual domain update operators.

**FULLMIXTUREREUSE matches full recomputation at substantially lower cost.** Figure 8 shows relative improvement in downstream performance (average BPB) over the natural distribution versus the total number of proxy runs across all 5 updates. At $c=3$, FULLMIXTUREREUSE achieves 95% of full recomputation's performance improvement (+11.6% versus +12.2%) while using 74% fewer total proxy runs (216 versus 832 runs).

**PARTIALMIXTUREREUSE helps close the remaining performance gap to full recomputation.** By selectively recomputing some unaffected domains, PARTIALMIXTUREREUSE achieves 98% of full recomputation's performance (+12%) with 272 runs—still 67% fewer than full recomputation.

**Mixture reuse outperforms swarm reuse.** Swarm reuse achieves +11.4% with 268 runs, performing worse and requiring more runs than FULLMIXTUREREUSE (+11.6% with 216 runs). This is likely because (1) representing old swarms on updated domain sets over-explores biased subspaces, and (2) swarms cannot be reused when domains are removed or revised.

**At matched proxy-run budgets, mixture reuse outperforms alternatives.** At a budget of 216–272 proxy runs, FULLMIXTUREREUSE and PARTIALMIXTUREREUSE achieve +11.6% and +12.0% improvement over the natural distribution, respectively; at the same budget, full recomputation at $c=1$ reaches +10.8%.

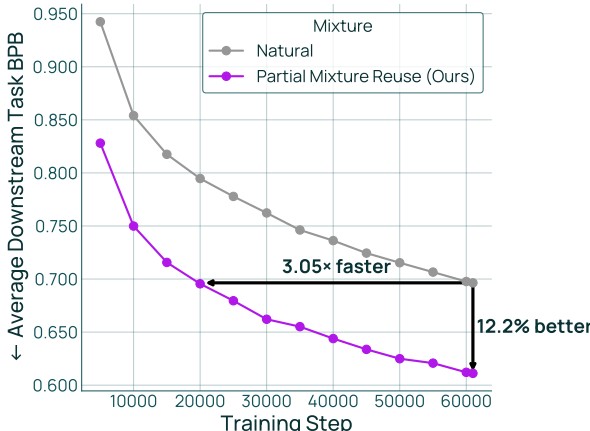

*Figure 9.* Downstream performance across training for PAR-TIALMIXTUREREUSE (best seed) versus the natural distribution.

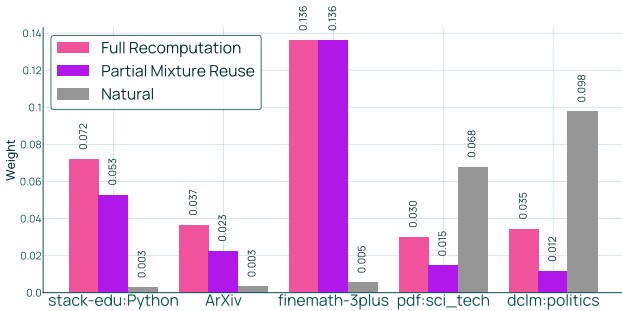

*Figure 10.* Proposed mixes for different mixing strategies. Domains shown have the greatest difference in ratios between full recomputation and natural.

**Our best mixture is $3.05\times$ more data-efficient than the natural distribution.** PARTIALMIXTUREREUSE reaches the natural distribution's final BPB in roughly 20K training steps versus 61K steps for the natural distribution (Figure 9).

**Qualitative analysis of learned mixtures.** Figure 10 shows the final proposed mixes (full mix in Table 13). PARTIALMIXTUREREUSE is substantially closer to full recomputation than to the natural distribution, with total variation distances of 0.067 and 0.127, respectively. This aligns with our findings on target model downstream performance.

## 5. Discussion

We presented OLMIX, addressing two key challenges in data mixing for language models: configuring effective mixing methods and efficiently updating mixtures as domain sets evolve. We conducted an empirical study of design choices and introduced mixture reuse. In future work, we aim to extend these ideas to online mixing and encourage co-design of mixing methods with other LM development workflows.

## Acknowledgments

We thank Neel Guha, Junmiao Hu, Ronny Junkins, Pang Wei Koh, Jerry Liu, Roberto Garcia Torres, Alex Wettig, Steven Euijong Whang, and Michael Zhang for helpful feedback and discussions. We thank the Olmo 3 team, particularly Allyson Ettinger, for iterative feedback and experimental validation of Olmix. MC was supported in part by the Office of Naval Research (ONR) under No. N000142312633 (Deep Signal Processing). Any opinions, findings, and conclusions or recommendations expressed in this material are those of the authors and do not necessarily reflect the views, policies, or endorsements, either expressed or implied, of ONR or the U.S. Government.

## Impact Statement

**Intended Use and Benefits.** Data mixing methods typically require many proxy training runs, making misconfiguration costly and limiting their use to researchers and developers without substantial compute resources. By systematically studying the design space of data mixing methods, we identify configurations that achieve strong performance with lower computational cost. In addition, our approach to handling evolving domain sets enables researchers and practitioners to update data mixtures efficiently without full recomputation. We intend for this work to be both a detailed guide to follow for data mixing, as well as motivation to invest more in accessible model development, particularly for those operating under constrained data or compute budgets.

**Potential Risks and Limitations.** The techniques presented in this paper do not introduce new model capabilities or application-facing behaviors. However, data mixing directly shapes the effective training distribution by emphasizing certain domains at the expense of others. Under a fixed total token budget, increasing weight on selected domains necessarily reduces exposure to the remaining data, potentially decreasing diversity relative to more uniform mixtures. While this ability to control the training distribution is a strength of data mixing as a technique in a practitioner's toolkit, it also introduces risks if the optimization objective is poorly specified or misaligned with desired model behavior. Data mixing is a powerful intervention with respect to a chosen set of downstream tasks, and its use requires practitioners to carefully consider which objectives are being optimized and which types of data or behaviors may be deprioritized as a result. Responsible use therefore depends on clear task definitions, informed dataset design, and awareness of the tradeoffs introduced by sharpening the training distribution.

Further, our target-model experiments use 1B parameter models. While necessary given we needed to train at our target scale to validate proxy-to-target generalization, we indeed lack validation for larger target scales. We expect our qualitative recommendations to use proxy-target rank correlation as a calibration test to remain useful, but practitioners training larger target models may need larger proxies and should re-estimate the smallest reliable proxy size in their own setting.

Finally, PARTIALMIXTUREREUSE requires choosing which unaffected domains to recompute ($\mathcal{D}_{\text{partial}}$). Theorem 4.1 specifies the objective—reused domains should be weakly coupled to recomputed ones—but we do not have a principled procedure for selecting $\mathcal{D}_{\text{partial}}$ to minimize this coupling, since estimating $\kappa$ directly would require the recomputation that reuse is meant to avoid. We instead rely on two practitioner-facing heuristics that sufficed in our experiments: recomputing domains of the same type as the update (e.g., DCLM software development and Stack-Edu), and recomputing "maxed-out" domains whose allocation already exhausts their available tokens (since freezing their ratio leaves high-utility tokens unused). These proxies are not guaranteed to identify the best subset to recompute, and a principled, cost-aware selection procedure remains an open direction.

**Broader Societal Considerations.** By reducing the computational cost of data mixing, this work may have positive environmental and accessibility impacts, enabling more efficient use of compute and lowering barriers to iterative model development. At the same time, increased efficiency may make large-scale language model training more accessible to a wider range of actors.

## Author Contributions

Mayee F. Chen led the project during an internship at Ai2: she developed the theoretical framework, implemented all methods, designed and ran all experiments, validated the approach at scale for Olmo 3, and wrote the paper. Kyle Lo and Luca Soldaini were the primary mentors and worked closely with Mayee on developing the methodology, designing experiments, integrating into Olmo 3, and writing the paper. Tyler Murray built and maintained the training infrastructure that enabled large-scale experimentation. David Heineman designed small-scale evaluation metrics for data mixing. Matt Jordan contributed to mixing methodology. Hannaneh Hajishirzi and Christopher Ré provided research guidance and helped frame the work.

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

# A. Extended Related Work

**Mixing Methods.** The offline mixing schema, also described as Function Fitting-based Methods in Liu et al. (2025b) has been extensively used in existing mixing methods. In Table 2, we provide an overview of several offline mixing methods and how they address the key design choices we study; however, there is significant variation. Some methods use explicit parametric regression models (Que et al., 2024), while others use nonparametric approaches like LightGBM and Gaussian Processes (Liu et al., 2025a; Chen et al., 2025b). Some methods explicitly model the role of proxy model size or training budget (Ge et al., 2025b; Ye et al., 2025; Kang et al., 2025) while others assume direct generalization from proxy models to target models (Held et al., 2025). There are also several methods that build on top of RegMix, exploring how it can be augmented with better domains (Wettig et al., 2025), iterative swarms (Diao et al., 2025), or more features (Belenki et al., 2025). Our work examines key design choices for the offline schema that underlies all these methods.

Aside from the offline schema, DoReMi (Xie et al., 2023) and DoGE (Fan et al., 2024) produce a set of mixture weights by using one or two proxy runs that dynamically explore the mixture weight space; this is in contrast to using many proxy runs with static mixes. While these approaches have proven effective in some settings, previous analysis (Chen et al., 2025a) has suggested some suboptimality in how the dynamic update rules are constructed, and the offline schema is generally considered simpler to implement.

Online mixing methods (Chen et al., 2023; Jiang et al., 2024; Albalak et al., 2023) adjust the mixture weights throughout the final training run. Rather than exploring the mixture space and learning from it offline, these methods implicitly do this on the fly during training. Note that our notion of dynamics, which is over the domain set and during LM development (i.e., before the final training run), is different from the dynamic aspect of online mixing methods.

Existing mixture literature largely assumes fixed domain sets. The recent work Chameleon (Xie et al., 2025) addresses adaptability to domain changes by computing domain weights from learned embeddings using kernel ridge leverage scores, which allows direct transfer to new data without proxy retraining. However, Chameleon focuses on adding new domains and computes weights directly from domain embeddings without the swarm-based regression approach of offline methods. In contrast, our mixture reuse approach explicitly handles various domain update operations (additions, removals, revisions, and partitions) and is designed to work within the offline mixing schema. Moreover, our approach provides theoretical and empirical analysis of when and why existing mixes can be reused.

**Data-constrained settings.** Several works have addressed training language models under data constraints. UniMax (Chung et al., 2023) proposes an allocation algorithm for multilingual pretraining that distributes a fixed token budget to maximize uniform coverage across languages while capping repetitions to avoid overfitting on low-resource languages. However, their allocation procedure is not integrated with data mixing methods and focuses specifically on the multilingual setting. Muennighoff et al. (2025) empirically study scaling laws when data is limited and must be repeated, finding that up to 4 epochs of repetition yields negligible degradation and proposing modified scaling laws that account for diminishing returns of repeated tokens. While these works address data constraints through allocation procedures and scaling laws, our work integrates repetition constraints directly into data mixing, enabling principled allocation of limited data budgets while producing a mix that yields strong downstream performance.

**LM Data Development.** Real-world LM development involves iterative refinement of training data. Works like DCLM (Li et al., 2024a), Dolma (Soldaini et al., 2024), and FineWeb (Penedo et al., 2024) extensively document the curation processes involved in creating high-quality pretraining corpora, including quality filtering, deduplication strategies, and careful domain selection. Continuous projects like OLMo 1-3 (Groeneveld et al., 2024; OLMo et al., 2024; Olmo et al., 2025) and SmolLM 1-3 (Ben Allal et al., 2024; Allal et al., 2025; Bakouch et al., 2025) showcase how training data evolves over time through these updates. For instance, the OLMo series documents changes in domain composition, filtering strategies, and data sources across model versions, while SmolLM similarly shows iterative improvements to the training mixture. These projects illustrate that domain sets are not static but undergo frequent revisions including dataset additions (e.g., new web crawls), removals (e.g., deprecated sources), quality improvements (e.g., enhanced filtering), and domain partitioning (e.g., splitting broad categories into specialized subdomains). Motivated by these works, OLMIX views data mixing as an ongoing process throughout LM development, providing both systematic methods for initial mixture configuration and efficient strategies for maintaining strong mixtures as domain sets evolve.

*Table 2.* Design choices across offline mixing methods. For each design choice and mixing method, we identify the method's configuration. We shade cells in pink if we found empirical justification for their configuration. In the final column, we present the configuration for OLMIXBASE.

| Design Choice | RegMix (Liu et al., 2025a) | DML (Ye et al., 2025) | AutoScale (Kang et al., 2025) | BiMix (Ge et al., 2025b) | ADMIRE-BayesOpt (Chen et al., 2025b) | CLIMB (Diao et al., 2025) | OLMIXBASE (Algorithm 1) |
|---|---|---|---|---|---|---|---|
| **Swarm Construction** | | | | | | | |
| Proxy model size | 1M | 70, 160, 305, 410M | Target | 280M | 1M, 60M | 350M | 30M |
| Swarm size (vs $m$ domains) | 512 $(m=17)$ | 20 $(m=7)$ | $2m+1$ | 4 | 101 $(m=17)$ | 112 $(m=21)$ | $3(m+1)$ |
| Swarm distribution | Dirichlet with natural prior | Exponential grid | Exponential grid | Entropy-weighted | Dynamic | Dirichlet with natural prior | Dirichlet with natural prior (sparse for topics, dense for sources) |
| **Regression Model** | | | | | | | |
| Regression model family | LightGBM | Log-Linear | Power Law | Power Law | Gaussian Process | LightGBM | Log-Linear |
| Regression granularity | Aggregated | Aggregated | Per-Task | Per-Task | Aggregated | Aggregated | Per-Task |
| **Mixture Optimization** | | | | | | | |
| Data repetition constraints | No | No | No | No | No | No | Yes |
| Optimization solver | Search | Search | Gradient Descent | Exact Solver | Search | Search | Exact Solver with KL reg. |

# B. Additional Design Choices

In addition to RQ1-4 in §3.2, we identify several other design choices in the offline mixing schema. We empirically study each design choice and report our findings.

We conduct an empirical study where we explore each OLMIX design choice and present our procedure on how to decide on each of them. Appendix C provides the exact experimental setup for each research question.

## B.1. RQ6: Swarm distribution

*How should we specify the distribution $\mathcal{P}$ to sample the mixes for the swarm runs?*

We study the distribution $\mathcal{P}$ from which swarm mixtures should be sampled. The distribution determines how effectively the swarm explores the mixture space; a well-chosen distribution that concentrates samples in informative regions can potentially improve performance for a fixed number of swarm runs.

We investigate two aspects of the swarm distribution. To evaluate each aspect, we measure the average BPB of 1B target models trained on the proposed mixes from these swarms. We also report the regression fit (Pearson correlation between predicted and true per-task BPB) of the regression model on a test set of mixtures unseen by any of the swarms.

- *First, should each swarm run include samples from all domains, or should some runs focus on subsets of domains?* We refer to these as sparse and dense distributions, respectively. Sparse distributions enable discovering that certain domains should be excluded, while dense distributions ensure all domains are represented in all swarm runs. We generate sparse and dense swarms at the **topic level** (on DCLM topics) and at the **source level** to capture both fine-grained and coarse-grained domains.

- *Second, does centering the swarm around a promising mixture region improve results?* We test three Dirichlet priors on DCLM topics: 1) the natural distribution based on domain token counts, 2) a strong prior (a mix known to perform well), and 3) a weak prior (a mix known to perform poorly).

Figure 11 (left) shows that the sparse swarm outperforms the dense swarm at the topic level and vice versa at the source level. The regression fit on held-out mixtures (Figure 11 right) follows the same pattern: sparse swarms achieve higher accuracy at

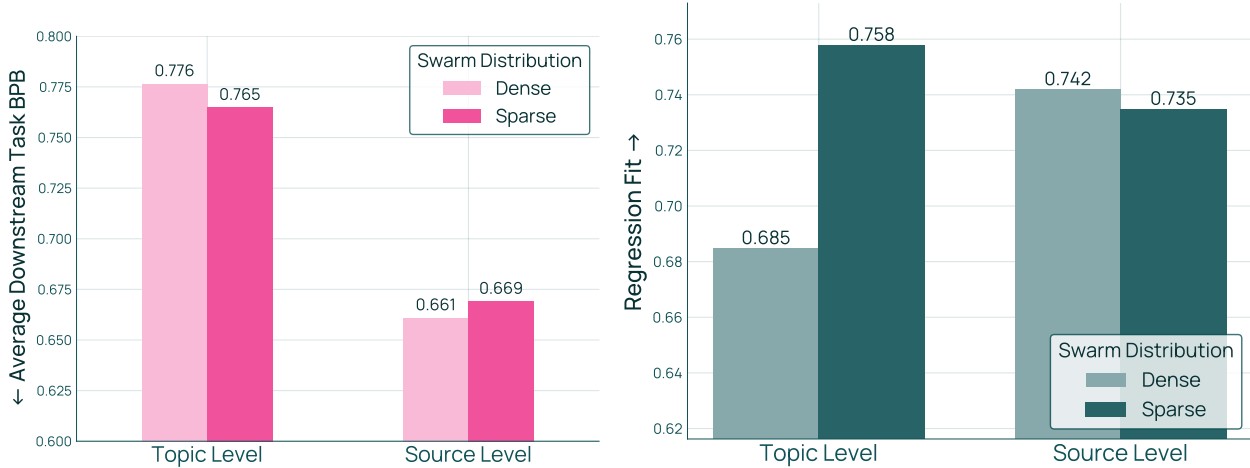

*Figure 11.* Performance (left) and regression fit (right) of dense versus sparse swarm distributions. While sparse swarms perform better at the topic level, dense swarms perform better at the source level, suggesting that this design choice is data-dependent.

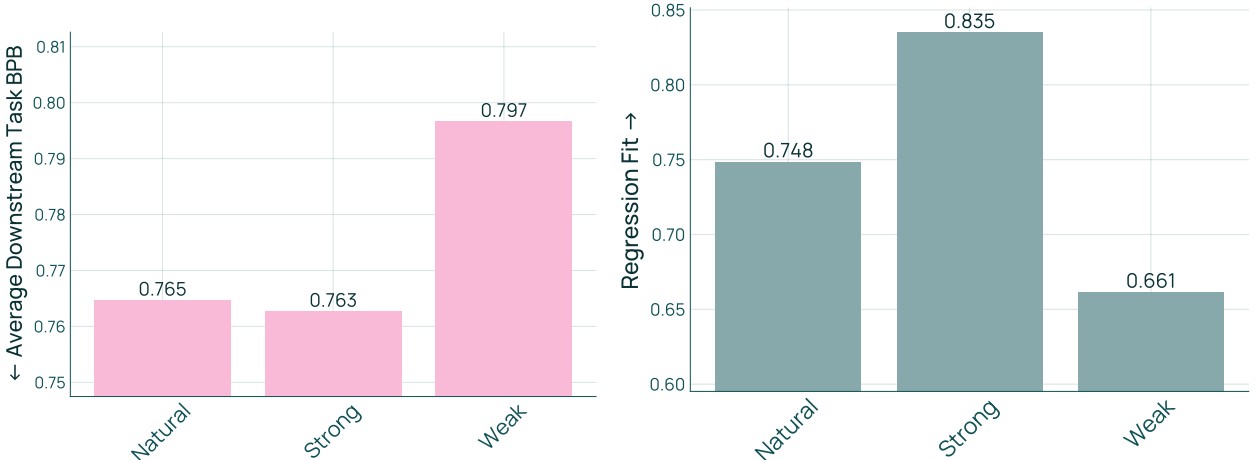

*Figure 12.* Performance (left) and regression fit (right) across Dirichlet priors. Natural and strong priors achieve roughly comparable downstream performance, but the weak prior does considerably worse, suggesting that this design choice is fairly robust and that the natural distribution is a reasonable configuration.

the topic level, while dense swarms achieve slightly higher accuracy at the source level. Neither sparse nor dense is universally better—the best swarm distribution depends on the domain set. One hypothesis is that the best mixes over topics exclude certain low-signal topics (e.g., adult content in DCLM), while the best mixes over sources utilize all sources. This hypothesis aligns with how these domain sets were constructed: DCLM was partitioned into topics that are potentially uninformative, while the sources we train on are intentionally curated with the expectation that they contribute signal. **We find that the choice of swarm distribution depends on the domain set—we recommend using sparse distributions for topic-level mixing and dense distributions for source-level mixing.**

Figure 12 shows that using a strong Dirichlet prior can improve the performance of the proposed mix. While centering on the natural and strong prior resulted in roughly similar downstream performance, using the weak prior significantly degraded performance. This indicates that a strong prior can improve or maintain mix quality, while a poor prior causes substantial harm. **We recommend that practitioners leverage domain knowledge to center the swarm around strong priors when available, as this can provide benefits without added computational cost. However, if prior knowledge is uncertain, the natural distribution remains a reasonable fallback.**

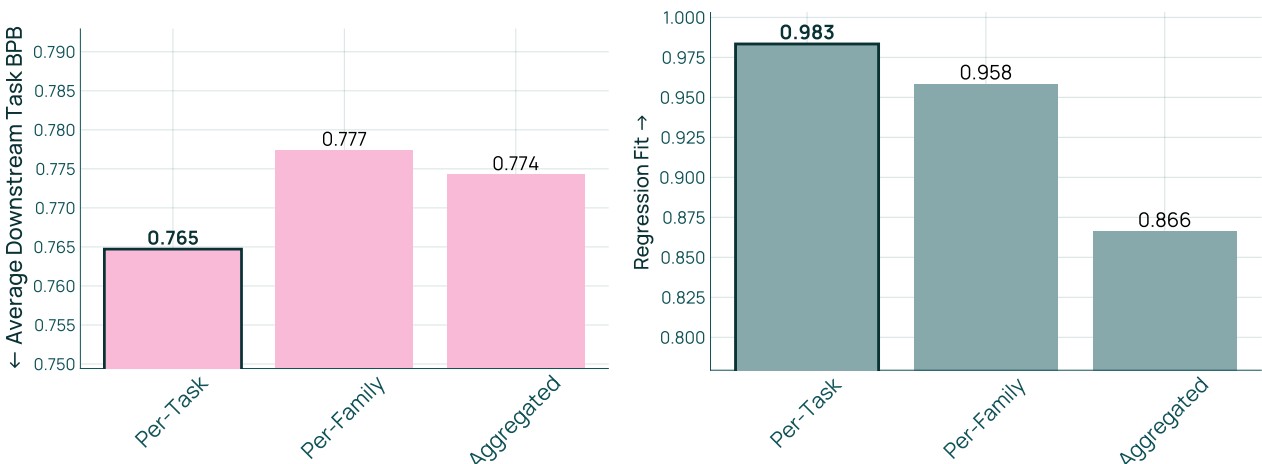

*Figure 13.* Performance (left) and regression fit (right) across regression granularities. As granularity of the regression target increases, the regression fit improves, and the downstream performance also trends toward improvement.

## B.2. RQ7: Regression granularity

*At what granularity should we fit the regression models in order to construct $\hat{f}(p)$?*

We study the granularity at which to fit regression models, considering one model per task, one model for the entire average BPB, and one model per task family (e.g., math, code, and QA). This choice involves a tradeoff between expressivity and noise. Per-task and per-family regression models can allow us to capture distinct relationships between mixtures and various capabilities, which increases the expressivity of the overall function approximating average BPB. For instance, under log-linear regression, using one model to approximate average BPB constrains the mixture optimization objective to be the form $c + \exp(A^\top p)$, while the per-task models allow for the form $c + \sum_{i=1}^{n} \exp(A_i^\top p)$. However, finer-grained models are fit to noisier individual measurements, while directly modeling only the average BPB may involve less noisy targets.

We compare three approaches. In the **per-task** approach, we fit individual functions $\hat{f}_i(p)$ using $\{(p^j, y_{ij})\}_{j=1}^{K}$ for each task $i \in [n]$, and then minimize the objective $\frac{1}{n}\sum_{i=1}^{n} \hat{f}_i(p)$. In the **per-family** approach, we group tasks into three families $F_1, F_2, F_3$: math, code, and QA. We fit one model $\bar{f}_i(p)$ to the average BPB of each family using $\{(p^j, \frac{1}{|F_i|}\sum_{k \in F_i} y_{kj})\}_{j=1}^{K}$ for $i = 1, 2, 3$, and then minimize the objective $\sum_{i=1}^{3} \frac{|F_i|}{n} \bar{f}_i(p)$ (which is equivalent to the average BPB over all tasks). In the **aggregated** approach, we fit a single function $\hat{f}_{\mathrm{avg}}(p)$ directly to the average BPB across tasks using $\{(p^j, \frac{1}{n}\sum_{i=1}^{n} y_{ij})\}_{j=1}^{K}$ and minimize $\hat{f}_{\mathrm{avg}}(p)$. We evaluate downstream performance (average BPB of 1B target models) and regression fit (Pearson correlation on held-out mixtures).

Figure 13 shows that the per-task approach yields the best downstream performance and highest regression fit. Regression fit degrades monotonically as granularity decreases, which is consistent with reduced expressivity. Interestingly, while per-family has better regression fit than aggregated, their downstream performance is comparable. We find that this is most likely due to noise in transferring from proxy to target models, since the performance on 30M models (Figure 20) exhibit trends that are consistent with the regression fit. Nevertheless, per-task provides clear benefits on both metrics. **Based on these findings, we recommend fitting separate regression functions per task.**

## B.3. RQ8: Optimization solver

*How do we solve the mixture optimization problem?*

We study how to solve for the mixture that minimizes the average predicted BPB $\hat{f}(p)$. Under a log-linear regression model, $\hat{f}(p)$ is convex, enabling the use of exact solvers. However, since the regression functions imperfectly predict true performance (Figure 16 right), it is unclear whether exact optimization of the surrogate objective yields the mix with the best downstream performance or if alternative approaches that involve some regularization may be better.

We consider three solving strategies. The **exact solver** uses CVXPY to compute the global optimal mix. The **search**-based

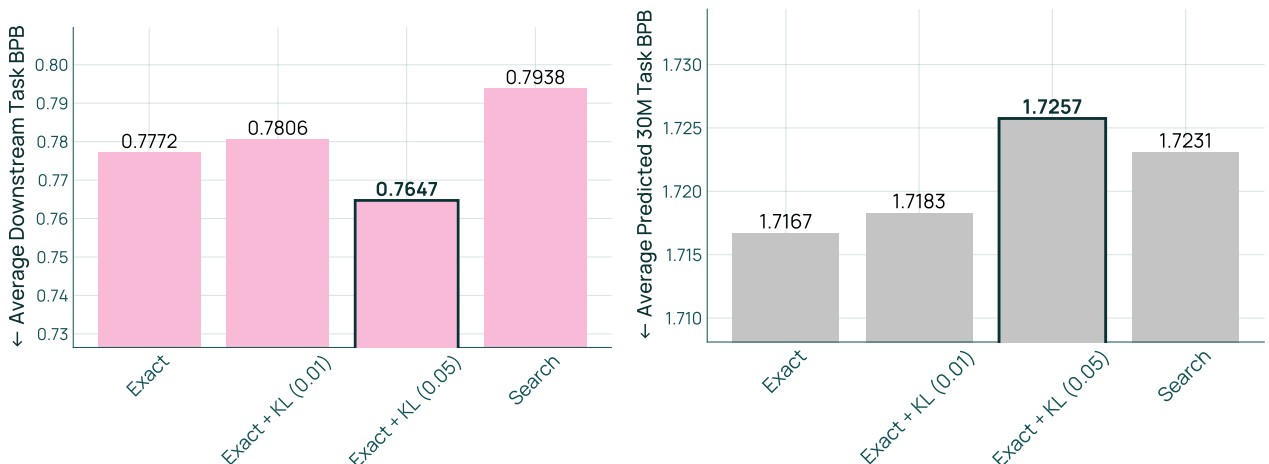

*Figure 14.* Downstream performance (left) and 30M predicted performance (right) across optimization solvers. The predicted performances of various solvers confirms that the exact solver minimizes the predicted BPB, as expected, followed by KL penalties of 0.01 and 0.05. However, the exact solver does not obtain the best downstream performance, and instead some KL regularization helps.

solver, used in RegMix (Liu et al., 2025a), samples candidate mixes from a Dirichlet distribution with a natural prior and performs a local search guided by the regression model. The **exact solver with KL regularization** modifies the objective to be minimize$_{p\in\triangle^{m-1}}\hat{f}(p)+\lambda D_{\mathrm{KL}}(p||p^{\mathrm{natural}})$, where $\lambda>0$ controls the strength of regularization towards the natural distribution, and we consider $\lambda\in\{0.01,0.05\}$. To evaluate these solvers, we measure the average BPB of 1B target models trained on the proposed mixes. In addition, we examine the optimal value of the objective itself, the predicted average BPB at the proxy model scale. This serves as a sanity check that the exact solver should obtain the lowest predicted BPB, followed by KL-regularized solvers with increasing $\lambda$.

Figure 14 (left) shows that using an exact solver with $\lambda=0.05$ achieves the best performance. In contrast, the right panel matches our expectations that the exact solver obtains the lowest predicted BPB, while adding a KL penalty degrades predicted performance. Taken together, these results indicate that although the average predicted BPB is optimized most effectively by the exact solver without regularization, noise arising from regression fitting and from transferring across model scales makes moderate regularization beneficial for downstream performance. We also see that the search baseline achieves poor performance, consistent with the fact that it is a heuristic. **Based on our findings, we recommend using an exact solver with a KL regularization of 0.05 to solve the mixture optimization problem**. See Figure 21 for additional results at the source level.

## C. Offline Schema Study Details

All experiments for the design choice study, unless specified, use the same configuration as OLMIXBASE. For mixing over the DCLM topics, we constructed a sparse swarm of size $K=128$. We also mix at the source level (over DCLM, Stack-Edu, ArXiv, FineMath 3+, olmOCR Science PDFs, Wikipedia, and Pes2o). For the sources, we constructed a dense swarm of size $K=64$. We use log-linear regression per task, an exact solver with $\lambda=0.05$, and enforced repetition constraints in the optimization problem with $R=6T$ and $k=4$.

### C.1. RQ1: Proxy model size details

Table 3 contains details about the architectures of the 1M, 15M, 30M, and 60M proxy models we train.

### C.2. RQ2: Swarm size details

In Figure 15, we vary $K=c(m+1)$ for $c=1,2,3,4$ and $m=6,24$ and study its effect on downstream BPB of 1B models. Despite having fewer $m$, the error curves when scaled according to $c$ collapse together, suggesting that the $\mathcal{O}(m)$ sample complexity holds at both 30M and 1B model scales.

For the results in Figure 4 and Figure 15, we constructed proposed mixes using a search-based solver rather than an exact solver. We also did not enforce repetition constraints and had originally included 5 additional metrics in our evaluation suite:

*Table 3.* Proxy Model Architecture Configurations

| Parameter | 1M | 15M | 30M | 60M |
|---|---|---|---|---|
| Vocab size | 100,352 | 100,352 | 100,352 | 100,352 |
| n_layers | 4 | 8 | 4 | 8 |
| n_heads | 4 | 4 | 8 | 8 |
| d_model | 16 | 128 | 256 | 384 |
| head_dim | 4 | 32 | 32 | 48 |

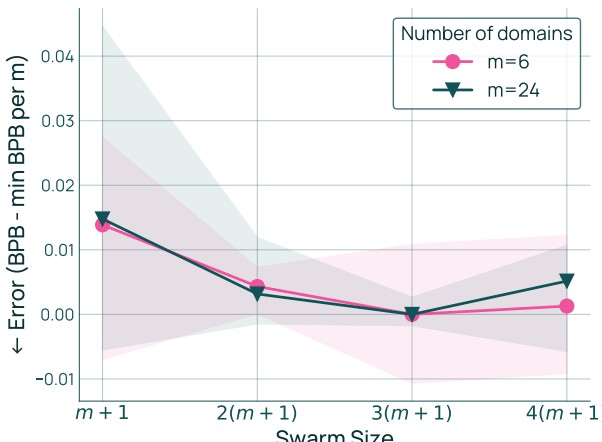

*Figure 15.* Error on 1B models versus swarm size. Similar to Figure 4, we see that the error curves collapse across both $m$, supporting our finding that $K$ needs $\mathcal{O}(m)$ runs to obtain strong downstream performance. Results are averaged across 3 random seeds, and the intervals indicate min and max results.

Qasper Yes/No, Sciriff Yes/No, LabBench DBQA, LabBench ProtocolQA, and MedQA EN.

For $m = 24$, we used all WebOrganizer DCLM topics. For $m < 24$, we selected the $m$ largest domains. In particular, for $m = 18$, we used: Art and Design, Crime and Law, Education and Jobs, Electronics and Hardware, Entertainment, Finance and Business, Games, Health, Literature, Politics, Religion, Science Math and Technology, Social Life, Software, Software Development, Sports and Fitness, Transportation, Travel and Tourism. For $m = 12$, we used: Crime and Law, Education and Jobs, Entertainment, Finance and Business, Games, Health, Literature, Politics, Religion, Science Math and Technology, Software, and Software Development. For $m = 6$, we used: Entertainment, Finance and Business, Games, Health, Politics, and Science Math and Technology.

### C.3. RQ6: Swarm distribution details

To construct the sparse and dense swarms, we first generated mixes sampled from a Dirichlet distribution. For the dense swarm, we discarded any mixes that have domains with 0 weight, and for the sparse swarm, we enforced that if the weight of a domain is less than 0.05, then it is clipped to 0 and the entire mixture is normalized afterwards.

For the centering experiments, we compared a swarm with a natural Dirichlet prior to a strong and weak prior. The strong prior was the proposed mix obtained using the natural swarm. The weak prior was a mix that aimed to maximize BPB using the the natural swarm using a simulation-based solver. Regardless of the choice of Dirichlet prior, we used the natural distribution in the KL regularization term of the objective for fair comparison.

For both sets of experiments, we constructed a held-out test set containing of 30 samples, where 15 were from a Dirichlet distribution centered around the sparse swarm's optimal mix and 15 were from a Dirichlet distribution centered around the dense swarm's optimal mix. This measures regression fit in high-performance regions of the mixture space, which matters in the mixture optimization step.

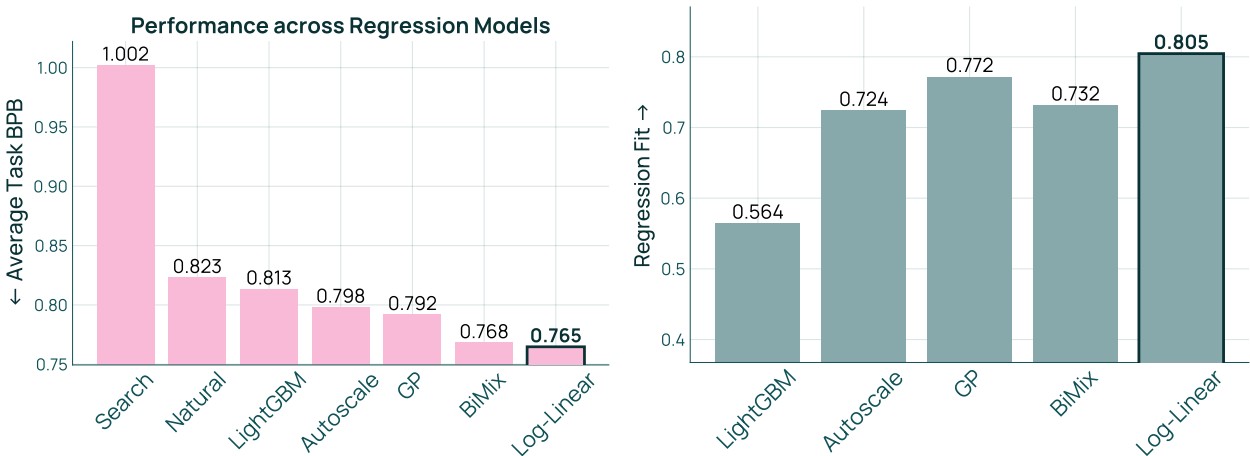

*Figure 16.* Performance (left) and regression fit (right) across regression model families when mixing on DCLM topics. For both metrics, the log-linear regression model outperforms other approaches.

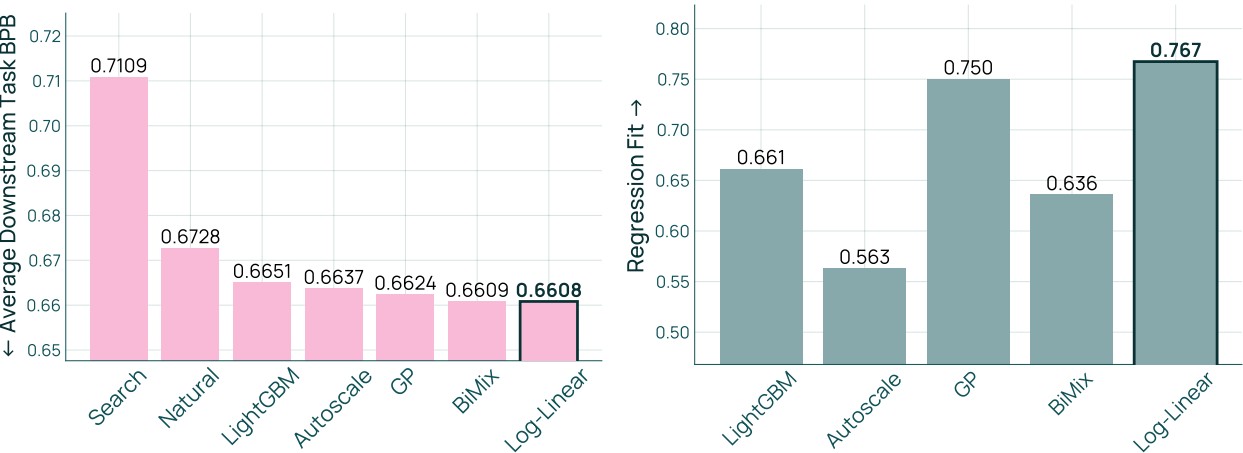

*Figure 17.* Performance (left) and regression fit (right) across regression model families when mixing across sources. For both metrics, the log-linear regression model outperforms other approaches again.

## C.4. RQ3: Regression model family details

Figures 16 and 17 compare the downstream performance and regression fit (measured on three random train-test splits of the swarm, each with 10 test mixes) across several regression model families for two settings: DCLM topics, and at the source level. In both settings and across both metrics, we see that the log-linear regression model outperforms other approaches.

In Figures 18 and 19, we plot the relationship between swarm size, number of domains, and regression fit across all regression model families. These figures reveal that different regression model families have different sample complexity and regimes in which they do well. This offers a potential explanation for why existing methods lack consensus.

We describe how we adapt the regression models from BiMix (Ge et al., 2025b), Autoscale (Kang et al., 2025), and Data Mixing Laws (Ye et al., 2025) to our setting where performance is measured on a set of downstream tasks.

- BiMix: The BiMix mixing law models the validation loss on domain $i$ as a function of the mixture ratio on domain $i$ and the number of training steps $s$:

$$\hat{f}_i(p,s) = \frac{A_i}{p_i^{\alpha_i}} \left( \frac{B_i}{s^{\beta_i}} + C_i \right) \forall i \in [m] \tag{5}$$

Since we do not model the number of steps and instead directly transfer the proposed mix from the proxy scale to the

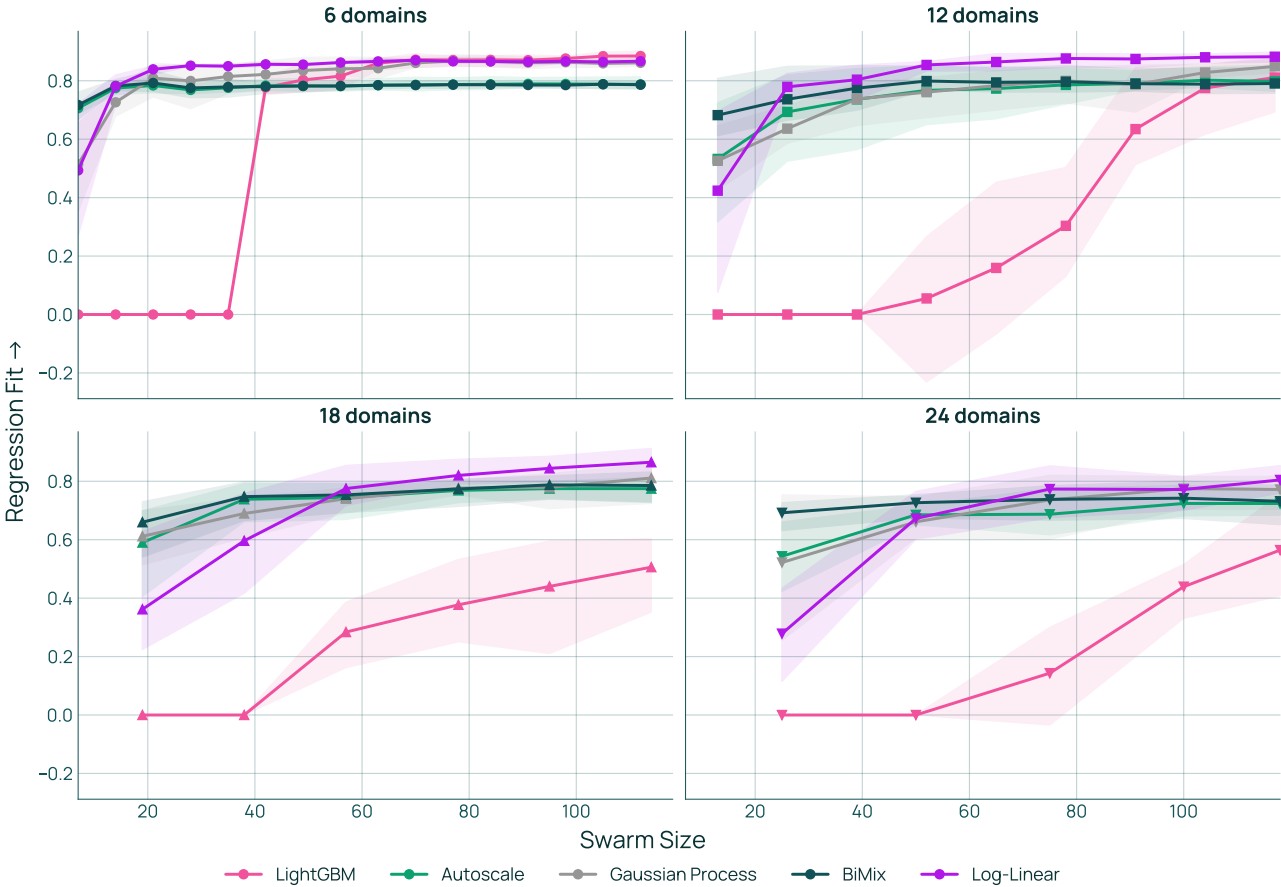

*Figure 18.* Regression fit versus swarm size and number of domains across regression models when mixing on DCLM topics. Across $m = 6, 12, 18, 24$ domains, each regression model requires different swarm sizes to obtain good regression fit. Results are averaged over 3 random seeds.

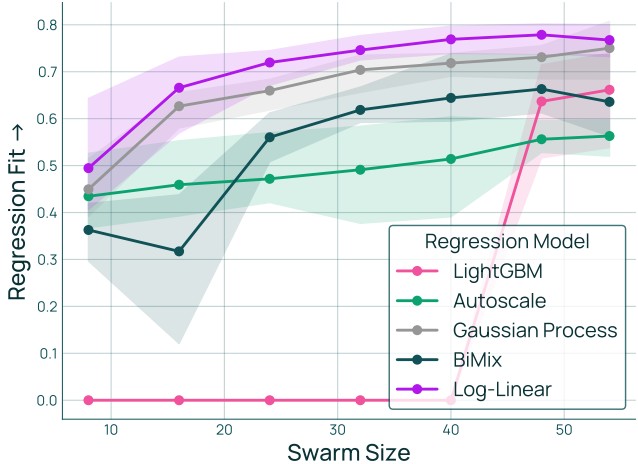

*Figure 19.* Regression fit versus swarm size and number of domains across regression models when mixing across sources. Across $m = 6, 12, 18, 24$ domains, each regression model requires different swarm sizes to obtain good regression fit. Results are averaged over 3 random seeds.

target scale, this mixing law simplifies to $\hat{f}_i(p) = A_i p_i^{-\alpha_i}$, where the constants are absorbed into $A_i$. To extend this mixing law to downstream tasks rather than validation loss on training domains, we assume that task BPB can be modeled as a weighted average of training domain-specific mixing laws. Therefore, for each downstream task $i$, the BiMix mixing law for our setting is:

$$\hat{f}_i(p) = \sum_{j=1}^{m} A_{ij} p_j^{-\alpha_{ij}} \tag{6}$$

- AutoScale: The AutoScale mixing law models validation loss on a held-out dataset $i$ as a function of the mixture and the number of tokens $R$:

$$\hat{f}(p, R) = \sum_{j=1}^{m} \left( (N_0^j + p_j R)^{-\gamma_j} + l_j \right), \tag{7}$$

where $N_0^j$ is the "effective" number of tokens of all other domains excluding $j$. The original paper fits separate parameters for each domain using perturbed runs; we instead fit all parameters jointly using our larger swarm. We make two modifications. First, we replace the per-domain constants $l_j$ with a single constant term $c_i$ corresponding to a given task $i$. Second, instead of directly learning $N_0^j$, which can be quite large given the order of magnitude of $R$, we rewrite $N_0^j + p_j R$ as $R(A_{ij} + p_j)$ and learn $A_{ij}$. Therefore, for each downstream task $i$, the AutoScale mixing law for our setting is:

$$\hat{f}_i(p) = c_i + \sum_{j=1}^{m} (R(A_{ij} + p_j))^{-\alpha_{ij}} \tag{8}$$

- Data Mixing Laws: The mixing law models the validation loss on domain $i$ as:

$$\hat{f}_i(p) = c_i + k_i \exp\left( \sum_{j=1}^{m} t_{ij} p_j \right) \tag{9}$$

The paper also extends this to out-of-domain validation sets by expressing them as a linear combination of training domains, namely $\hat{f}(p) = \sum_{i=1}^{m} s_i \left( c_i + k_i \exp\left( \sum_{j=1}^{m} t_{ij} p_j \right) \right)$. We use the first, simpler form (9) directly for downstream tasks, treating each task's performance as a function of the mixture on training domains. Moreover, we observe that $k_i$ can be absorbed into the linear term; rewriting (9) as $c_i + \exp(\sum_{j=1}^{m} t_{ij} p_j + \log k_i)$ shows that $k_i$ acts as an intercept term. Since $p$ sums up to 1, this intercept is redundant and can be dropped. Therefore, our final log-linear mixing law is

$$\hat{f}_i(p) = c_i + \exp\left( \sum_{j=1}^{m} A_{ij} p_j \right) \tag{10}$$

Next, we provide implementation details for each model:

- Search: we select the swarm run that has the best average BPB and satisfies the data repetition constraints.

- LightGBM: we use the hyperparameters from RegMix (Liu et al., 2025a). However, we did not use an evaluation set for early stopping, since we noticed that RegMix used the same set of mixes for early stopping and for reporting performance. We solve the optimization problem using search, as is done in RegMix.

- Gaussian Process: we use scikit-learn's GaussianProcessRegressor. We solve the optimization problem using search, as is done in Chen et al. (2025b).

- BiMix: we use SciPy least squares to fit the regression models. We solve the optimization problem using search, an exact solver, and exact solver with KL regularization ($\lambda = 0.05$) in Table 4. Based on the results, we decided to use a search-based solver when evaluating BiMix.

*Table 4.* Downstream performance of BiMix and AutoScale regression models on DCLM topics using exact optimization, exact + KL regularization ($\lambda = 0.05$), and search. Based on these findings, we use an exact solver with Autoscale and a search-based solver for BiMix for the rest of our study.

| Method | Exact (0.0) | Exact + KL (0.05) | Search |
|---|---|---|---|
| BiMix | 0.776733 | 0.790188 | **0.768153** |
| Autoscale | **0.798084** | 0.804814 | 0.808775 |

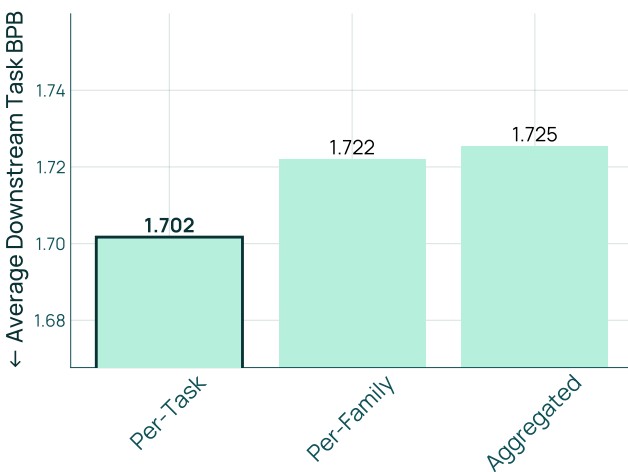

*Figure 20.* 30M performance across regression granularities. The performance is monotonic in the task granularity, matching with the regression fit findings from Figure 13.

- AutoScale: we use SciPy least squares to fit the regression models. We solve the optimization problem using search, an exact solver, and exact solver with KL regularization ($\lambda = 0.05$)in Table 4. Based on the results, we decide to use an exact solver when evaluating Autoscale.

- Log-linear: we use the fitting code from Ye et al. (2025). We solve the optimization problem using CVXPY, while results for other solvers are discussed in Appendix B.3.

### C.5. RQ7: Regression granularity details

Figure 20 shows the performance of various regression granularities at the 30M scale. While the downstream BPB results at the 1B scale (Figure 13) demonstrates that per-family performs the worst, the 30M performance results are in line with the regression fits of each granularity; per-task performs the best, followed by per-family and aggregated.

The task families we used are defined in Table 8: Math, Code, and QA. We constructed a held-out test set containing of 45 samples, consisting of 15 samples from a Dirichlet distribution centered around each regression granularity's respective proposed mix. This measures regression fit in high-performance regions of the mixture space, which matters in the mixture optimization step.

### C.6. RQ4: Data repetition constraints details

We study several ways of enforcing repetition constraints. 1) We use a constrained swarm and unconstrained optimization: we construct a swarm where all of its mixes satisfy the repetition constraint. Mixture optimization solves the unconstrained problem minimize$_{p \in \triangle^{m-1}} \hat{f}(p)$, relying on the regression models to encode what mixes are feasible. 2) We consider an unconstrained swarm and constrained optimization: we sample swarm mixes from $\mathcal{P}$ without restrictions, and we add the repetition constraints to the mixture optimization problem. 3) We consider both a constrained swarm and constrained optimization. We test with $k = 4$, and for each approach we verify whether the proposed mixture satisfies the repetition constraint.

Table 5 shows that approach 2, using an unconstrained swarm with constrained optimization results in a proposed mix that satisfies the repetition constraints while maintaining performance. Approach 1) fails to satisfy the constraints, resulting in

*Table 5.* Performance and repetition constraint satisfaction. Using a constrained swarm with unconstrained optimization does not satisfy a repetition constraint with $k = 4$; the proposed mix repeats samples of a domain 5 times. Between the unconstrained and constrained swarm with constrained optimization (approaches 2 and 3), the latter approach achieves the best downstream performance.

| Approach | Satisfies rep constraint? ($k = 4$) | Average task BPB $\downarrow$ |
|---|---|---|
| 1) Constrained Swarm, Unconstrained Opt | No (5) | 0.774694 |
| 2) Unconstrained Swarm, Constrained Opt | Yes (4) | 0.764718 |
| 3) Constrained Swarm, Constrained Opt | Yes (4) | 0.785517 |

*Figure 21.* Performance (left) and regression fit (right) across optimization solvers when mixing across sources. The predicted performances of various solvers confirms that the exact solver minimizes the predicted BPB, as expected, followed by KL penalties of 0.01 and 0.05. However, the exact solver does not obtain the best downstream performance, and instead some KL regularization helps.

samples from one of the domains being repeated 5 times, which is more than $k = 4$. Approach 3) satisfies constraints, but yields worse downstream performance than 2), likely because the constrained swarm provides less coverage of the mixture space.

To construct the constrained swarm, we used $K = 128$ proxy runs. We set $R = 6$T requested tokens and a repetition factor of $k = 4$. The constrained swarm is constructed via rejection sampling from the Dirichlet prior. For constrained optimization, we specified the repetition constraint in CVXPY.

### C.7. RQ8: Optimization solver details

Figure 21 shows the performance and predicted performance across optimization solvers when mixing at the source level. Similar to Figure 14, we see that Exact + KL (0.05) outperforms the other solvers in terms of downstream BPB. We also sanity check our optimizers by examining predicted performance, confirming that the exact optimizer obtains the lowest predicted performance while adding a KL term degrades performance.

For the search approach, we used the adaptive search proposed in Wettig et al. (2025), which proceeds in multiple rounds such that the best mix from the current round is used as a Dirichlet prior to generate candidate mixes for the next round.

## D. Algorithm Details

#### D.0.1. SOLVING THE CONDITIONAL MIXING PROBLEM

We explain how to solve (3), by using a change of variables that transforms it into a standard mixing problem (without the explicit $q_{\mathcal{D}_{\text{fix}}} = \tilde{p}_{\mathcal{D}_{\text{fix}}}$ constraint) over a *collapsed* space. After this change of variables, the constraint is absorbed into the variable and standard mixing methods like OLMIXBASE can be applied in the collapsed space.

For any feasible $q = [\rho \tilde{p}_{\mathcal{D}_{\text{fix}}}, (1 - \rho) q_{\mathcal{D}_{\text{comp}}}]$, let the corresponding mix in collapsed space be $r = [\rho, (1 - \rho) q_{\mathcal{D}_{\text{comp}}}] \in \triangle^{|\mathcal{D}_{\text{comp}}|}$. $r$ is defined over a virtual domain, containing all of $\mathcal{D}_{\text{fix}}$, and the domains in $\mathcal{D}_{\text{comp}}$. Formally, we index the elements of $r$ by $\{v\} \cup \mathcal{D}_{\text{comp}}$ and define mapping functions that convert between $r$ and $q$.

- Collapse $\Psi(q)=r$: sum all weights on $\mathcal{D}_{\text{fix}}$ into the virtual domain weight $r_v=\sum_{j\in\mathcal{D}_{\text{fix}}}q_j$, and keep weights on $\mathcal{D}_{\text{comp}}$ the same, $r_j=q_j$ for $j\in\mathcal{D}_{\text{comp}}$.

- Expand $\Phi_{\tilde{p}_{\mathcal{D}_{\text{fix}}}}(r)=q$: multiply the virtual domain weight by $\tilde{p}_{\mathcal{D}_{\text{fix}}}$, i.e., $q_j=r_v\cdot\tilde{p}_j$ for $j\in\mathcal{D}_{\text{fix}}$, and keep weights on $\mathcal{D}_{\text{comp}}$ the same, $q_j=r_j$ for $j\in\mathcal{D}_{\text{comp}}$.

Then, we can write the mixture reuse problem (3) in terms of $r$:

$$\text{minimize}_{r\in\triangle^{|\mathcal{D}_{\text{comp}}|}}\frac{1}{n}\sum_{i=1}^{n}g_i(r) \tag{11}$$

$$\text{s.t.}\quad r_v\le\min_{j\in\mathcal{D}_{\text{fix}}}\left\{\frac{kN'_j}{R\tilde{p}_j}\right\}\quad r_j\le\frac{kN'_j}{R}\quad\forall j\in\mathcal{D}_{\text{comp}}$$

where $g_i(r):=f_i(\text{LM}(S,R,\Phi_{\tilde{p}_{\mathcal{D}_{\text{fix}}}}(r)))$ is the performance of the mix $\Phi_{\tilde{p}_{\mathcal{D}_{\text{fix}}}}(r)$ on task $i$, and the repetition constraints have been rewritten in terms of $r$. This optimization problem looks similar to (2)—determine the mix $r$ that minimizes the average BPB, subject to some per-domain constraints. Therefore, we can apply OLMIXBASE to obtain our FULLMIXTUREREUSE method, detailed in Algorithm 2. The blue coloring indicates aspects of this algorithm that are different from OLMIXBASE, showing that the key difference is simply two expansion steps, and otherwise handling everything in collapsed space over $r$.

---

**Algorithm 2** FULLMIXTUREREUSE

1: **Input:** Domain set $\mathcal{D}'=\mathcal{D}_{\text{fix}}\cup\mathcal{D}_{\text{comp}}$, existing mix $\tilde{p}\in\triangle^{m-1}$, swarm size $K=\mathcal{O}(m)$, repetition factor $k$, requested tokens $R$.

2: Sample mixes $r^1,...,r^K\in\triangle^{|\mathcal{D}_{\text{comp}}|}$ and expand each collapsed mix $r^j$ into $q^j:=\Phi_{\tilde{p}_{\mathcal{D}_{\text{fix}}}}(r^j)$.

3: Train proxy models on the expanded mixes and evaluate on downstream tasks to get a dataset of mixes and performance, $\{(r^j,\{y_{ij}\}_{i=1}^n)\}_{j=1}^K$, where $y_{ij}:=f_i(\text{LM}(S_{\text{small}},R_{\text{small}},q^j))$.

4: **for** $i\in[n]$ **do**

5:     Use $\{(r^j,y_{ij})\}_{j=1}^K$ to fit the log-linear model $\hat{g}_i(r)=d_i+\exp(B_i^\top r)$, where $d_i\in\mathbb{R}^+$ and $B_i\in\mathbb{R}^{|\mathcal{D}_{\text{comp}}|}$.

6: **end for**

7: Solve the following optimization problem to get $r^\star(\tilde{p}_{\mathcal{D}_{\text{fix}}})$:

$$\text{minimize}_{r\in\triangle^{|\mathcal{D}_{\text{comp}}|}}\frac{1}{n}\sum_{i=1}^{n}\hat{g}_i(r) \tag{12}$$

$$\text{s.t.}\quad r_v\le\min_{j\in\mathcal{D}_{\text{fix}}}\left\{\frac{kN'_j}{R\tilde{p}_j}\right\},r_j\le\frac{kN'_j}{R}\quad\forall j\in\mathcal{D}_{\text{comp}}$$

8: **Return** $q^\star(\tilde{p}_{\mathcal{D}_{\text{fix}}}):=\Phi_{\tilde{p}_{\mathcal{D}_{\text{fix}}}}(r^\star(\tilde{p}_{\mathcal{D}_{\text{fix}}}))$, the expanded form of $r^\star(\tilde{p}_{\mathcal{D}_{\text{fix}}})$.

---

# E. Proofs for Section 4.3

We first provide definitions, assumptions, and notations.

**Definitions.** Define performance of $q$ as $F(q):=\frac{1}{n}\sum_{i=1}^n f_i(\text{LM}(S,R,q))$. We also write $F(q)$ as a bivariate version, $F(r,p_1):=F(\Phi_{p_1}(r))$. Furthermore, define $\bar{F}(\cdot)$ to be the objective without the constant term of the log-linear regression model, $\bar{F}(\cdot)=F(\cdot)-\frac{1}{n}\sum_{i=1}^n c_i$.

Recall $q^\star$ is the optimal solution to (2) on $\mathcal{D}'$, and $q^\star(\tilde{p}_{\mathcal{D}_{\text{fix}}})$ is solution returned by FULLMIXTUREREUSE when reusing $\tilde{p}_{\mathcal{D}_{\text{fix}}}$ (3). We define $q^\star_{\mathcal{D}_{\text{fix}}}$ as the optimal mix on $\mathcal{D}_{\text{fix}}$ after the domain set update; that is, $q^\star=[\rho^\star q^\star_{\mathcal{D}_{\text{fix}}},(1-\rho^\star)q^\star_{\mathcal{D}_{\text{comp}}}]$. We can also write $q^\star(\tilde{p}_{\mathcal{D}_{\text{fix}}})=[\rho^\star(\tilde{p}_{\mathcal{D}_{\text{fix}}}\tilde{p}_{\mathcal{D}_{\text{fix}}},(1-\rho^\star(\tilde{p}_{\mathcal{D}_{\text{fix}}}))q^\star_{\mathcal{D}_{\text{comp}}}(\tilde{p}_{\mathcal{D}_{\text{fix}}})]$.

We can also write $\tilde{p}$ in terms of $\mathcal{D}_{\text{fix}}$ and $\mathcal{D}\setminus\mathcal{D}_{\text{fix}}$: any $p$ can be expressed as $p=[\pi p_{\mathcal{D}_{\text{fix}}},(1-\pi)p_{\mathcal{D}\setminus\mathcal{D}_{\text{fix}}}]$, where $\pi\in[0,1]$ (similar to $\rho$).

The log-linear model is $f_i(q)=c_i+\sum_{i=1}^n\exp(A_i^\top q)$, where $A_{ij}$ intuitively captures a notion of how much domain $j$ impacts task $i$. We split each $A_i$ into $[A_{i,\text{fix}},A_{i,\text{comp}}]$ corresponding to $\mathcal{D}_{\text{fix}}$ and $\mathcal{D}_{\text{comp}}$. Define $\alpha_{\text{fix}},\alpha_{\text{comp}}\in\mathbb{R}^n$ where

$\alpha_{i,\text{fix}} = \|A_{i,\text{fix}}\|, \alpha_{i,\text{comp}} = \|A_{i,\text{comp}}\|.$

Finally, we define the coupling term using the log-linear coefficients over $\mathcal{D}_{\text{fix}}$ and $\mathcal{D}_{\text{comp}}$.

**Definition E.1.** The coupling term $\kappa(\mathcal{D}_{\text{fix}}, \mathcal{D}_{\text{comp}}) = \kappa(\alpha_{\text{fix}}, \alpha_{\text{comp}})$ is defined as as

$$\kappa(\alpha_{\text{fix}}, \alpha_{\text{comp}}) := \|(1 + \alpha_{\text{fix}} + \alpha_{\text{comp}}) \odot \alpha_{\text{fix}}\|, \tag{13}$$

where $\odot$ is the Hadamard product.

**Notation.** Denote $p_1 = \tilde{p}_{\mathcal{D}_{\text{fix}}}$ and $p_1' = q_{\mathcal{D}_{\text{fix}}}^\star$ for simplicity. Let $p_2 = q_{\mathcal{D}_{\text{comp}}}$. Recall that $r^\star(p_1)$ is the solution to (3) when $q_{\mathcal{D}_{\text{fix}}}$ is constrained to be $p_1$. Any $r$ can be written as $r = [\rho, (1-\rho), p_2]$, so we similarly define $\rho^\star(p_1)$ and $p_2^\star(p_1)$.

### E.1. Proof for Theorem 4.1

**Assumption E.2.** We make the following assumptions:

1. The log-linear model accurately captures the relationship between the mixture ratios and target model performance on each task; that is, $f_i(\text{LM}(S, R, q)) = c_i + \exp(A_i^\top q)$ for some $c_i \in \mathbb{R}^+, A_i \in \mathbb{R}^{m'}$.

2. For simplicity, applying OLMIXBASE (Algorithm 1) exactly solves (2). That is, we ignore (i) proxy-to-target transfer error and (ii) estimation error in learning $c_i$ and $A_i$ from the swarm. However, extending the analysis to include these errors is straightforward by adding corresponding approximation terms.

3. We assume that $F(r, p_1)$ is $\mu$-strongly convex in $r$.

4. Let $\mathcal{S}(q_{\mathcal{D}_{\text{fix}}})$ be the feasible set of (11) defined by repetition constraints:

$$r_v \leq \min_{j \in \mathcal{D}_{\text{fix}}} \left\{ \frac{kN_j'}{Rq_j} \right\}, r_j \leq \frac{kN_j'}{R} \, \forall j \in \mathcal{D}_2',$$

together with the simplex constraints on $r$. We assume **mutual feasibility** of $\tilde{p}_{\mathcal{D}_{\text{fix}}}$ and $q_{\mathcal{D}_{\text{fix}}}^\star$: that $r^\star(\tilde{p}_{\mathcal{D}_{\text{fix}}}) \in \mathcal{S}(q_{\mathcal{D}_{\text{fix}}}^\star)$ and $r^\star(q_{\mathcal{D}_{\text{fix}}}^\star) \in \mathcal{S}(\tilde{p}_{\mathcal{D}_{\text{fix}}})$. This holds in the following two cases:

   - Both $\tilde{p}_{\mathcal{D}_{\text{fix}}}$ and $q_{\mathcal{D}_{\text{fix}}}^\star$ have an active repetition constraint on at least one domain. Intuitively, this means that among the unaffected domains, there exist some high-utility domains that are data-constrained.

   - $r_v^\star(\tilde{p}_{\mathcal{D}_{\text{fix}}}) \leq \min_{j \in \mathcal{D}_{\text{fix}}}\{\frac{kN_j'}{Rq_j}\}$ and $r_v^\star(\tilde{p}_{\mathcal{D}_{\text{fix}}}) \leq \min_{j \in \mathcal{D}_{\text{fix}}}\{\frac{kN_j'}{R\tilde{p}_j}\}$. Intuitively, the contents of the virtual domain are sufficiently low utility such that the proposed mix does not "push against" the tighter of the two $r_v$ constraints.

   This assumption is reasonable in our setting: when the repetition constraint is active, a domain is valuable enough that we would allocate more weight if we could. When it is slack, the domain does not warrant allocating weight up to its availability limit. Our two cases simply rule out the marginal situation where one mix is availability-limited while the other is not.

First, we show that the mixture reuse problem and the standard mixing problem are equivalent when the existing mix on the unaffected domains is equal to the optimal mix on the unaffected domains after the domain update. This is the "equality" condition of Theorem 4.1: when $\tilde{p}_{\mathcal{D}_{\text{fix}}} = q_{\mathcal{D}_{\text{fix}}}^\star$, the performance gap is 0.

**Lemma E.3.** *The solution of (3), $r^\star(\tilde{p}_{\mathcal{D}_{\text{fix}}})$, is equivalent to the solution of (2), $q^\star = [\rho^\star q_{\mathcal{D}_{\text{fix}}}^\star, (1-\rho^\star)q_{\mathcal{D}_2'}^\star]$, when the existing mix satisfies $\tilde{p}_{\mathcal{D}_{\text{fix}}} = q_{\mathcal{D}_{\text{fix}}}^\star$. In other words, $\Phi_{q_{\mathcal{D}_{\text{fix}}}^\star}(r^\star(q_{\mathcal{D}_{\text{fix}}}^\star)) = q^\star$, and $q^\star(q_{\mathcal{D}_{\text{fix}}}^\star) = q^\star$.*

*Proof.* For (2), restricting the feasible set to the subspace $q_{\mathcal{D}_{\text{fix}}} = q_{\mathcal{D}_{\text{fix}}}^\star$ does not change the optimal value of this problem, since this subspace still contains the optimal solution to the original problem. Therefore, (2) is equivalent to:

$$\text{minimize}_{q \in \triangle^{m'-1}} \frac{1}{n} \sum_{i=1}^m f_i(\text{LM}(S, R, q))$$

$$\text{s.t.} \quad q_j \leq \frac{kN_j'}{R} \quad \forall j \in [m']$$

$$q_{\mathcal{D}_{\text{fix}}} = q_{\mathcal{D}_{\text{fix}}}^\star$$

Comparing this formulation of (2) to (3), we can see that the problems are equivalent when $\tilde{p}_{\mathcal{D}_{\text{fix}}} = q^{\star}_{\mathcal{D}_{\text{fix}}}$. □

Based on this result, we can write $q^{\star} = q^{\star}(q^{\star}_{\mathcal{D}_{\text{fix}}}) = q^{\star}(p'_1)$. Similarly, $\rho^{\star} = \rho^{\star}(p'_1)$, $r^{\star} = r^{\star}(p'_1)$, $p^{\star}_2 = p^{\star}_2(p'_1)$. Furthermore, the problem of bounding the performance gap now becomes one of bounding $F(q^{\star}(\tilde{p}_{\mathcal{D}_{\text{fix}}})) - F(q^{\star}(q^{\star}_{\mathcal{D}_{\text{fix}}}))$ in terms of $\|q^{\star}(\tilde{p}_{\mathcal{D}_{\text{fix}}}) - q^{\star}_{Dfix}\|$. We can apply standard tools from optimization and convexity over $F(r, p_1)$ to address this.

### E.1.1. PERFORMANCE GAP LEMMAS

**Lemma E.4** (FOC Inequality). *Recall that $r^{\star}(p_1) = \operatorname{argmin}_{r \in \mathcal{S}(p_1)} F(r, p_1)$ and $r^{\star}(p'_1) = \operatorname{argmin}_{r \in \mathcal{S}(p'_1)} F(r, p'_1)$. Then,*

$$\langle \nabla_r F(r^{\star}(p_1), p_1) - \nabla_r F(r^{\star}(p'_1), p'_1), r^{\star}(p'_1) - r^{\star}(p_1) \rangle \geq 0.$$

*Proof.* Since $r^{\star}(p_1)$ minimizes $F(\cdot, p_1)$ over the convex set $\mathcal{S}(p_1)$, we have the first-order condition

$$\langle \nabla_r F(r^{\star}(p_1), p_1), r - r^{\star}(p_1) \rangle \geq 0 \quad \forall r \in \mathcal{S}(p_1).$$

Using the mutual feasibility assumption, $r^{\star}(p'_1) \in \mathcal{S}(p_1)$, so plugging $r = r^{\star}(p'_1)$ yields

$$\langle \nabla_r F(r^{\star}(p_1), p_1), r^{\star}(p'_1) - r^{\star}(p_1) \rangle \geq 0.$$

Similarly, since $r^{\star}(p'_1)$ minimizes $F(\cdot, p'_1)$ over $\mathcal{S}(p'_1)$ and $r^{\star}(p_1) \in \mathcal{S}(p'_1)$,

$$\langle \nabla_r F(r^{\star}(p'_1), p'_1), r^{\star}(p_1) - r^{\star}(p'_1) \rangle \geq 0.$$

Adding the two inequalities completes the proof. □

**Lemma E.5** (Mean value bound for $\nabla_r F$ along a segment). *Fix $r \in \mathbb{R}^{|\mathcal{D}'_2|+1}$ and consider the map*

$$G(\cdot) := \nabla_r F(r, \cdot).$$

*Assume $G$ is continuously differentiable on the line segment*

$$p_1^t := t p'_1 + (1-t) p_1, \qquad t \in [0,1].$$

*Then*

$$\|\nabla_r F(r, p'_1) - \nabla_r F(r, p_1)\| \leq \sup_{t \in [0,1]} \|\nabla^2_{r, p_1} F(r, p_1^t)\| \cdot \|p'_1 - p_1\|.$$

*Proof.* Define the scalar function

$$h(t) := G(p_1^t) \in \mathbb{R}^{|\mathcal{D}'_2|+1}, \qquad t \in [0,1].$$

Since $G$ is continuously differentiable on the segment, $h$ is differentiable and

$$h'(t) = \nabla_{p_1} G(p_1^t)(p'_1 - p_1),$$

where $\nabla_{p_1} G(p_1^t)$ denotes the Jacobian of $G$ at $p_1^t$. By the fundamental theorem of calculus,

$$G(p'_1) - G(p_1) = h(1) - h(0) = \int_0^1 h'(t) dt = \int_0^1 \nabla_{p_1} G(p_1^t)(p'_1 - p_1) dt.$$

Taking norms and applying the triangle inequality,

$$\|G(p'_1) - G(p_1)\| \leq \int_0^1 \|\nabla_{p_1} G(p_1^t)(p'_1 - p_1)\| dt$$

$$\leq \int_0^1 \|\nabla_{p_1} G(p_1^t)\| \|p'_1 - p_1\| dt$$

$$\leq \left( \sup_{t \in [0,1]} \|\nabla_{p_1} G(p_1^t)\| \right) \cdot \|p'_1 - p_1\|.$$

Substituting $G(\cdot) = \nabla_r F(r, \cdot)$ yields the equivalent mixed-derivative form. □

**Lemma E.6** (Behavior of $r^\star(\cdot)$ under changes in $p_1$). *Let $\Delta := p_1' - p_1$ and $\Delta r := r^\star(p_1') - r^\star(p_1)$.*

*Then*

$$\|\Delta r\| \le \frac{M}{\mu}\|\Delta\|,$$

*where*

$$M := \sup_{t \in [0,1]} \big\|\nabla^2_{r,p_1} F\big(r^\star(p_1'), p_1^t\big)\big\|, \quad p_1^t := tp_1' + (1-t)p_1.$$

*Proof.* By $\mu$-strong convexity of $F(\cdot, p_1)$ in $r$, for $r = r^\star(p_1)$ and $r' = r^\star(p_1')$,

$$F(r', p_1) \ge F(r, p_1) + \langle \nabla_r F(r, p_1), r' - r \rangle + \frac{\mu}{2}\|r' - r\|^2, \quad F(r, p_1) \ge F(r', p_1) + \langle \nabla_r F(r', p_1), r - r' \rangle + \frac{\mu}{2}\|r - r'\|^2.$$

Adding yields

$$\mu\|\Delta r\|^2 \le \langle \nabla_r F(r^\star(p_1'), p_1) - \nabla_r F(r^\star(p_1), p_1), \Delta r \rangle.$$

Next, we add and subtract $\nabla_r F(r^\star(p_1'), p_1')$:

$$\mu\|\Delta r\|^2 \le \langle \nabla_r F(r^\star(p_1'), p_1) - \nabla_r F(r^\star(p_1'), p_1'), \Delta r \rangle$$
$$+ \langle \nabla_r F(r^\star(p_1'), p_1') - \nabla_r F(r^\star(p_1), p_1), \Delta r \rangle.$$

By Lemma E.4, the second inner product is less than 0. Thus

$$\mu\|\Delta r\|^2 \le \|\nabla_r F(r^\star(p_1'), p_1) - \nabla_r F(r^\star(p_1'), p_1')\| \cdot \|\Delta r\|.$$

Finally, applying Lemma E.5, we have

$$\mu\|\Delta r\|^2 \le \sup_{t \in [0,1]} \big\|\nabla^2_{r,p_1} F(r^\star(p_1'), p_1^t)\big\| \cdot \|\Delta\| \cdot \|\Delta r\|$$

We cancel $\|\Delta r\|$ to complete the proof. $\qquad\square$

**Lemma E.7** (Decomposing the performance gap). *Define $\Delta := p_1' - p_1$, $\Delta r := r^\star(p_1') - r^\star(p_1)$. Assume $F(\cdot, \cdot)$ is convex in $r$ and differentiable in both arguments. Then*

$$F(r^\star(p_1), p_1) - F(r^\star(p_1'), p_1') \le \|\nabla_r F(r^\star(p_1), p_1)\| \cdot \|\Delta r\| + \|\Delta\| \cdot \sup_{t \in [0,1]} \|\nabla_{p_1} F(r^\star(p_1'), p_1^t)\|.$$

*Proof.* Add and subtract $F(r^\star(p_1'), p_1)$:

$$F(r^\star(p_1), p_1) - F(r^\star(p_1'), p_1') = \big(F(r^\star(p_1), p_1) - F(r^\star(p_1'), p_1)\big) + \big(F(r^\star(p_1'), p_1) - F(r^\star(p_1'), p_1')\big).$$

For the first difference, convexity of $F(\cdot, p_1)$ implies

$$F(r^\star(p_1'), p_1) \ge F(r^\star(p_1), p_1) + \langle \nabla_r F(r^\star(p_1), p_1), r^\star(p_1') - r^\star(p_1) \rangle,$$

so

$$F(r^\star(p_1), p_1) - F(r^\star(p_1'), p_1) \le \langle \nabla_r F(r^\star(p_1), p_1), r^\star(p_1) - r^\star(p_1') \rangle \le \|\nabla_r F(r^\star(p_1), p_1)\| \cdot \|\Delta r\|.$$

For the second difference, we apply the mean value theorem along $p_1^t$:

$$|F(r^\star(p_1'), p_1) - F(r^\star(p_1'), p_1')| \le \|\Delta\| \cdot \sup_{t \in [0,1]} \|\nabla_{p_1} F(r^\star(p_1'), p_1^t)\|.$$

Combining the two bounds completes our proof. $\qquad\square$

### E.1.2. GRADIENT LEMMAS

**Lemma E.8.** *The gradient norm* $\|\nabla_r F(r^\star(p_1), p_1)\|$ *can be bounded by*

$$\nabla_r F(r^\star(p_1), p_1) \le \bar{F}(q^\star(p_1)) \big\| |A_1 p_1| + \alpha_{comp} \big\|$$

*Proof.* First, we write $r$ as $[\rho, b_2]$, where $b_2 = (1-\rho)p_2$. Define $s_i = A_{i,1}^\top \rho p_1 + A_{i,2}^\top b_2$. We compute the gradient of $F(r, p_1)$ with respect to $\rho$ and $b_2$:

$$\nabla_\rho F(r, p_1) = \frac{1}{n} \sum_{i=1}^n \exp(s_i) A_{i,1}^\top p_1$$

$$\nabla_{b_2} F(r, p_1) = \frac{1}{n} \sum_{i=1}^n \exp(s_i) A_{i,2}$$

We can consolidate this into a single expression if we define $a_i = \begin{bmatrix} \langle A_{i,1}, p_1 \rangle \\ A_{i,2} \end{bmatrix} \in \mathbb{R}^{|\mathcal{D}_2'|+1}$:

$$\nabla_r F(r, p_1) = \frac{1}{n} \sum_{i=1}^n \exp(s_i) a_i$$

Let us write $r^\star(p_1)$ as $[\rho^\star(p_1), (1-\rho^\star(p_1))p_2^\star(p_1)]$. The expression for the gradient at $r^\star(p_1)$ is

$$\nabla_r F(r^\star(p_1), p_1) = \frac{1}{n} \sum_{i=1}^n \exp(A_{i,1}^\top \rho^\star(p_1)p_1 + A_{i,2}^\top (1-\rho^\star(p_1))p_2^\star(p_1)) a_i$$

We now bound the norm. First, applying the Cauchy-Schwarz inequality twice, we have:

$$\|\nabla_r F(r^\star(p_1), p_1)\| \le \frac{1}{n} \sum_{i=1}^n \exp(A_{i,1}^\top \rho^\star(p_1)p_1 + A_{i,2}^\top (1-\rho^\star(p_1))p_2^\star(p_1)) \|a_i\|$$

$$\le \frac{1}{n} \sqrt{\sum_{i=1}^n \exp(2(A_{i,1}^\top \rho^\star(p_1)p_1 + A_{i,2}^\top (1-\rho^\star(p_1))p_2^\star(p_1)))} \sqrt{\sum_{i=1}^n \|a_i\|^2}$$

$$\le \frac{1}{n} \sum_{i=1}^n \exp(A_{i,1}^\top \rho^\star(p_1)p_1 + A_{i,2}^\top (1-\rho^\star(p_1))p_2^\star(p_1)) \sqrt{\sum_{i=1}^n \|a_i\|^2}$$

Observe that the first summation is equivalent to $\bar{F}(q^\star(p_1))$. Then, to bound the remaining term $\sqrt{\sum_{i=1}^n \|a_i\|^2}$, we have

$$\sqrt{\sum_{i=1}^n \|a_i\|^2} = \sqrt{\sum_{i=1}^n \langle A_{i,1}, p_1 \rangle^2 + \|A_{i,2}\|^2}$$

$$\le \sqrt{\sum_{i=1}^n (|\langle A_{i,1}, p_1 \rangle| + \|A_{i,2}\|)^2}$$

$$= \big\| |A_1 p_1| + \alpha_{\text{comp}} \big\|$$

where $|\cdot|$ is an elementwise absolute value, and $\alpha_{\text{comp}} \in \mathbb{R}^n$ is constructed where $\alpha_{i,2} = \|A_{i,2}\|$.

Therefore,

$$\nabla_r F(r^\star(p_1), p_1) \le \bar{F}(q^\star(p_1)) \big\| |A_1 p_1| + \alpha_{\text{comp}} \big\|$$

$\square$

**Lemma E.9.** *The gradient norm $\nabla_{p_1} F(r^\star(p_1'), p_1^t)$ can be bounded by*

$$\|\nabla_{p_1} F(r^\star(p_1'), p_1^t)\| \leq \bar{F}(q^\star(p_1')) \|\exp(\alpha_{fix} \|\triangle\|) \odot \alpha_{fix}\|$$

*where $\odot$ is the Hadamard product, and $\exp$ is elementwise.*

*Proof.* Define $s_i = A_{i,1}^\top \rho p_1 + A_{i,2}^\top (1-\rho) p_2$. We compute the gradient of $F(r, p_1)$ with respect to $p_1$:

$$\nabla_{p_1} F(r, p_1) = \frac{1}{n} \sum_{i=1}^n \exp(s_i) \rho A_{i,1}$$

Let us write $r^\star(p_1')$ as $[\rho^\star(p_1'), (1 - \rho^\star(p_1')) p_2^\star(p_1')]$. Then, the gradient of $F(r, p_1)$ at $r^\star(p_1'), p_1^t$ is

$$\nabla_{p_1} F(r^\star(p_1'), p_1^t) = \frac{1}{n} \sum_{i=1}^n \exp(A_{i,1}^\top \rho^\star(p_1') p_1^t + A_{i,2}^\top (1 - \rho^\star(p_1')) p_2^\star(p_1')) \rho^\star(p_1') A_{i,1}$$

We now bound the norm of the gradient over $t \in [0, 1]$. First, we add and subtract $p_1'$ to create an $\exp$ term defined over $p_1'$ and a difference term with $p_1^t - p_1'$:

$$\|\nabla_{p_1} F(r^\star(p_1'), p_1^t)\| \leq \frac{1}{n} \sum_{i=1}^n \exp(A_{i,1}^\top \rho^\star(p_1')(p_1' + p_1^t - p_1') + A_{i,2}^\top (1 - \rho^\star(p_1')) p_2^\star(p_1')) \rho^\star(p_1') \|A_{i,1}\|$$

$$\leq \frac{1}{n} \sum_{i=1}^n \exp(A_{i,1}^\top \rho^\star(p_1') p_1' + A_{i,2}^\top (1 - \rho^\star(p_1')) p_2^\star(p_1')) \exp(A_{i,1}^\top \rho^\star(p_1')(p_1^t - p_1')) \|A_{i,1}\|$$

Then, we apply Cauchy-Schwarz inequality to get

$$\|\nabla_{p_1} F(r^\star(p_1'), p_1^t)\|$$

$$\leq \frac{1}{n} \sqrt{\sum_{i=1}^n \exp(2 A_{i,1}^\top \rho^\star(p_1')(p_1^t - p_1')) \|A_{i,1}\|^2} \sqrt{\sum_{i=1}^n \exp(2(A_{i,1}^\top \rho^\star(p_1') p_1' + A_{i,2}^\top (1 - \rho^\star(p_1')) p_2^\star(p_1')))}$$

$$\leq \frac{1}{n} \sqrt{\sum_{i=1}^n \exp(2 A_{i,1}^\top \rho^\star(p_1')(p_1^t - p_1')) \|A_{i,1}\|^2 \cdot \sum_{i=1}^n \exp(A_{i,1}^\top \rho^\star(p_1') p_1' + A_{i,2}^\top (1 - \rho^\star(p_1')) p_2^\star(p_1'))}$$

We observe that we can replace the second summation with $\bar{F}(q^\star(p_1'))$, so we only need to further simplify $\sqrt{\sum_{i=1}^n \exp(2 A_{i,1}^\top \rho^\star(p_1')(p_1^t - p_1')) \|A_{i,1}\|^2}$. We observe that $\|p_1^t - p_1'\|$ is always bounded by $\|\triangle\|$, and use the fact that $\rho^\star(p_1') \leq 1$:

$$\sqrt{\sum_{i=1}^n \exp(2 A_{i,1}^\top \rho^\star(p_1')(p_1^t - p_1')) \|A_{i,1}\|^2} \leq \sqrt{\sum_{i=1}^n \exp(2 \|A_{i,1}\| \|p_1^t - p_1'\|) \|A_{i,1}\|^2}$$

$$\leq \sqrt{\sum_{i=1}^n \exp(2 \|A_{i,1}\| \|\triangle\|) \|A_{i,1}\|^2} \leq \|\exp(\alpha_{fix} \|\triangle\|) \odot \alpha_{fix}\|$$

Therefore,

$$\|\nabla_{p_1} F(r^\star(p_1'), p_1^t)\| \leq \bar{F}(q^\star(p_1')) \|\exp(\alpha_{fix} \|\triangle\|) \odot \alpha_{fix}\|$$

$\square$

**Lemma E.10.** *Let $a_{\max} = \max_i\{\|A_{i,1}\|\}$, the largest norm of $A_{i,1}$ across all tasks. Define $b_{\max} \in \mathbb{R}^n$ where $b_{i,\max} = \max\{|\langle A_{i,1},p_1\rangle|,|\langle A_{i,1},p_1'\rangle|\}$ as the larger of the two dot products per task. The gradient norm $\|\nabla^2_{r,p_1}F(r^\star(p_1'),p_1^t)\|$ can be bounded by*

$$\|\nabla^2_{r,p_1}F(r^\star(p_1'),p_1^t)\| \le \exp(a_{\max}\|\Delta\|)\cdot \bar{F}(q^\star(p_1'))\cdot\big\|(\mathbf{1}+b_{\max}+\alpha_{comp})\odot\alpha_{fix}\big\|$$

*where $\odot$ is the Hadamard product.*

*Proof.* We derive an expression for $\nabla^2_{r,p_1}F(r,p_1)$. We write $r$ as $[\rho,b_2]$, where $b_2 := (1-\rho)p_2$, and define $s_i = A_{i,1}^\top\rho p_1 + A_{i,2}^\top b_2$ to be the expression inside the $\exp$. First, we have that $\nabla_{p_1}F(r,p_1)$ is

$$\nabla_{p_1}F(r,p_1) = \frac{1}{n}\sum_{i=1}^n \exp(s_i)\rho A_{i,1}.$$

Then, we can compute $\nabla_{\rho,p_1}F(r,p_1)$ and $\nabla_{b_2,p_1}F(r,p_1)$:

$$\nabla_{\rho,p_1}F(r,p_1) = \nabla_\rho\left(\frac{1}{n}\sum_{i=1}^n \exp(s_i)\rho A_{i,1}\right) = \frac{1}{n}\sum_{i=1}^n (\rho\exp(s_i)\langle A_{i,1},p_1\rangle + \exp(s_i))A_{i,1}^\top$$

$$\nabla_{b_2,p_1}F(r,p_1) = \nabla_{b_2}\left(\frac{1}{n}\sum_{i=1}^n \exp(s_i)\rho A_{i,1}\right) = \frac{1}{n}\sum_{i=1}^n \rho\exp(s_i)A_{i,2}A_{i,1}^\top$$

We can consolidate these into

$$\nabla^2_{r,p_1}F(r,p_1) = \frac{1}{n}\sum_{i=1}^n \exp(s_i)(\rho a_i + e_1)A_{i,1}^\top,$$

where $a_i = \begin{bmatrix}\langle A_{i,1},p_1\rangle \\ A_{i,2}\end{bmatrix} \in \mathbb{R}^{|\mathcal{D}_2'|+1}$ and $e_1$ is the standard basis vector $\begin{bmatrix}1\\0\\ \vdots \\0\end{bmatrix} \in \mathbb{R}^{|\mathcal{D}_2'|+1}$. Therefore, the gradient $\nabla^2_{r,p_1}F(r,p_1)$ at $r^\star(p_1'),p_1^t$ is

$$\nabla^2_{r,p_1}F(r^\star(p_1'),p_1^t) = \frac{1}{n}\sum_{i=1}^n \exp(A_{i,1}^\top\rho^\star(p_1')p_1^t + A_{i,2}^\top(1-\rho^\star(p_1'))p_2^\star(p_1'))(\rho^\star(p_1')a_i + e_1)A_{i,1}^\top$$

where $a_i$ is now $\begin{bmatrix}\langle A_{i,1},p_1^t\rangle \\ A_{i,2}\end{bmatrix}$. Now, we would like to upper bound $\sup_{t\in[0,1]}\|\nabla^2_{r,p_1}F(r^\star(p_1'),p_1^t)\|$. Using Cauchy-Schwarz inequality, we have

$$\|\nabla^2_{r,p_1}F(r^\star(p_1'),p_1^t)\| \le \frac{1}{n}\sum_{i=1}^n \exp(A_{i,1}^\top\rho^\star(p_1')p_1^t + A_{i,2}^\top(1-\rho^\star(p_1'))p_2^\star(p_1'))\|(\rho^\star(p_1')a_i + e_1)A_{i,1}^\top\|$$

$$\le \frac{1}{n}\sqrt{\sum_{i=1}^n \exp\big(2(A_{i,1}^\top\rho^\star(p_1')p_1^t + A_{i,2}^\top(1-\rho^\star(p_1'))p_2^\star(p_1'))\big)}\sqrt{\sum_{i=1}^n\|(\rho^\star(p_1')a_i + e_1)A_{i,1}^\top\|^2} \tag{14}$$

We simplify the first summation in (14) by adding and subtracting $p_1'$ to $p_1^t$:

$$\sum_{i=1}^{n} \exp\big(2(A_{i,1}^{\top}\rho^{\star}(p_1')p_1^t + A_{i,2}^{\top}(1-\rho^{\star}(p_1'))p_2^{\star}(p_1')))\big)$$

$$= \sum_{i=1}^{n} \exp\big(2A_{i,1}^{\top}\rho^{\star}(p_1')(p_1' + p_1^t - p_1') + 2A_{i,2}^{\top}(1-\rho^{\star}(p_1'))p_2^{\star}(p_1')\big)$$

$$= \sum_{i=1}^{n} \exp\big(2(A_{i,1}^{\top}\rho^{\star}(p_1')p_1' + A_{i,2}^{\top}(1-\rho^{\star}(p_1'))p_2^{\star}(p_1')) + 2A_{i,1}^{\top}\rho^{\star}(p_1')(p_1^t - p_1')\big)$$

$$\leq \max_{i}\{\exp(2A_{i,1}^{\top}\rho^{\star}(p_1')(p_1^t - p_1'))\}\left(\sum_{i=1}^{n}\exp\big(A_{i,1}^{\top}\rho^{\star}(p_1')p_1' + A_{i,2}^{\top}(1-\rho^{\star}(p_1'))p_2^{\star}(p_1')\big)\right)^2$$

We note that $\sum_{i=1}^{n} \exp\big(A_{i,1}^{\top}\rho^{\star}(p_1')p_1' + A_{i,2}^{\top}(1 - \rho^{\star}(p_1'))p_2^{\star}(p_1')\big)$ is $n \cdot \bar{F}(q^{\star}(p_1'))$. Moreover, using the fact that $\rho^{\star}(p_1') \leq 1$ and $\|p_1^t - p_1'\| \leq \|\Delta\|$, we can bound $\max_i\{\exp(2A_{i,1}^{\top}\rho^{\star}(p_1')(p_1^t - p_1'))\} \leq \max_i\{\exp(2\|A_{i,1}\|\|p_1^t - p_1'\|)\} \leq \exp(2\|\Delta\|\max_i\{\|A_{i,1}\|\})$. Let $a_{\max} = \max_i\{\|A_{i,1}\|\}$, the largest norm of $A_{i,1}$ across all tasks. Therefore, we have:

$$\frac{1}{n}\sqrt{\sum_{i=1}^{n}\exp\big(2(A_{i,1}^{\top}\rho^{\star}(p_1')p_1^t + A_{i,2}^{\top}(1-\rho^{\star}(p_1'))p_2^{\star}(p_1')))\big)} \leq \frac{1}{n}\sqrt{\exp(2a_{\max}\|\Delta\|) \cdot n^2 \cdot \bar{F}(q^{\star}(p_1'))^2}$$

$$= \exp(a_{\max}\|\Delta\|) \cdot \bar{F}(q^{\star}(p_1'))$$

Next, we simplify the second summation in (14), $\sum_{i=1}^{n}\|(\rho^{\star}(p_1')a_i + e_1)A_{i,1}^{\top}\|^2$, using Cauchy-Schwarz inequality, the definition of the norm, and the fact that $\rho^{\star}(p_1') \leq 1$:

$$\sum_{i=1}^{n}\|(\rho^{\star}(p_1')a_i + e_1)A_{i,1}^{\top}\|^2 \leq \sum_{i=1}^{n}\|\rho^{\star}(p_1')a_i + e_1\|^2\|A_{i,1}\|^2$$

$$= \sum_{i=1}^{n}((1+\rho^{\star}(p_1')\langle A_{i,1}, p_1^t\rangle)^2 + \rho^{\star}(p_1')^2\|A_{i,2}\|^2)\|A_{i,1}\|^2$$

$$\leq \sum_{i=1}^{n}((1+\rho^{\star}(p_1')\langle A_{i,1}, p_1^t\rangle)^2 + \|A_{i,2}\|^2)\|A_{i,1}\|^2$$

Note that $\langle A_{i,1}, p_1^t\rangle$ is bound by $\langle A_{i,1}, p_1'\rangle$ and $\langle A_{i,1}, p_1\rangle$. Therefore, it holds that $(1 + \rho^{\star}(p_1')\langle A_{i,1}, p_1^t\rangle)^2 \leq (1 + \max\{|\langle A_{i,1}, p_1\rangle|, |\langle A_{i,1}, p_1'\rangle|\})^2$. We denote $b_{i,\max} = \max\{|\langle A_{i,1}, p_1\rangle|, |\langle A_{i,1}, p_1'\rangle|\}$ as the larger of the two dot products per task, and then we can write:

$$\sqrt{\sum_{i=1}^{n}\|(\rho^{\star}(p_1')a_i + e_1)A_{i,1}^{\top}\|^2} \leq \sqrt{\sum_{i=1}^{n}((1+b_{i,\max})^2 + \|A_{i,2}\|^2)\|A_{i,1}\|^2}$$

$$\leq \sqrt{\sum_{i=1}^{n}(1+b_{i,\max}+\|A_{i,2}\|)^2\|A_{i,1}\|^2}$$

$$= \|(\mathbf{1}+b_{\max}+\alpha_{\text{comp}})\odot\alpha_{\text{fix}}\|$$

where addition is elementwise, $\odot$ is the Hadamard product, and $\alpha_{\text{fix}}, \alpha_{\text{comp}} \in \mathbb{R}^n$ are constructed where $\alpha_{i,1} = \|A_{i,1}\|$ and $\alpha_{i,2} = \|A_{i,2}\|$. Putting everything together, our gradient norm bound in (14) becomes

$$\|\nabla_{r,p_1}^2 F(r^{\star}(p_1'), p_1^t)\| \leq \exp(a_{\max}\|\Delta\|) \cdot \bar{F}(q^{\star}(p_1')) \cdot \|(\mathbf{1}+b_{\max}+\alpha_{\text{comp}})\odot\alpha_{\text{fix}}\|$$

$\square$

E.1.3. THEOREM 4.1 (FORMAL)

**Theorem E.11.** *Define* $a_{\max} = \max_i \{\|A_{i,1}\|\}$. *The performance gap between using* $q^\star$ *and* $q^\star(\tilde{p}_{\mathcal{D}_{\text{fix}}})$ *is bounded by:*

$$F(q^\star(\tilde{p}_{\mathcal{D}_{\text{fix}}})) - F(q^\star) \leq C \|\tilde{p}_{\mathcal{D}_{\text{fix}}} - q^\star_{\mathcal{D}_{\text{fix}}}\|,$$

*where*

$$C = \bar{F}(q^\star)\exp(a_{\max}\|\tilde{p}_{\mathcal{D}_{\text{fix}}} - q^\star_{\mathcal{D}_{\text{fix}}}\|)\left(\frac{\bar{F}(q^\star(\tilde{p}_{\mathcal{D}_{\text{fix}}}))}{\mu}\cdot\|\alpha_{\text{fix}}+\alpha_{comp}\|\cdot\kappa(\alpha_{\text{fix}},\alpha_{comp})+\|\alpha_{\text{fix}}\|\right)$$

*Proof.* Recall our simplified notation $p_1 = \tilde{p}_{\mathcal{D}_{\text{fix}}}$, $p'_1 = q^\star_{\mathcal{D}_{\text{fix}}}$. Applying Lemma E.6 to Lemma E.7, we can bound the performance gap to be linear in $\|p_1 - p'_1\|$:

$$F(r^\star(p_1),p_1) - F(r^\star(p'_1),p'_1) \leq C\|\Delta\|$$

where $C$ is a term that consists of three gradient norms:

$$C = \frac{1}{\mu}\sup_{t\in[0,1]}\left(\|\nabla^2_{r,p_1}F(r^\star(p'_1),p_1^t)\|\right)\|\nabla_r F(r^\star(p_1),p_1)\| + \sup_{t\in[0,1]}\|\nabla_{p_1}F(r^\star(p'_1),p_1^t)\|$$

and $p_1^t = tp'_1 + (1-t)p_1$. We can apply the gradient norm bounds from Lemmas E.8, E.9, E.10:

$$C \leq \frac{1}{\mu}\exp(a_{\max}\|\Delta\|)\cdot\bar{F}(q^\star(p'_1))\cdot\|(\mathbf{1}+b_{\max}+\alpha_{\text{comp}})\odot\alpha_{\text{fix}}\|\cdot\bar{F}(q^\star(p_1))\|\|A_1 p_1|+\alpha_{\text{comp}}\|$$
$$+\bar{F}(q^\star(p'_1))\|\exp(\alpha_{\text{fix}}\|\triangle\|)\odot\alpha_{\text{fix}}\|$$

We simplify this bound a bit further. First, note that the elements of $|A_1 p_1|$ are $|\langle A_{i,1},p_1\rangle| \leq \|A_{i,1}\|\|p_1\| \leq \|A_{i,1}\|$. Therefore, $|A_1 p_1| \preceq \alpha_{\text{fix}}$.

Second, note that $b_{i,\max} = \max\{|\langle A_{i,1},p_1\rangle|,|\langle A_{i,1},p'_1\rangle|\} \leq \|A_{i,1}\|$, so $b_{\max} \preceq \alpha_{\text{fix}}$ as well.

Lastly, $\|\exp(\alpha_{\text{fix}}\|\triangle\|)\odot\alpha_{\text{fix}}\|$ can be bounded by $\exp(a_{\max}\|\Delta\|)\cdot\|\alpha_{\text{fix}}\|$.

Then, we can bound $C$ as

$$C \leq \bar{F}(q^\star)\exp(a_{\max}\|\Delta\|)\left(\frac{\bar{F}(q^\star(\tilde{p}_{\mathcal{D}_{\text{fix}}}))}{\mu}\|(\mathbf{1}+\alpha_{\text{fix}}+\alpha_{\text{comp}})\odot\alpha_{\text{fix}}\|\cdot\|\alpha_{\text{fix}}+\alpha_{\text{comp}}\|+\|\alpha_{\text{fix}}\|\right)$$

Replacing $\|(\mathbf{1}+\alpha_{\text{fix}}+\alpha_{\text{comp}})\odot\alpha_{\text{fix}}\|$ with $\kappa(\alpha_{\text{fix}},\alpha_{\text{comp}})$ completes our proof.

$\square$

## E.2. Proof for Theorem 4.2

When domains are being added and $\tilde{p}$ is the optimal mix on $\mathcal{D}$, $\tilde{p}$ is equal to $\tilde{p}_{\mathcal{D}_{\text{fix}}}$ is the solution to the following optimization problem:

$$\text{minimize}_{p\in\triangle^{m-1}}\frac{1}{n}\sum_{i=1}^m f_i(\text{LM}(S,R,p)) \tag{15}$$

$$\text{s.t.}\quad p_j \leq \frac{kN_j}{R}\quad \forall j\in[m]$$

**Assumption E.12.** We make the following assumptions beyond Assumption E.2.

1. We assume that $F(r,p_1)$ is $\mu_1$-strongly convex in $p_1$.

2. mutual feasibility

3. Let $\mathcal{S}(r)$ be the feasible set of (2) defined by the repetition constraints on $p_1$:

$$p_j \leq \frac{kN_j}{R\rho} \, \forall j \in \mathcal{D}_{\text{fix}}$$

We assume **mutual feasibility** of $\tilde{p}_{\mathcal{D}_{\text{fix}}}$ and $q^\star_{\mathcal{D}_{\text{fix}}}$: that $\tilde{p}_{\mathcal{D}_{\text{fix}}} \in \mathcal{S}(r^\star)$, and $q^\star_{\mathcal{D}_{\text{fix}}} \in \mathcal{S}(r_0)$, where $r_0 = [1, \mathbf{0}]$ corresponds to the mix before new domains are added. That is, we assume that $\tilde{p}_j \leq \frac{kN_j}{R\rho^\star}$ and $q^\star_j \leq \frac{kN_j}{R}$ for $j \in \mathcal{D}_{\text{fix}}$, meaning that the existing mix cannot have active repetition constraints.

First, we present a general lemma that allows us to bound any $\|p_1 - p_1'\|$.

**Lemma E.13** (Argmin stability with mutual feasibility). *Let $\mathcal{S}, \mathcal{S}' \subset \triangle^{\|D_1\|-1}$ be convex sets. Let $h : \mathcal{S} \to \mathbb{R}$ be differentiable and $\mu_1$-strongly convex on $\mathcal{S}$. Let $h' : \mathcal{S}' \to \mathbb{R}$ be a differentiable function. Define*

$$p_1 = \arg\min_{p_1 \in \mathcal{S}} h(p_1), \qquad p_1' = \arg\min_{p_1 \in \mathcal{S}'} h'(p_1)$$

*We assume mutual feasibility; that is, $p_1 \in \mathcal{S}'$ and $p_1' \in \mathcal{S}$. Then,*

$$\|p_1 - p_1'\| \leq \frac{1}{\mu_1} \|\nabla h(p_1') - \nabla h'(p_1')\|.$$

*Proof.* Applying our mutual feasibility assumption, the first order condition at $p_1$ for $h$ and $p_1'$ for $h$ yields

$$\langle \nabla h(p_1), p_1' - p_1 \rangle \geq 0$$
$$\langle \nabla h'(p_1'), p_1 - p_1' \rangle \geq 0$$

Adding these together and adding and subtracting $\nabla h(p_1')$, we have

$$\langle \nabla h(p_1) - \nabla h'(p_1'), p_1' - p_1 \rangle \geq 0$$
$$\Rightarrow \langle \nabla h(p_1) - \nabla h(p_1') + \nabla h(p_1') - \nabla h'(p_1'), p_1' - p_1 \rangle \geq 0$$
$$\Rightarrow \langle \nabla h(p_1) - \nabla h(p_1'), p_1 - p_1' \rangle \leq \langle \nabla h(p_1') - \nabla h'(p_1'), p_1' - p_1 \rangle$$

Next, due to $\mu_1$ strong convexity of $h$, we have

$$h(p_1) \geq h(p_1') + \langle \nabla h(p_1'), p_1 - p_1' \rangle + \frac{\mu_1}{2} \|p_1 - p_1'\|^2$$
$$h(p_1') \geq h(p_1) + \langle \nabla h(p_1), p_1' - p_1 \rangle + \frac{\mu_1}{2} \|p_1 - p_1'\|^2$$

Adding these together and applying the bound from the first order conditions, we get

$$\mu \|p_1 - p_1'\|^2 \leq \langle \nabla h(p_1) - \nabla h(p_1'), p_1 - p_1' \rangle$$
$$\leq \langle \nabla h(p_1') - \nabla h'(p_1'), p_1' - p_1 \rangle$$
$$\leq \|\nabla h(p_1') - \nabla h'(p_1')\| \cdot \|p_1' - p_1\|$$

Dividing both sides by $\mu \|p_1' - p_1\|$ completes our proof. $\square$

**Lemma E.14** (When $\tilde{p}_{\mathcal{D}_{\text{fix}}} = q^\star_{\mathcal{D}_{\text{fix}}}$ for adding). *Consider the* adding *update, where $\mathcal{D}_2 = \emptyset$ and $\mathcal{D}' = [\mathcal{D}_1 \mathcal{D}_2']$ with $\mathcal{D}_2' \neq \emptyset$. Let $\tilde{p} \in \triangle^{m-1}$ be an optimal solution to the mixing problem on the pre-update domain set $\mathcal{D} = \mathcal{D}_1$ (equivalently, $\tilde{p}_{\mathcal{D}_1} = \tilde{p}$), and let $q^\star$ be the optimal solution to the post-update mixing problem on $\mathcal{D}'$. If $\rho^\star = 1$, then*

$$\tilde{p}_{\mathcal{D}_{\text{fix}}} = q^\star_{\mathcal{D}_{\text{fix}}}.$$

*Proof.* Since $\mathcal{D}_2 = \emptyset$, the pre-update problem over $\mathcal{D} = \mathcal{D}_1$ is equivalent to the post-update problem restricted to mixtures that assign zero mass to all added domains:

$$\text{minimize}_{q \in \triangle^{m'-1}} \frac{1}{n} \sum_{i=1}^{n} f_i(\text{LM}(S,R,q))$$

$$\text{s.t.} \quad q_j \leq \frac{kN_j'}{R} \quad \forall j \in \mathcal{D}_1 \cup \mathcal{D}_2'$$

$$q_j = 0 \quad \forall j \in \mathcal{D}_2'.$$

If $\rho^\star = 1$, then $q_j^\star = 0$ for all $j \in \mathcal{D}_2'$. This means that the additional constraints $q_j = 0$ on the post-update problem do not change the optimal value, since the restricted feasible set still contains the optimal solution of $q_j^\star = 0$. Therefore, if $\rho^\star = 1$, the restricted problem above and the unrestricted post-update problem have the same optimum, and their optimizers coincide on $\mathcal{D}_1$; in particular, $q_{\mathcal{D}_1}^\star = \tilde{p}_{\mathcal{D}_1}$. Replacing $\mathcal{D}_1$ with $\mathcal{D}_{\text{fix}}$ completes our proof. $\qquad\square$

**Lemma E.15.** *Assume mutual feasibility between $p_1$ and $p_1'$. Then, when the domain update involves adding domains,*

$$\|p_1 - p_1'\| \leq \frac{2(1-\rho^\star)}{\mu_1} \| \sup_{t \in [0,1]} \|\nabla_{p_1,r}^2 F(r^t,p_1')\| \qquad r^t = tr_0 + (1-t)r^\star$$

*Proof.* Define $r_0 = [1,\mathbf{0}]$ where $\rho_0 = 1$. Then, $p_1 = \arg\min_{p_1 \in \mathcal{S}(r_0)} F(r_0, p_1)$. We can write $p_1'$ similarly as $p_1' = \text{argmin}_{p_1 \in \mathcal{S}(r^\star)} F(r^\star, p_1)$.

If we assume mutual feasibility and define $h(p_1) = F(r_0,p_1), h'(p_1) = F(r^\star,p_1)$, then applying Lemma E.13 gives us

$$\|p_1 - p_1'\| \leq \frac{1}{\mu_1} \|\nabla_{p_1} F(r_0,p_1') - F(r^\star,p_1')\|$$

Define $r^t = tr_0 + (1-t)r^\star$, where $t \in [0,1]$. By the fundamental theorem of calculus, it holds that

$$\nabla_{p_1} F(r_0,p_1') - \nabla_{p_1} F(r^\star,p_1') = \int_0^1 \nabla_{p_1,r}^2 F(r^t,p_1')(r_0 - r^\star)dt$$

Therefore,

$$\|p_1 - p_1'\| \leq \frac{1}{\mu_1} \|r_0 - r^\star\| \sup_{t \in [0,1]} \|\nabla_{p_1,r}^2 F(r^t,p_1')\|$$

Lastly, we bound $\|r_0 - r^\star\|$:

$$\|r_0 - r^\star\| = \|[1,\mathbf{0}] - [\rho^\star,(1-\rho^\star)q_{\mathcal{D}_2'}^\star]\| = \|(1-\rho^\star)[1,-q_{\mathcal{D}_2'}^\star]\| \leq 2(1-\rho^\star)$$

This completes our proof. $\qquad\square$

**Lemma E.16.** *Define $r^t = tr_0 + (1-t)r^\star(p_1')$, where $r_0 = [1,\mathbf{0}]$. The gradient norm $\|\nabla_{r,p_1}^2 F(r^t,p_1')\|$ can be bounded by*

$$\|\nabla_{r,p_1}^2 F(r^t,p_1')\| \leq \bar{F}(q^\star)\exp(c_{\max}(1-\rho^\star))\|(1+\alpha_{fix}+\alpha_{comp}) \odot \alpha_{fix}\|$$

*where $c_{\max} = max_i\{\|A_{i,1}\| + \|A_{i,2}\|\}$ and $\odot$ is the Hadamard product.*

*Proof.* First, we can write $r^t$ as

$$r^t = t[1,\mathbf{0}] + (1-t)[\rho^\star(p_1'),(1-\rho^\star(p_1'))p_2^\star(p_1')]$$
$$= [t+(1-t)\rho^\star(p_1'),(1-t)(1-\rho^\star(p_1'))p_2^\star(p_1')]$$
$$= [t+(1-t)\rho^\star(p_1'),(1-(t+(1-t)\rho^\star(p_1')))p_2^\star(p_1')]$$

Therefore, we can express $r^t = [\rho^t, (1-\rho^t)p_2^t]$ where $\rho^t := t + (1-t)\rho^\star(p_1')$, and $p_2^t = p_2^\star(p_1')$.

Applying Lemma E.10, we can write the gradient as

$$\nabla_{r,p_1}^2 F(r^t, p_1') = \frac{1}{n}\sum_{i=1}^n \exp(A_{i,1}^\top \rho^t p_1' + A_{i,2}^\top(1-\rho^t)p_2^t)(\rho^t a_i + e_1)A_{i,1}^\top$$

where $a_i = \begin{bmatrix} \langle A_{i,1}, p_1' \rangle \\ A_{i,2} \end{bmatrix} \in \mathbb{R}^{|\mathcal{D}_2'|+1}$ and $e_1$ is the standard basis vector $\begin{bmatrix} 1 \\ 0 \\ \vdots \\ 0 \end{bmatrix} \in \mathbb{R}^{|\mathcal{D}_2'|+1}$. Then, using Cauchy-Schwarz inequality,

the gradient norm can be bounded as

$$\|\nabla_{r,p_1}^2 F(r^t, p_1')\| \le \frac{1}{n}\sum_{i=1}^n \exp(A_{i,1}^\top \rho^t p_1' + A_{i,2}^\top(1-\rho^t)p_2^t)\|(\rho^t a_i + e_1)A_{i,1}^\top\| \tag{16}$$

$$\le \frac{1}{n}\sqrt{\sum_{i=1}^n \exp(2(A_{i,1}^\top \rho^t p_1' + A_{i,2}^\top(1-\rho^t)p_2^t))}\sqrt{\sum_{i=1}^n \|(\rho^t a_i + e_1)A_{i,1}^\top\|^2}$$

We bound the first summation of (16) by adding and subtracting $\rho^\star(p_1')$:

$$\sum_{i=1}^n \exp(2(A_{i,1}^\top \rho^t p_1' + A_{i,2}^\top(1-\rho^t)p_2^t))$$

$$= \sum_{i=1}^n \exp(2(A_{i,1}^\top(\rho^t + \rho^\star(p_1') - \rho^\star(p_1'))p_1' + A_{i,2}^\top((1-\rho^t) + (1-\rho^\star(p_1')) - (1-\rho^\star(p_1')))p_2^t))$$

$$= \sum_{i=1}^n \exp(2(A_{i,1}^\top \rho^\star(p_1')p_1' + A_{i,2}^\top(1-\rho^\star(p_1'))p_2^t))\exp(2(A_{i,1}^\top(\rho^t - \rho^\star(p_1'))p_1' + A_{i,2}^\top(\rho^\star(p_1') - \rho^t)p_2^\star(p_1')))$$

$$= \sum_{i=1}^n \exp(2(A_{i,1}^\top \rho^\star(p_1')p_1' + A_{i,2}^\top(1-\rho^\star(p_1'))p_2^t))\exp(2t(1-\rho^\star(p_1'))(A_{i,1}^\top p_1' - A_{i,2}^\top p_2^\star(p_1')))$$

We can then replace some terms in (16) with $\bar{F}(r^\star(p_1'), p_1')$

$$\frac{1}{n}\sqrt{\sum_{i=1}^n \exp(2(A_{i,1}^\top \rho^t p_1' + A_{i,2}^\top(1-\rho^t)p_2^t))}$$

$$\le \frac{1}{n}\sum_{i=1}^n \exp(A_{i,1}^\top \rho^\star(p_1')p_1' + A_{i,2}^\top(1-\rho^\star(p_1'))p_2^t) \cdot \max_i\{\exp(t(1-\rho^\star(p_1'))(A_{i,1}^\top p_1' - A_{i,2}^\top p_2^\star(p_1')))\}$$

$$= \bar{F}(r^\star(p_1'), p_1')\max_i\{\exp(t(1-\rho^\star(p_1'))(A_{i,1}^\top p_1' - A_{i,2}^\top p_2^\star(p_1')))\}$$

We can bound $A_{i,1}^\top p_1' - A_{i,2}^\top p_2^\star(p_1') \le \|A_{i,1}\| + \|A_{i,2}\|$ using Cauchy-Schwarz. Let $c_{\max} = \max_i\{\|A_{i,1}\| + \|A_{i,2}\|\}$. Using the fact that $t \in [0,1]$, the first part of (16) can be bounded by

$$\frac{1}{n}\sqrt{\sum_{i=1}^n \exp(2(A_{i,1}^\top \rho^t p_1' + A_{i,2}^\top(1-\rho^t)p_2^t))} \le \bar{F}(r^\star(p_1'), p_1')\exp(c_{\max}(1-\rho^\star(p_1')))$$

The second summation of (16) can be bounded in the exact same way as the second summation of (14) in Lemma E.10. Therefore, we have that

$$\sqrt{\sum_{i=1}^n \|(\rho^t a_i + e_1)A_{i,1}^\top\|^2} \le \|(1 + \alpha_{\text{fix}} + \alpha_{\text{comp}}) \odot \alpha_{\text{fix}}\|$$

This completes our proof. □

**Theorem E.17.** *Define $\tilde{p}$ is the solution to (2) on $\mathcal{D}$ and suppose that new domains are added. Let $c_{\max} = \max_i \{\|A_{i,1}\| + \|A_{i,2}\|\}$. Then, the difference between $\tilde{p}_{\mathcal{D}_{fix}}$ and $q^{\star}_{\mathcal{D}_{fix}}$ is bounded by*

$$\|\tilde{p}_{\mathcal{D}_{fix}} - q^{\star}_{\mathcal{D}_{fix}}\| \leq \frac{2\bar{F}(q^{\star})\exp(c_{\max}(1-\rho^{\star}))}{\mu_1} \cdot \kappa(\alpha_{fix}, \alpha_{comp}) \cdot (1-\rho^{\star})$$

*Proof.* Applying Lemma E.16 and Lemma E.15, we have

$$\|p_1 - p_1'\| \leq \frac{2(1-\rho^{\star})}{\mu_1} \bar{F}(q^{\star})\exp(c_{\max}(1-\rho^{\star}))\|(1+\alpha_{fix}+\alpha_{comp})\odot\alpha_{fix}\|$$

$$= \frac{2\bar{F}(q^{\star})\exp(c_{\max}(1-\rho^{\star}))}{\mu_1} \cdot \kappa(\alpha_{fix}, \alpha_{comp}) \cdot (1-\rho^{\star})$$

□

## E.3. Additional Theoretical Results

### E.3.1. REMOVING DOMAINS

Our results for are symmetric to the addition update, but we state the full derivation for completeness.

**Lemma E.18** (When $\tilde{p}_{\mathcal{D}_{fix}} = q^{\star}_{\mathcal{D}_{fix}}$ for removing). *Consider the removing update, where $\mathcal{D}_2 \neq \emptyset$ and $\mathcal{D}' = \mathcal{D}_1$ (equivalently, $\mathcal{D}_2' = \emptyset$). Let $\tilde{p} \in \triangle^{m-1}$ be an optimal solution to the mixing problem on the pre-update domain set $\mathcal{D} = [\mathcal{D}_1, \mathcal{D}_2]$, and let $q^{\star}$ be the optimal solution to the post-update mixing problem on $\mathcal{D}' = \mathcal{D}_1$ (equivalently, $q^{\star}_{\mathcal{D}_1} = q^{\star}$). If $\pi^{\star} = 1$, then*

$$\tilde{p}_{\mathcal{D}_{fix}} = q^{\star}_{\mathcal{D}_{fix}}.$$

*Proof.* Since $\mathcal{D}_2' = \emptyset$, the post-update problem over $\mathcal{D}' = \mathcal{D}_1$ is equivalent to the pre-update problem restricted to mixtures that assign zero mass to all removed domains:

$$\text{minimize}_{p \in \triangle^{m-1}} \frac{1}{n}\sum_{i=1}^{n} f_i(\text{LM}(S,R,p))$$

$$\text{s.t.} \quad p_j \leq \frac{kN_j}{R} \quad \forall j \in \mathcal{D}_1 \cup \mathcal{D}_2$$

$$p_j = 0 \quad \forall j \in \mathcal{D}_2'.$$

If $\pi^{\star} = 1$, then $\tilde{p}_j = 0$ for all $j \in \mathcal{D}_2$. This means that the additional constraints $p_j = 0$ on the pre-update problem do not change the optimal value, since the restricted feasible set still contains the optimal solution of $\tilde{p}_j = 0$. Therefore, if $\pi^{\star} = 1$, the restricted problem above and the unrestricted pre-update problem have the same optimum, and their optimizers coincide on $\mathcal{D}_1$; in particular, $q^{\star}_{\mathcal{D}_1} = \tilde{p}_{\mathcal{D}_1}$. Replacing $\mathcal{D}_1$ with $\mathcal{D}_{fix}$ completes our proof. □

### E.3.2. PARTITIONING A DOMAIN

**Lemma E.19** (When $\tilde{p}_{\mathcal{D}_{fix}} = q^{\star}_{\mathcal{D}_{fix}}$ for partitioning). *Consider the partitioning update operator, where $\mathcal{D} \setminus \mathcal{D}_1 = \{D\}$ consists of a single domain and $\mathcal{D}_2' = \{D_1', ..., D_\ell'\}$ are subdomains with $D = \bigcup_{j=1}^{\ell} D_j'$.*

*Define the natural distribution $p_{\text{nat}} \in \triangle^{\ell-1}$ by*

$$[p_{\text{nat}}]_j = \frac{N_j'}{\sum_{t=1}^{\ell} N_t'} \quad \forall j \in [\ell].$$

*If $q^{\star}_{\mathcal{D}_2'} = p_{\text{nat}}$, then*

$$\tilde{p}_{\mathcal{D}_{fix}} = q^{\star}_{\mathcal{D}_{fix}}.$$

*Proof.* Since $\mathcal{D}_2 = \{D\}$ is a single domain, sampling $(1-\pi)R$ tokens from $D$ induces a distribution over the subdomains $\{D'_1,...,D'_\ell\}$ that is proportional to token counts, i.e., $p_{\text{nat}}$. Therefore, the pre-update optimization problem can be written as the post-update optimization problem restricted to mixtures that preserve this conditional distribution within the partition:

$$\text{minimize}_{q \in \triangle^{m'-1}} \frac{1}{n} \sum_{i=1}^{n} f_i(\text{LM}(S,R,q))$$

$$\text{s.t.} \quad q_j \le \frac{kN'_j}{R} \quad \forall j \in \mathcal{D}_1 \cup \mathcal{D}'_2$$

$$q_{\mathcal{D}'_2} = p_{\text{nat}},$$

If $q^\star_{\mathcal{D}'_2} = p_{\text{nat}}$, then $q^\star$ lies in the restricted feasible set above. The additional constraint $q_{\mathcal{D}'_2} = p_{\text{nat}}$ does not change the optimal value of the post-update problem, since the restricted feasible set still contains the optimal solution. Therefore, the optimizer on the unrestricted post-update problem must coincide with the optimizer of the restricted problem on the unaffected domains, implying $q^\star_{\mathcal{D}_1} = \tilde{p}_{\mathcal{D}_1}$. Replacing $\mathcal{D}_1$ with $\mathcal{D}_{\text{fix}}$ completes our proof. $\square$

### E.3.3. REVISING A DOMAIN

**Lemma E.20** (Exactness for revising). *Consider the* revising *update operator, where $\mathcal{D}_2 = \{D\}$ and $\mathcal{D}'_2 = \{D'\}$ consists of a single domain whose contents are modified. Assume the log-linear model holds both pre- and post-update:*

$$F^{\text{old}}(q) = \frac{1}{n} \sum_{i=1}^{n} \exp\big((A_{i,1})^\top q_{\mathcal{D}_{\text{fix}}} + (A^{\text{old}}_{i,2})^\top q_{\mathcal{D}_2}\big), \qquad F^{\text{new}}(q) = \frac{1}{n} \sum_{i=1}^{n} \exp\big((A_{i,1})^\top q_{\mathcal{D}_{\text{fix}}} + (A^{\text{new}}_{i,2})^\top q_{\mathcal{D}_2}\big),$$

*where $\mathcal{D} = [\mathcal{D}_1, \mathcal{D}_2]$ and $\mathcal{D}' = [\mathcal{D}_1, \mathcal{D}'_2]$ have the same dimensionality, and the repetition constraints are unchanged.*

*Let $\tilde{p}$ be an optimal solution to the pre-update mixing problem with objective $F^{\text{old}}$, and let $q^\star$ be an optimal solution to the post-update mixing problem with objective $F^{\text{new}}$. If $A^{\text{new}}_{i,2} = A^{\text{old}}_{i,2}$ for all $i \in [n]$, then $\tilde{p} = q^\star$, and in particular $\tilde{p}_{\mathcal{D}_{\text{fix}}} = q^\star_{\mathcal{D}_{\text{fix}}}$.*

*Proof.* If $A^{\text{new}}_{i,2} = A^{\text{old}}_{i,2}$ for all $i \in [n]$, then $F^{\text{new}}(q) = F^{\text{old}}(q)$ for all $q$. Since the feasible set (simplex and repetition constraints) is unchanged under revising, the pre-update and post-update optimization problems are identical. Therefore their optimizers coincide, so $\tilde{p} = q^\star$. $\square$

### E.4. Empirical Validation of Theorems 4.1 and 4.2

We empirically measure the terms in Theorems 4.1 and 4.2 to assess whether they track the performance of mixture reuse.

### E.4.1. SETUP

We consider two examples of an `Add` update:

- $\mathcal{D}_1 =$ DCLM (Li et al., 2024a) partitioned into 24 topic-based domains using WebOrganizer (Wettig et al., 2025), $\mathcal{D}'_2$ = Stack-Edu (Allal et al., 2025) partitioned into 15 programming languages.

- $\mathcal{D}_1 =$ DCLM partitioned into 24 topic-based domains using WebOrganizer, $\mathcal{D}'_2$ = olmOCR Science PDFs (Olmo et al., 2025) partitioned into 21 topic-based domains using WebOrganizer.

### E.4.2. RESULTS

**Performance is correlated with the reuse gap.** Mixture reuse yields $q^\star(\tilde{p}_{\mathcal{D}_{\text{fix}}})$ from Algorithm 2 given reused ratios $\tilde{p}_{\mathcal{D}_{\text{fix}}}$. Full recomputation yields $q^\star$ from directly solving (2) for the optimal mix on $\mathcal{D}'$. Theorem 4.1 shows that the performance gap of mixture reuse, $F(q^\star(\tilde{p}_{\mathcal{D}_{\text{fix}}})) - F(q^\star)$, where $F$ is the average downstream task BPB, is bounded in terms of the reuse gap, $\|\tilde{p}_{\mathcal{D}_{\text{fix}}} - q^\star_{\mathcal{D}_{\text{fix}}}\|$, where $q^\star_{\mathcal{D}_{\text{fix}}}$ is $q^\star$ normalized over $\mathcal{D}_{\text{fix}}$. We validate that the reuse gap predicts the performance gap.

To do this, we construct different values of $\tilde{p}_{\mathcal{D}_{\text{fix}}}$ with varying reuse gaps and measure the resulting performance gaps. We first approximate $q^\star_{\mathcal{D}_{\text{fix}}}$ (and $F(q^\star)$) by running full recomputation on $\mathcal{D}'$ using OLMIXBASE and normalizing the resulting mix over $\mathcal{D}_{\text{fix}}$. We then construct two $\tilde{p}_{\mathcal{D}_{\text{fix}}}$:

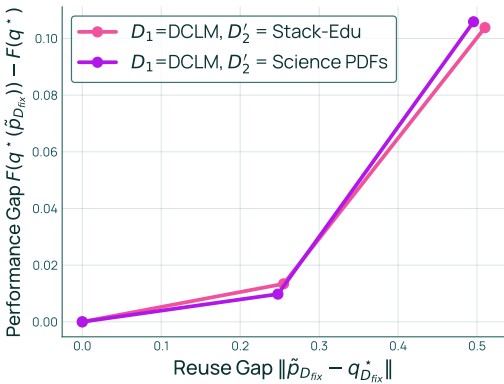

*Figure 22.* Performance vs Reuse Gap. The optimality of $\tilde{p}_{\mathcal{D}_{\text{fix}}}$ after the domain update is correlated with the success of mixture reuse.

- Weak mix: to construct a $\tilde{p}_{\mathcal{D}_{\text{fix}}}$ far from $q^{\star}_{\mathcal{D}_{\text{fix}}}$, we run OLMIXBASE on $\mathcal{D}'$ with a modified objective that maximizes BPB and then normalize the resulting mix over $\mathcal{D}_{\text{fix}}$.

- Intermediate mix: we average the weak mix with $q^{\star}_{\mathcal{D}_{\text{fix}}}$.

Figure 22 confirms the findings of Theorem 4.1: as the reuse gap increases, the performance gap also increases across both settings.

**The reuse gap is controlled by how much the domain update shifts the optimal mixture.** When adding domains, Theorem 4.2 shows that the reuse gap (and consequently the performance gap) depends on how much total weight shifts to the affected domains. Specifically, the reuse gap depends on $1 - \rho^{\star}$, where $\rho^{\star}$ is the total weight on the reused domains $\mathcal{D}_1$ in the optimal mix $q^{\star}$. We validate that $1 - \rho^{\star}$ predicts both the reuse gap and performance gap.

To do this, we construct settings with different $\rho^{\star}$ by varying the repetition constraint. A tight constraint (small $k$ or large $R$ in (2)) forces the mix to stay close to the natural distribution, yielding $\rho^{\star} \approx 1$ when $\mathcal{D}_1$ is much larger than the added domains. A relaxed constraint allows more weight to shift to new domains, potentially yielding smaller $\rho^{\star}$. We test two settings: $R = 1\text{T}$ (relaxed) and $R = 6\text{T}$ (tight). For each $R$, we 1) compute $\tilde{p}_{\mathcal{D}_{\text{fix}}}$ by running OLMIXBASE on $\mathcal{D}$; 2) apply FULLMIXTUREREUSE on $\mathcal{D}'$ using $\tilde{p}_{\mathcal{D}_{\text{fix}}}$; and 3) use full recomputation (run OLMIXBASE on $\mathcal{D}'$) to get an approximate $\rho^{\star}$, the reuse gap, and the performance gap.

Figure 23 shows that larger $1 - \rho^{\star}$ correlates with both a larger reuse gap and a larger performance gap across both settings, validating that the extent to which the domain update shifts the optimal mixture directly impacts mixture reuse performance.

**PARTIALMIXTUREREUSE reduces coupling and improves performance.** Theorems 4.1 and 4.2 show that the coupling term $\kappa$ controls the *rate* at which changes in optimality translate to the performance gap. The coupling term is large when both reused domains $\mathcal{D}_{\text{fix}}$ and recomputed domains $\mathcal{D}_{\text{comp}}$ impact similar downstream tasks. We validate that reducing coupling—by adjusting $\mathcal{D}_{\text{fix}}$ vs $\mathcal{D}_{\text{comp}}$ via PARTIALMIXTUREREUSE—improves both the reuse gap and performance gap.

To do this, we analyze a scenario where coupling is high: adding Stack-Edu to DCLM topics when $R = 1\text{T}$ (from Figure 23 left, purple). Figure 24 compares the reused ratios $\tilde{p}_{\mathcal{D}_{\text{fix}}}$ against the optimal ratios $q^{\star}_{\mathcal{D}_{\text{fix}}}$ over DCLM topics. There is a stark difference in the weight on DCLM's software development topic, suggesting high coupling between software development and Stack-Edu. Based on this, we test whether PARTIALMIXTUREREUSE with $\mathcal{D}_{\text{comp}} = \text{Stack-Edu} \cup \{\text{DCLM:software\_development}\}$ reduces coupling and improves performance compared to FULLMIXTUREREUSE (which uses $\mathcal{D}_{\text{comp}} = \text{Stack-Edu}$).

Figure 25 shows that recomputing DCLM:software\_development along with Stack-Edu reduces both the reuse gap and performance gap, despite $1 - \rho^{\star}$ being larger for PARTIALMIXTUREREUSE. This breaks the pattern seen in Figure 23: larger $1 - \rho^{\star}$ no longer translates to a larger performance gap, suggesting that the reduction in the coupling term $\kappa$ dominates.

To confirm that lower $\kappa$ is responsible for PARTIALMIXTUREREUSE's performance, we estimate $\kappa$ (see Definition E.1) for FULLMIXTUREREUSE and for PARTIALMIXTUREREUSE with $\mathcal{D}_{\text{comp}} = \text{Stack-Edu} \cup \{\text{DCLM:topic}\}$ for each DCLM topic. Operationally, we compute this using the fitted log-linear regression coefficients produced from a swarm over $\mathcal{D}'$. We split the coefficients into those associated with $\mathcal{D}_{\text{fix}}$ and $\mathcal{D}_{\text{comp}}$ and use their norms to obtain $\kappa(\mathcal{D}_{\text{fix}}, \mathcal{D}_{\text{comp}})$.

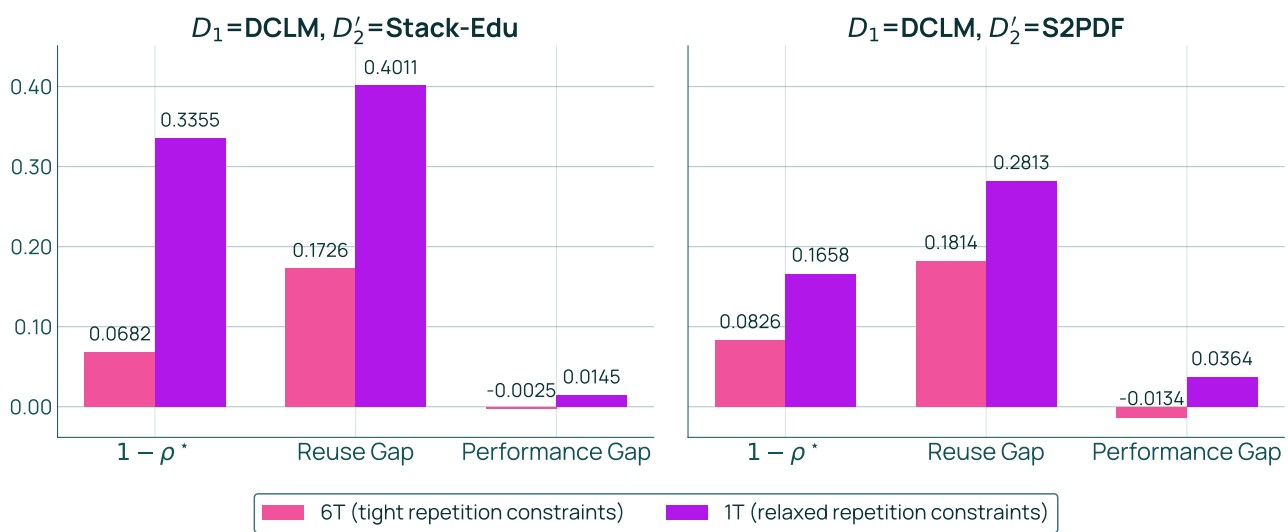

*Figure 23.* Performance gap vs reuse gap vs $1-\rho^{\star}$ when adding Stack-Edu to DCLM (left) and olmOCR Science PDFs to DCLM (right). Varying $R$ changes $\rho^{\star}$, and we observe that $1-\rho^{\star}$ propagates to both the reuse gap and performance gap, validating Theorem 4.2.

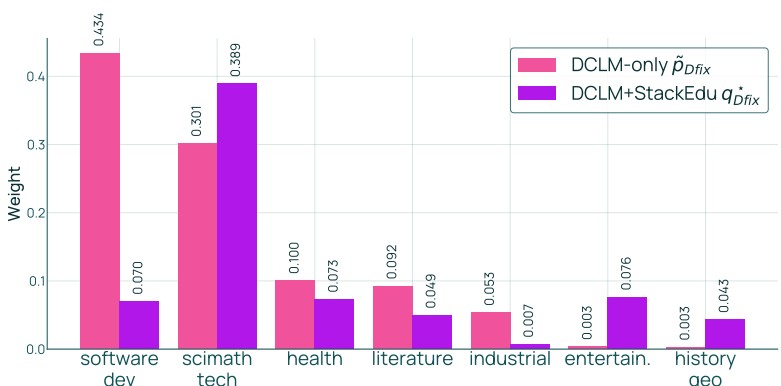

*Figure 24.* Comparison of $\tilde{p}_{\mathcal{D}_{\text{fix}}}$ versus $q^{\star}_{\mathcal{D}_{\text{fix}}}$ (top-7 domains according to $\tilde{p}_{\mathcal{D}_{\text{fix}}}$) when $\mathcal{D}_1 =$DCLM, $\mathcal{D}'_2=$Stack-Edu with $R=1$T. The only domain that significantly differs in mixture weight is software development, suggesting that PARTIALMIXTUREREUSE can reduce coupling if we recompute DCLM:software_development.

Figure 26 shows that recomputing DCLM:software_development reduces $\kappa$ far more than recomputing any other DCLM topic, validating that PARTIALMIXTUREREUSE reduces coupling.

## F. Experiment Details

**Domains**   We provide a breakdown of the number of tokens per each domain in Table 6 and a description of each domain update in our simulated LM development workflow in Table 7.

**Model**   We train 1B parameter decoder-only transformer models using Olmo 2 architecture (OLMo et al., 2024). In particular, we set n_layers=16, n_head=16, d_model=2048, head_dim=128.

We use a batch size of 512, a learning rate of 0.0018 with a cosine scheduler with warmup and linear decay, and a sequence length of 4096. We use the Dolma 2 tokenizer.

All 1B models were trained on 32 NVIDIA H100s (80 GB), while proxy models were trained on one NVIDIA H100.

**Evaluation**   Table 8 lists all the downstream tasks.

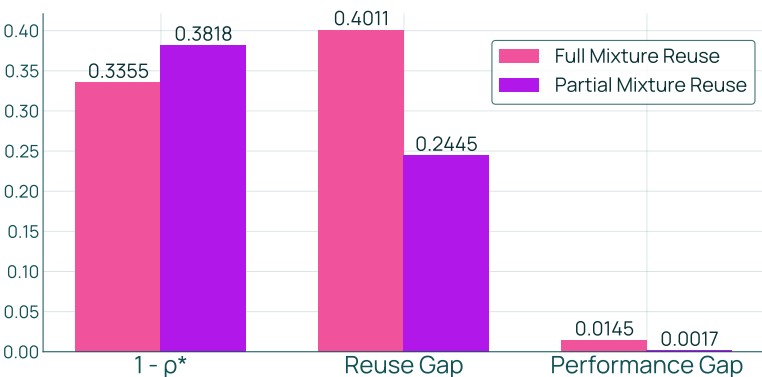

*Figure 25.* Performance of PARTIALMIXTUREREUSE (recompute DCLM:software_development) for $\mathcal{D}_1$=DCLM, $\mathcal{D}'_2$=Stack-Edu with $R=1$T. Recomputing DCLM:software development along with Stack-Edu reduces both the reuse gap and the performance gap, even though its $1-\rho^\star$ is higher.

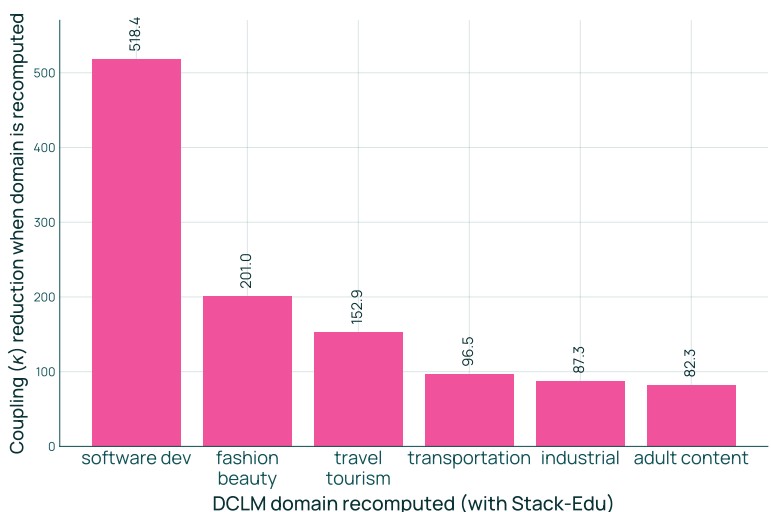

*Figure 26.* The top-6 DCLM domains for which recomputation results in the greatest reduction in $\kappa$ for $\mathcal{D}$=DCLM, $\mathcal{D}'$=DCLM+Stack-Edu. The reduction from recomputing DCLM:software development via PARTIALMIXTUREREUSE is substantially higher than recomputing other DCLM topics.

## F.1. Implementation Details

For FULLMIXTUREREUSE and PARTIALMIXTUREREUSE, the first $\tilde{p}$ is from applying OLMIXBASE on the initial web corpus. After each update, the newly recomputed mix becomes the $\tilde{p}$ of the next one.

Across all swarm-based methods we evaluate on (OLMIXBASE, Swarm Reuse, FULLMIXTUREREUSE, PARTIALMIXTUREREUSE), we use $S_{\text{small}} = 30$M parameters (architecture in Table 3) and train for 5x Chinchilla, with a batch size of 64, a learning rate of $0.007$, and a sequence length of $2048$.

For any $m$ domains we want to recompute, we set $K$ to approximate $c(m+1)$ for linear multiplier $c=1,2,3$. In particular, for $c=3$, we set $K=2^x$ to be the power of 2 that is closest to $3(m+1)$. For $c=2$, we set $K=2^{x-1}$. For $c=1$, we set $K=m+1$. When we recompute over $m=2$ domains, we use search instead of regression, as the search space is one-dimensional. See Tables 9, 10, 11, and 12 for exact swarm sizes. We solve all of our mixture optimization problems using CVXPY and a KL regularization term with $\lambda=0.05$. We set $k=4$ and $R=1$T for the repetition constraint of the experiments in §4.5.

Algorithm 3 corresponds to the Swarm Reuse approach in §4.5, presenting another way to reuse prior information from mixing. The blue coloring indicates aspects of this algorithm that are different from OLMIXBASE.

*Table 6.* Token counts for all domains in the final domain set. Sources with topic-level partitions (DCLM, Stack-Edu, olmOCR Science PDFs) are shown with indented topics. Single-source domains are shown without subdivision.

| Domain | Token Count | Domain | Token Count |
|---|---|---|---|
| **DCLM** (Li et al., 2024a) | | **Stack-Edu** (Allal et al., 2025) | |
| Adult Content | 67,760,078,203 | C | 4,735,074,247 |
| Art and Design | 70,659,711,995 | C# | 7,204,525,343 |
| Crime and Law | 170,130,914,779 | C++ | 12,530,445,761 |
| Education and Jobs | 184,690,792,861 | Go | 1,398,595,118 |
| Electronics and Hardware | 80,168,541,745 | Java | 31,347,725,888 |
| Entertainment | 441,768,061,760 | JavaScript | 8,886,972,357 |
| Fashion and Beauty | 37,256,539,512 | Markdown | 28,916,320,218 |
| Finance and Business | 310,313,927,581 | PHP | 7,395,033,318 |
| Food and Dining | 105,937,299,687 | Python | 18,017,832,560 |
| Games | 229,992,491,702 | Ruby | 1,386,775,805 |
| Health | 393,496,227,836 | Rust | 1,418,447,132 |
| History and Geography | 161,049,719,459 | SQL | 7,063,472,860 |
| Home and Hobbies | 126,910,777,314 | Shell | 2,542,637,875 |
| Industrial | 43,572,140,450 | Swift | 1,510,019,025 |
| Literature | 364,834,344,848 | TypeScript | 2,495,753,789 |
| Politics | 611,198,130,192 | | |
| Religion | 277,776,929,208 | **olmOCR Science PDFs** (Olmo et al., 2025) | |
| Science, Math and Technology | 427,131,054,341 | Adult | 303,073,226 |
| Social Life | 218,731,841,124 | Art and Design | 6,833,185,034 |
| Software | 108,039,380,021 | Crime and Law | 42,538,674,743 |
| Software Development | 223,384,974,282 | Education and Jobs | 138,127,926,093 |
| Sports and Fitness | 196,759,999,355 | Entertainment | 6,069,602,783 |
| Transportation | 90,793,306,202 | Fashion and Beauty | 557,917,820 |
| Travel and Tourism | 57,642,815,530 | Finance and Business | 61,150,044,703 |
| | | Food and Dining | 2,322,982,204 |
| **Single-Source Domains** | | Games | 2,486,095,532 |
| **AlgebraicStack** (Azerbayev et al., 2023) | 11,818,955,329 | Health | 108,215,933,374 |
| **ArXiv** (Azerbayev et al., 2023) | 20,773,846,846 | Home and Hobbies | 3,924,579,643 |
| **FineMath-3+** (Allal et al., 2025) | 34,057,973,953 | Industrial | 29,389,278,657 |
| **Pes2o** (Soldaini & Lo, 2023) | 58,552,461,187 | Literature | 31,886,391,090 |
| **Wikipedia** (Soldaini et al., 2024) | 10,067,758,073 | Politics | 39,234,116,889 |
| | | Religion | 24,729,732,953 |
| | | Science and Technology | 424,245,385,160 |
| | | Software | 9,146,853,216 |
| | | Software Development | 41,841,278,724 |
| | | Sports and Fitness | 5,450,913,796 |
| | | Transportation | 17,149,342,957 |
| | | Travel | 2,102,425,717 |

| Operation | Dataset(s) | $\Delta m$ |
|---|---|---|
| Initial | DCLM (Li et al., 2024a) partitioned into topical domains using WebOrganizer (Wettig et al., 2025)) | 24 |
| Add | Stack-Edu (Allal et al., 2025), partitioned by programming language | +15 |
| Add | ArXiv (Azerbayev et al., 2023), FineMath 3+ (Allal et al., 2025), olmOCR Science PDFs (Olmo et al., 2025), Dolma 1 Wikipedia (Soldaini et al., 2024), Algebraic-Stack (Azerbayev et al., 2023), pes2o (Soldaini & Lo, 2023) | +6 |
| Revise | olmOCR Science PDFs (Olmo et al., 2025) (reformatted tables and references) | 0 |
| Remove | AlgebraicStack (Azerbayev et al., 2023) | −1 |
| Partition | olmOCR Science PDFs (Olmo et al., 2025), partitioned into topical domains using WebOrganizer (Wettig et al., 2025) | +20 |

*Table 7.* Full specification of datasets used for simulated domain updates, corresponding to Table 1 in the main text.

*Table 8.* **Details of the BPB evaluation suite**. We use the OLMOBASEEVAL easy evaluation suite (Olmo et al., 2025), formatted as bits-per-byte (BPB) over gold continuations. [†] = few-shot examples are built-in the task; [α] = human-written few-shot examples. We treat subtasks as standalone tasks, except for MMLU, where we use averages over the four MMLU categories.

| | Task | Capability | # ICL | Metric | # Subtasks |
|---|---|---|---|---|---|
| *Math* | Minerva MATH (2022) | Math Gen | $4^{\alpha}$ | BPB | 7 |
| | *Subtasks: Algebra, Counting and Probability, Geometry, Intermediate Algebra* | | | | |
| | *Number Theory, Prealgebra, Precalculus* | | | | |
| *Code* | HumanEval (2021) | Code Gen | 3 | BPB | - |
| | MBPP (2021) | Code Gen | 3 | BPB | - |
| | MT MBPP (2022; 2025) | Code Gen | 3 | BPB | 17 |
| | *Subtasks: Bash, C, C++, C#, Go, Haskell, Java, JavaScript, MatLab, PHP* | | | | |
| | *Python, R, Ruby, Rust, Scala, Swift, TypeScript* | | | | |
| *QA* | ARC (2018) | Science QA | 5 | BPB | 2 |
| | *Subtasks: ARC-Easy, ARC-Challenge* | | | | |
| | MMLU STEM (2021) | General QA | 5 | BPB | 19 |
| | MMLU Humanities (2021) | General QA | 5 | BPB | 13 |
| | MMLU Social Sci. (2021) | General QA | 5 | BPB | 12 |
| | MMLU Other (2021) | General QA | 5 | BPB | 14 |
| | CSQA (2019) | Commonsense QA | 5 | BPB | - |
| | HellaSwag (2019) | Language Modeling | 5 | BPB | - |
| | WinoGrande (2020) | Language Modeling | 5 | BPB | - |
| | SocialIQA (2019) | Social QA | 5 | BPB | - |
| | PiQA (2020) | Physical QA | 5 | BPB | - |
| | CoQA (2019) | Conversation QA | $0^{\dagger}$ | BPB | - |
| | DROP (2019) | Passage QA | 5 | BPB | - |
| | Jeopardy (2024) | Trivia QA | 5 | BPB | - |
| | NaturalQs (2019) | General QA | 5 | BPB | - |
| | SQuAD (2016) | General QA | 5 | BPB | - |
| | SciQ (2017) | Science QA | 5 | BPB | - |
| | Basic Skills (2025) | Basic QA | 5 | BPB | 6 |
| | *Subtasks: Basic Arithmetic, String Manipulation, Simple Coding* | | | | |
| | *Elementary Logical Reasoning, Basic Common Sense, Simple Pattern Recognition* | | | | |
| | Lambada (2016) | Language Modeling | 0 | BPB | - |
| | MedMCQA (2022) | Medical QA | 5 | BPB | - |

*Table 9.* Swarm details per stage for **Full Recomputation**. We list the number of domains recomputed at each stage, and the columns with $c$ contain the number of proxy runs $K \approx c(m+1)$.

| Stage | # domains recomputed | $c=1$ | $c=2$ | $c=3$ |
|---|---|---|---|---|
| Initial | 24 | 25 | 32 | 64 |
| Add (Stack-Edu languages) | 39 | 40 | 64 | 128 |
| Add (ArXiv, FineMath 3+, PDFs, Wiki, AlgebraicStack, pes2o) | 45 | 46 | 64 | 128 |
| Revise (PDFs) | 45 | 46 | 64 | 128 |
| Remove (AlgebraicStack) | 44 | 45 | 64 | 128 |
| Partition (PDFs) | 64 | 65 | 128 | 256 |
| **Total # proxy runs** | | 267 | 416 | 832 |

*Table 10.* Swarm details per stage for **Swarm Reuse**. We list the number of domains recomputed at each stage, and the columns with $c$ contain the number of additional proxy runs needed, $K \approx c(m+1)$. $c=3$ corresponds to Figure 8.

| Stage | # domains recomputed | $c=1$ | $c=2$ | $c=3$ |
|---|---|---|---|---|
| Initial | 24 | 25 | 32 | 64 |
| Add (Stack-Edu languages) | 39 | 15 | 32 | 64 |
| Add (ArXiv, FineMath 3+, PDFs, Wiki, AlgebraicStack, pes2o) | 45 | 6 | 6 | 6 |
| Revise (PDFs) | 45 | 6 | 6 | 6 |
| Remove (AlgebraicStack) | 44 | 5 | 5 | 5 |
| Partition (PDFs) | 64 | 20 | 59 | 123 |
| **Total # proxy runs** | | 77 | 140 | 268 |

## F.2. Additional Results

**6T results.** In Figure 27, we compare the performance versus cost of mixing strategies when $R = 6$T, a more data-constrained setting. We exclude PARTIALMIXTUREREUSE since FULLMIXTUREREUSE performs well at $R = 6$T and only falls short

*Table 11.* Swarm details per stage for **Full Mixture Reuse**. We list the number of domains recomputed at each stage, and the columns with $c$ contain the number of proxy runs $K \approx c(m+1)$. $c=3$ corresponds to Figure 8.

| Stage | # domains recomputed | $c=1$ | $c=2$ | $c=3$ |
|---|---|---|---|---|
| Initial | 24 | 25 | 32 | 64 |
| `Add` (Stack-Edu languages) | 16 | 17 | 32 | 64 |
| `Add` (ArXiv, FineMath 3+, PDFs, Wiki, AlgebraicStack, pes2o) | 7 | 8 | 8 | 16 |
| `Revise` (PDFs) | 2 | 3 | 4 | 8 |
| `Remove` (AlgebraicStack) | 0 | 0 | 0 | 0 |
| `Partition` (PDFs) | 22 | 23 | 32 | 64 |
| **Total # proxy runs** | | 76 | 108 | 216 |

*Table 12.* Swarm details per stage for **Partial Mixture Reuse**. We list the number of domains recomputed at each stage, and the columns with $c$ contain the number of proxy runs $K \approx c(m+1)$. $c=3$ corresponds to Figure 8.

| Stage | # domains ($m$) | Proxy Runs ($c=3$) |
|---|---|---|
| Initial | 24 | 64 |
| `Add` (Stack-Edu languages) | 17 | 64 |
| `Add` (ArXiv, FineMath 3+, PDFs, Wiki, AlgebraicStack, pes2o) | 8 | 32 |
| `Revise` (PDFs) | 8 | 32 |
| `Remove` (AlgebraicStack) | 7 | 16 |
| `Partition` (PDFs) | 27 | 64 |
| **Total # proxy runs** | | 272 |

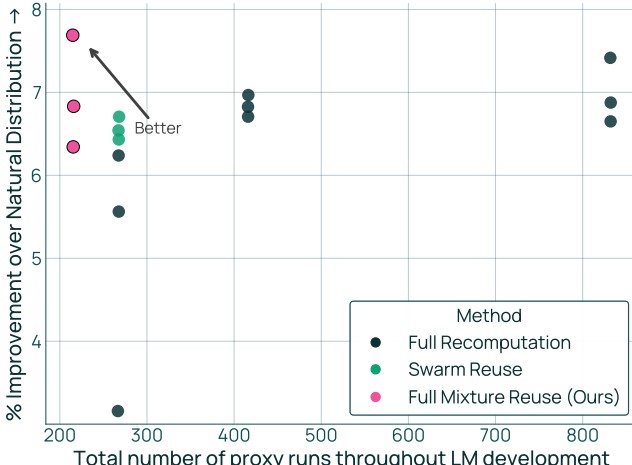

*Figure 27.* Performance improvement versus cost of mixing under evolving domains. FULLMIXTUREREUSE achieves 99% of the improvement of full recomputation while using 74% fewer proxy runs.

at $R=1$T (Figure 23).

We find that FULLMIXTUREREUSE achieves 99% of full recomputation's ($c=3$) performance improvement (+6.94% versus +6.97%) while using 74% fewer total proxy runs (216 versus 832 runs). Swarm reuse achieves 93% of full recomputation's performance with 52 more runs than FULLMIXTUREREUSE.

**Low proxy run budget regime.** While Figure 8 assesses performance versus number of proxy runs for cumulative costs ranging from 216 to 832 runs, it is unclear how recomputation strategies behave in a lower budget regime. We evaluate full recomputation with $c=1$ (i.e., the minimum swarm size possible) and all other methods with $K \approx c(m+1)$ for $c=1,2,3$, such that the range of total proxy runs considered is from 76 to 272.

Figure 28 shows performance versus number of proxy runs in a low budget regime. We make three observations:

- The performance of swarm reuse versus FULLMIXTUREREUSE at low budgets is similar.

---

**Algorithm 3** Swarm Reuse Mixing

---

1: **Input:** Old domain set $\mathcal{D}$, new domain set $\mathcal{D}'$, old swarm $\{(p_{\text{old}}^j, \{y_{ij}^{\text{old}}\}_{i=1}^n)\}_{j=1}^{K_{\text{old}}}$, new swarm size $K_{\text{new}} = \mathcal{O}(|\mathcal{D}'|)$, Dirichlet distribution $\mathcal{P}'$, repetition factor $k$, requested tokens $R$.

2: Sample new mixes $p_{\text{new}}^1, \dots, p_{\text{new}}^{K_{\text{new}}} \sim \mathcal{P}'$ on $\mathcal{D}'$.

3: Train proxy models on new mixes and evaluate on downstream tasks to get new swarm data $\{(p_{\text{new}}^j, \{y_{ij}^{\text{new}}\}_{i=1}^n)\}_{j=1}^{K_{\text{new}}}$.

4: Map old swarm mixes to new domain set: for $j \in [K_{\text{old}}]$, compute $\tilde{p}_{\text{old}}^j \in \triangle^{m'-1}$ from $p_{\text{old}}^j \in \triangle^{m-1}$ where:

   - If **Add**: $\mathcal{D} = [\mathcal{D}_1, \emptyset]$ and $\mathcal{D}' = [\mathcal{D}_1, \mathcal{D}_2']$. Then $\tilde{p}_{\text{old}}^j(D_i') = p_{\text{old}}^j(D_i)$ for $D_i \in \mathcal{D}_1$, and $\tilde{p}_{\text{old}}^j(D_i') = 0$ for $D_i' \in \mathcal{D}_2'$.

   - If **Partition**: $\mathcal{D} = [\mathcal{D}_1, \{D_{\text{par}}\}]$ and $\mathcal{D}' = [\mathcal{D}_1, \{D_{\text{par}}^1, \dots, D_{\text{par}}^\ell\}]$ where $D_{\text{par}} = \bigcup_{k=1}^\ell D_{\text{par}}^k$. Then $\tilde{p}_{\text{old}}^j(D_i') = p_{\text{old}}^j(D_i)$ for $D_i \in \mathcal{D}_1$, and $\tilde{p}_{\text{old}}^j(D_{\text{par}}^k) = p_{\text{old}}^j(D_{\text{par}}) \cdot \frac{N_{D_{\text{par}}^k}}{\sum_{k'=1}^\ell N_{D_{\text{par}}^{k'}}}$ for the partitioned domains.

5: Construct combined swarm: $\mathcal{S} = \{(\tilde{p}_{\text{old}}^j, \{y_{ij}^{\text{old}}\})\}_{j=1}^{K_{\text{old}}} \cup \{(p_{\text{new}}^j, \{y_{ij}^{\text{new}}\})\}_{j=1}^{K_{\text{new}}}$.

6: **for** $i \in [n]$ **do**

7:    Use combined swarm $\mathcal{S}$ to fit the log-linear model $\hat{f}_i(p) = c_i + \exp(A_i^\top p)$, where $c_i \in \mathbb{R}^+$ and $A_i \in \mathbb{R}^{|\mathcal{D}'|}$.

8: **end for**

9: Solve the optimization problem:

$$\text{minimize}_{p \in \triangle^{|\mathcal{D}'|-1}} \frac{1}{n} \sum_{i=1}^n \hat{f}_i(p) \tag{17}$$

$$\text{subject to} \quad p_j \leq \frac{k N_j'}{R} \ \forall j \in |\mathcal{D}'|$$

10: **Return** $p^\star$, the solution on new domain set $\mathcal{D}'$.

---

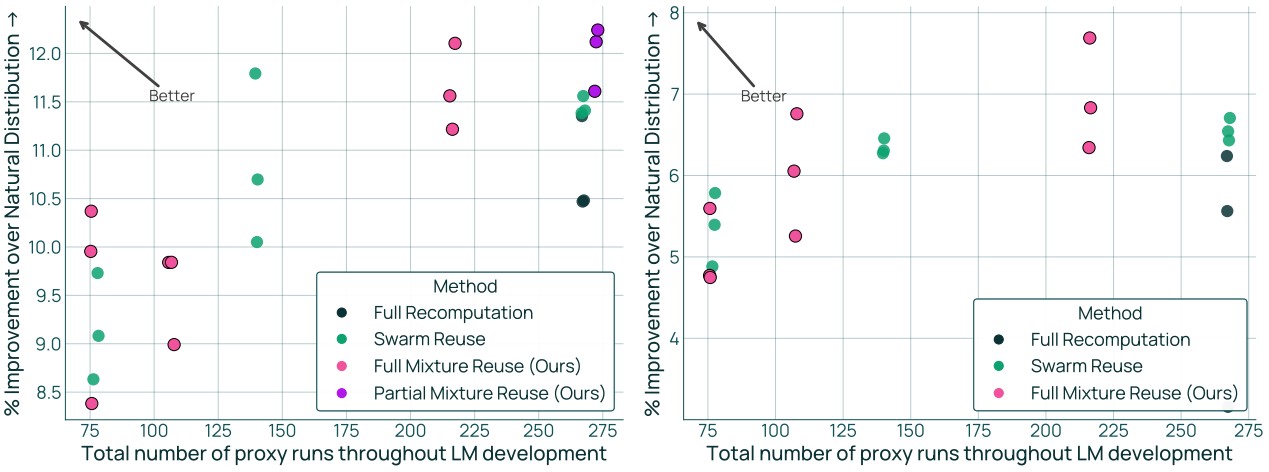

*Figure 28.* Performance improvement versus cost of mixing under evolving domains for $R = 1\text{T}$ (left) and $R = 6\text{T}$ (right). Total number of proxy runs vary from 76 to 272.

- While the minimum number of proxy runs needed for full recomputation is 267, mixture reuse and swarm reuse unlock a lower budget regime, requiring as little as 76 runs. These reuse strategies can be attractive to practitioners with strict compute budgets and large domain sets that undergo many updates.

- With only 76 runs, mixture reuse still achieves a $+9.6\%$ and $+5\%$ improvement over the natural distribution when $R = 1\text{T}$ and $6\text{T}$, respectively.

**Proposed mixes.** Table 13 shows the full mixtures corresponding to the natural distribution, full recomputation (best seed), and PARTIALMIXTUREREUSE (best seed).

*Table 13.* Mixture weights (full version of Figure 10) for the natural distribution, full recomputation, and PARTIALMIXTUREREUSE.

| Domain | Nat. | Full | Partial | Domain | Nat. | Full | Partial |
|---|---|---|---|---|---|---|---|
| arxiv | 0.003320 | 0.036555 | 0.022549 | pdf:fashion_beauty | 0.000089 | 0.000003 | 0.000043 |
| dclm:adult_content | 0.010828 | 0.006309 | 0.004050 | pdf:finance_business | 0.009771 | 0.004230 | 0.001101 |
| dclm:art_and_design | 0.011291 | 0.006265 | 0.008097 | pdf:food_dining | 0.000371 | 0.000113 | 0.009292 |
| dclm:crime_and_law | 0.027186 | 0.009991 | 0.005497 | pdf:games | 0.000397 | 0.000074 | 0.009944 |
| dclm:education_and_jobs | 0.029513 | 0.019003 | 0.008874 | pdf:health | 0.017292 | 0.013871 | 0.006251 |
| dclm:electronics_and_hardware | 0.012811 | 0.003289 | 0.003360 | pdf:home_hobbies | 0.000627 | 0.000396 | 0.015698 |
| dclm:entertainment | 0.070592 | 0.055739 | 0.021609 | pdf:industrial | 0.004696 | 0.002512 | 0.000749 |
| dclm:fashion_and_beauty | 0.005953 | 0.003134 | 0.000769 | pdf:literature | 0.005095 | 0.005713 | 0.007648 |
| dclm:finance_and_business | 0.049587 | 0.022324 | 0.009896 | pdf:politics | 0.006269 | 0.001782 | 0.038680 |
| dclm:food_and_dining | 0.016928 | 0.015615 | 0.003785 | pdf:religion | 0.003952 | 0.001056 | 0.000088 |
| dclm:games | 0.036752 | 0.018811 | 0.013172 | pdf:science_tech | 0.067792 | 0.029994 | 0.014598 |
| dclm:health | 0.062879 | 0.071471 | 0.050413 | pdf:software | 0.001462 | 0.001752 | 0.000000 |
| dclm:history_and_geography | 0.025735 | 0.028328 | 0.008994 | pdf:software_dev | 0.006686 | 0.001929 | 0.015815 |
| dclm:home_and_hobbies | 0.020280 | 0.017861 | 0.005614 | pdf:sports_fitness | 0.000871 | 0.000105 | 0.000060 |
| dclm:industrial | 0.006963 | 0.004541 | 0.007756 | pdf:transportation | 0.002740 | 0.001798 | 0.068597 |
| dclm:literature | 0.058299 | 0.048124 | 0.026746 | pdf:travel | 0.000336 | 0.000043 | 0.007881 |
| dclm:politics | 0.097667 | 0.034541 | 0.011652 | stack-edu:C | 0.000757 | 0.004088 | 0.001655 |
| dclm:religion | 0.044387 | 0.018381 | 0.006006 | stack-edu:CSharp | 0.001151 | 0.014910 | 0.015118 |
| dclm:science_math_and_technology | 0.068254 | 0.077043 | 0.088744 | stack-edu:Cpp | 0.002002 | 0.012993 | 0.036761 |
| dclm:social_life | 0.034952 | 0.018543 | 0.011095 | stack-edu:Go | 0.000223 | 0.003558 | 0.004103 |
| dclm:software | 0.017264 | 0.003887 | 0.002666 | stack-edu:Java | 0.005009 | 0.015251 | 0.025259 |
| dclm:software_development | 0.035696 | 0.018979 | 0.025232 | stack-edu:JavaScript | 0.001420 | 0.004582 | 0.026072 |
| dclm:sports_and_fitness | 0.031441 | 0.017050 | 0.006516 | stack-edu:Markdown | 0.004621 | 0.028525 | 0.084832 |
| dclm:transportation | 0.014508 | 0.006398 | 0.003515 | stack-edu:PHP | 0.001182 | 0.003491 | 0.007554 |
| dclm:travel_and_tourism | 0.009211 | 0.005433 | 0.001804 | stack-edu:Python | 0.002879 | 0.072071 | 0.052859 |
| finemath-3plus | 0.005442 | 0.136232 | 0.136232 | stack-edu:Ruby | 0.000222 | 0.000868 | 0.004068 |
| pes2o | 0.009356 | 0.012147 | 0.006478 | stack-edu:Rust | 0.000227 | 0.000008 | 0.004161 |
| pdf:adult | 0.000048 | 0.000907 | 0.000016 | stack-edu:SQL | 0.001129 | 0.001833 | 0.002143 |
| pdf:art_design | 0.001092 | 0.000231 | 0.004289 | stack-edu:Shell | 0.000406 | 0.008734 | 0.007459 |
| pdf:crime_law | 0.006797 | 0.002672 | 0.000038 | stack-edu:Swift | 0.000241 | 0.006040 | 0.004430 |
| pdf:education_jobs | 0.022072 | 0.010150 | 0.005608 | stack-edu:TypeScript | 0.000399 | 0.009983 | 0.001015 |
| pdf:entertainment | 0.000970 | 0.000087 | 0.003318 | wikipedia | 0.001609 | 0.017654 | 0.011671 |

**Ablations for each domain update operator.** We demonstrate that FULLMIXTUREREUSE attains performance similar to full recomputation for each type of domain update operator. We evaluate the following domain updates:

- `Add`: we start with 24 DCLM topic-based domains. Then, we add 15 Stack-Edu programming language domains.

- `Remove`: we start with 39 domains consisting of 24 DCLM topics and 15 Stack-Edu programming languages. Then, we remove the 15 Stack-Edu domains.

- `Partition`: we start with 24 DCLM topics and a single Stack-Edu source domain. Then, we partition the Stack-Edu source into 15 programming language domains.

- `Revise`: we start with 24 DCLM topics and a single StackV2 (Lozhkov et al., 2024) source domain. Then, the StackV2 domain is replaced with the Stack-Edu domain.

For each operator, we compare FULLMIXTUREREUSE and the natural distribution over $\mathcal{D}'$ (as a control) against full recomputation. We measure the total variation distance of the proposed mixtures with respect to full recomputation's mix, defined as $\mathrm{TV}(q, q^\star) = \frac{1}{2} \sum_{i=1}^{m'} |q_i - q_i^\star| \in [0,1]$. We also measure performance gap relative to full recomputation's downstream performance (average task BPB) for `Add`, `Remove`, and `Partition`. For our setup, we used $k = 4$, $R = 6$T.

Figure 29 shows that FULLMIXTUREREUSE achieves substantially smaller performance gaps and TV distances compared to the natural distribution across `Add`, `Remove`, and `Partition`. For `Revise`, the TV distance between FULLMIXTUREREUSE and full recomputation was only 0.21%—orders of magnitude smaller than other operators—indicating nearly identical mixes. We therefore did not evaluate downstream performance for `Revise`, as variation would be dominated by noise. These results demonstrate that mixture reuse works effectively across all domain update operators.

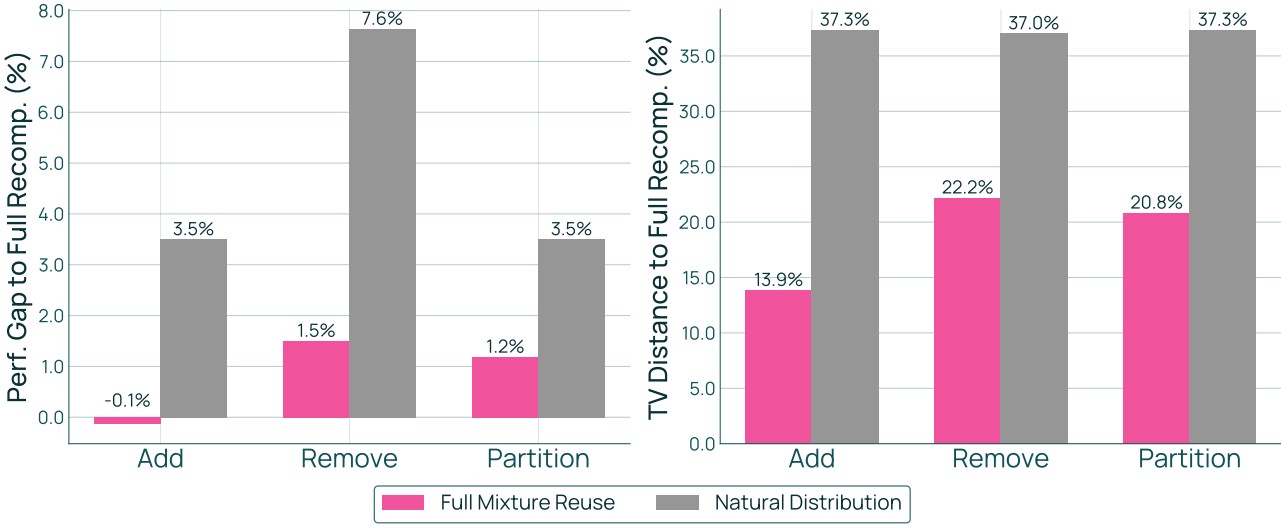

*Figure 29.* Relative performance gap (left) and total variation distance (right) w.r.t. full recomputation across domain update operators. FULLMIXTUREREUSE is substantially closer to full recomputation than the natural distribution across all three operators, in both performance and mixes.

