# OpenReview forum: "Olmix: A Framework for Data Mixing Throughout LM Development"
_ICML.cc/2026/Conference — ICML 2026 regular_

### Official Review · Reviewer_aGzz · 2026-03-03

**Soundness:** 3
**Presentation:** 3
**Significance:** 3
**Originality:** 3
**Overall Recommendation:** 4
**Confidence:** 4

**Summary:**

This paper introduces the OLMIX framework to address offline configuration optimization and evolving domain collections in data mixture for language model pretraining. Through ablation studies, optimal configurations are identified (proxy models $>15$M parameters, log-linear regression), and the Mixture Reuse mechanism is proposed to reuse historical mixture ratios during domain updates, achieving 74% computational savings while maintaining 95% performance and improving 11.6% over uniform mixture in experiments with 1B models across 5 domain updates.

**Compliance With Llm Reviewing Policy:**

Affirmed.

**Final Justification:**

The authors have addressed my concerns regarding repetition constraints and the scalability of the proxy models; I maintain my positive rating and recommend acceptance.

**Key Questions For Authors:**

1. What is the selection strategy for $D_{\text{partial}}$ in PartialMixtureReuse? Please provide a complete algorithmic description (including hyperparameters) and clarify whether there are heuristic rules (e.g., based on domain change magnitude).
2. What is the trade-off between different $D_{\text{partial}}$ selection strategies in terms of final performance and computational cost? Please provide sensitivity ablations across the 5 domain update cases.
3. What is the impact of different repetition factors $k$ on final training performance and overfitting? When the repetition upper bound is reached, does the optimizer overfit to that domain, leading to performance degradation in other domains?

**Limitations:**

yes

**Strengths And Weaknesses:**

**Strengths**

1. The paper introduces the evolving domain problem setting, addressing a gap in existing data mixture research for dynamic domain scenarios with strong practical significance.
2. The Mixture Reuse design is elegant and concise: aggregating unchanged domains into a virtual domain reduces optimization dimensionality and is supported by theoretical guarantees under the log-linear assumption.
3. The offline configuration study covers key design dimensions: ablation experiments on proxy model size, proxy runs, and regression model families yield clear and actionable conclusions (proxy models $>15$M parameters, proxy runs scaling linearly with domain count $m$, log-linear regression).
4. The end-to-end case study closely mirrors real-world development scenarios, covering diverse operations such as adding, removing, modifying, and splitting domains, with significant computational savings.
5. The integrated design of repetition constraints considers data-limited scenarios, addressing practical data budget issues in real-world development.

**Weaknesses**

1. The repetition constraint experiments (RQ4) only demonstrate the impact of different $k$ values on the proposed mixture, without reporting corresponding downstream task performance. This makes it impossible to determine whether the optimized repetition constraints genuinely improve model quality or merely satisfy the constraints.
2. The target model scale is limited to 1B models and 100B tokens. It is recommended to discuss the method's scalability or expected performance at larger scales.
3. The algorithmic description of PartialMixtureReuse is incomplete, particularly lacking clear algorithmic procedures and hyperparameter specifications for the $D_{\text{partial}}$ selection strategy, which may affect the reproducibility of the method.

---

> ### Author Rebuttal · Authors · 2026-03-31
>
> Thank you for your detailed review! We are glad you found our work to be well-grounded theoretically and mirror real-world model development. We address your concerns below.
>
> **W1/Q3 Repetition Constraint Experiments** Thank you for the question. Our contribution in RQ6 is not to establish that excessive repetition degrades performance, but instead to show how this concern can be incorporated into the mixture optimization problem itself. The importance of limiting repetition is well-documented in prior work: for instance [1] shows that beyond ~4 epochs of repetition, returns diminish significantly, and [2] demonstrates that deduplication consistently improves downstream performance.
>
> Regarding overfitting to domains, this concern extends beyond repetition constraints — it applies to any mixing method that identifies a high-utility domain. The offline schema can assign a large weight to such a domain, improving performance on its corresponding evals at the potential expense of others, though the average BPB objective ensures this tradeoff is net positive. Repetition constraints are one mechanism to prevent this: by capping how many tokens can be drawn from any single domain, they implicitly bound domain weights. More targeted approaches (e.g., Pareto constraints or per-domain weight caps) are complementary and worth exploring in future work.
>
> [1] https://arxiv.org/abs/2305.16264
> [2] https://arxiv.org/abs/2107.06499
>
> **W2 Target Model Size** We agree that behaviors at larger scales could be different. Unfortunately, measuring rank correlation requires training many models (we trained 40 proxy-target pairs). For 7B models with 5x Chinchilla scaling, this requires over 350K H100-hours, which is computationally challenging for us to complete for the rebuttal.
>
> However, we do believe that for larger target models, using proportionally larger proxy models could suffice. In our response to G78a ("Q3 Scaling the Rank Correlation"), we observe that scaling law literature [3] demonstrates that the relative ratio of proxy to target model—not the absolute sizes—roughly dictates the prediction error. Furthermore, recent works like [4] demonstrate that the mixture optimized for a 7B MoE also performs well on a 55B MoE, suggesting that proxy-to-target transfer still holds on larger models.
>
> [3] https://arxiv.org/abs/2410.11840
> [4] https://arxiv.org/abs/2603.08022
>
> **W3/Q1/Q2 Partial Mixture Reuse**: We thank the reviewer for this question. In the paper, the selection strategy for $\mathcal{D}\_{\text{partial}}$ (in Section 4.5 and Appendix F.3) is based on two heuristics:
>
> 1. **Recompute domains of the same type as the update.** When adding Stack-Edu (code) to DCLM, we recomputed DCLM:software_development, and validated this post-hoc by confirming its $\kappa$ was substantially higher than other DCLM domains (Appendix F.3).
> 2. **Recompute domains that are "maxed out."** If a domain's current allocation exhausts all available tokens, freezing its ratio during reuse leaves high-utility data unused. We observed this in Section 4.5 and therefore always recomputed source-level ratios after each update.
>
> These were the only two heuristics we tried — we did not sweep over $\mathcal{D}\_{\text{partial}}$ configurations and select the best. Once $\mathcal{D}\_{\text{partial}}$ is selected, we apply Algorithm 2 with $\mathcal{D}\_{\text{fix}} = \mathcal{D}\_{\text{partial}}$, $\mathcal{D}\_{\text{comp}} = \mathcal{D}\_1 \backslash \mathcal{D}\_{\text{partial}}  \cup \mathcal{D}\_2’$. There are no additional hyperparameters beyond the choice of $\mathcal{D}\_{\text{partial}}$ itself.
>
> More broadly, we introduced PartialMixtureReuse to show that when full mixture reuse falls short, practitioners need not fall back to full recomputation — and that even simple choices of $\mathcal{D}\_{\text{partial}}$ can recover near-full-recomputation performance at reduced cost. The contribution is demonstrating that this spectrum exists and is navigable, not proposing a fully specified algorithm with tunable hyperparameters.
>
> Regarding sensitivity ablations across different $\mathcal{D}\_{\text{partial}}$ strategies: we strongly agree this would be informative. To run this experiment, 272 proxy models are required across the 5 updates for each of the three partial mixture reuse points in Figure 6, requiring a total of at least ~600 H100 GPU hours that cannot be parallelized since the 5 updates are sequential. This makes the experiment you suggest challenging to complete within the rebuttal period, but we will aim to incorporate this into the final paper.

---

> > ### Author Rebuttal · Reviewer_aGzz · 2026-04-04
> >
> > The authors have satisfactorily addressed my concerns in the rebuttal, and I will maintain my current positive rating to recommend the paper for acceptance.

---

### Official Review · Reviewer_KoZP · 2026-03-08

**Soundness:** 3
**Presentation:** 4
**Significance:** 3
**Originality:** 3
**Overall Recommendation:** 4
**Confidence:** 4

**Summary:**

The paper studies data mixing in language model (LM) pre-training, focusing on two key challenges: (1) the lack of clear guidance for configuring offline mixing methods (e.g., proxy model size and regression choice), and (2) the high computational cost of recomputing mixtures when the domain set evolves (e.g., adding or removing datasets). To address the first issue, the authors conduct a systematic empirical study and propose OlmixBase, a robust configuration for offline mixing. They show that proxy models should be larger than previously assumed (>15M parameters), swarm complexity scales linearly with the number of domains, and log-linear regression performs best. To address the second issue, they introduce Mixture Reuse, which reuses mixing ratios from unaffected domains when the domain set changes. They propose FullMixtureReuse and PartialMixtureReuse, achieving a 74% reduction in computational cost while maintaining performance close to full re-computation and significantly outperforming natural data distributions.

**Compliance With Llm Reviewing Policy:**

Affirmed.

**Key Questions For Authors:**

1. In Figure 4, LightGBM's fit has not plateaued, suggesting it might outperform the log-linear model if provided with more training data (larger swarm sizes). Have the authors explored the performance limit of LightGBM with more substantial proxy data?

2. Why was Pearson correlation chosen to evaluate regression fit? Did the authors evaluate the regression models using rank correlation metrics (e.g., Spearman's rank), and do the conclusions regarding the optimal regression family remain the same?

3. I noticed that the evaluation is reported exclusively in BPB. Do the "strong" mixtures identified by Olmix also outperform the baselines when measured by standard accuracy?

**Limitations:**

yes

**Strengths And Weaknesses:**

Strengths
1. The paper conducts systematic and thorough empirical studies to investigate key design choices in data mixing strategies.
2. It addresses a problem of significant industrial relevance—the evolving nature of datasets. By studying how to handle dataset updates (Add, Remove, Partition, Revise), the work offers high practical value for iterative model development and real-world deployment.
3. The proposed FullMixtureReuse method achieves impressive efficiency gains, requiring 74% fewer proxy runs while capturing 95% of the performance gains of full re-computation.

Weaknesses
1. The validation is performed on 1B parameter models. While the authors show rank correlation holds between 15M+ proxy models and 1B targets, there is always a risk that mixing behave differently at much larger scales (e.g., 7B), particularly regarding the emergence of specific capabilities.
2. While PartialMixtureReuse is effective, the selection of which domains to recompute vs. reuse relies on intuition or manual selection (e.g., knowing that "software development" is coupled with "code"). A more automated method to detect domain coupling would strengthen the contribution.
3. The optimization and evaluation rely primarily on Bits-Per-Byte (BPB) on downstream tasks. While this is a reasonable proxy for capability, it is not a perfect measure.

---

> ### Author Rebuttal · Authors · 2026-03-31
>
> Thank you for your positive feedback! We are glad you found the problem of evolving domains to be of significant industrial relevance and the empirical study to be systematic and thorough. We address your concerns below.
>
> **W1 Target Model Size** We agree that behaviors at larger scales could be different. Unfortunately, measuring rank correlation requires training many models (we trained 40 proxy-target pairs). For 7B models with 5x Chinchilla scaling, this requires over 350K H100-hours, which is computationally challenging for us to complete for the rebuttal.
>
> However, we do believe that for larger target models, using proportionally larger proxy models could suffice. In our response to G78a(Q3 Scaling the Rank Correlation), the scaling law literature [1] demonstrates that the relative ratio of proxy to target model—not the absolute sizes—roughly dictates the prediction error. Furthermore, recent works like [2] demonstrate that the mixture optimized for a 7B MoE also performs well on a 55B MoE, suggesting that transfer still holds on larger models.
>
> [1] https://arxiv.org/abs/2410.11840
> [2] https://arxiv.org/abs/2603.08022
>
> **W2 Partial Mixture Reuse** We agree that automating domain coupling detection is a valuable direction. However, we want to clarify that the intent of the partial mixture reuse contribution is not an automated selection method for $\mathcal{D}\_{\text{partial}}$; instead, it shows that when full mixture reuse falls short, practitioners need not fall back to full recomputation — there exist choices of $\mathcal{D}\_{\text{partial}}$ that recover near-full-recomputation performance at reduced cost (e.g., Figure 23). The coupling term $\kappa$ theoretically explains when and why this works.
>
> That said, we believe the current manual selection is stronger than it may appear. We set $\mathcal{D}\_{\text{partial}}$ using two simple heuristics:
> 1. **Recompute domains of the same type as the update.** When adding Stack-Edu (code) to DCLM, we recomputed DCLM:software_development, and validated this post-hoc by confirming its $\kappa$ was substantially higher than other DCLM domains (Appendix F.3).
> 2. **Recompute domains that are "maxed out."** If a domain's current allocation exhausts all available tokens, freezing its ratio during reuse leaves high-utility data unused. We observed this in Section 4.5 and therefore always recomputed source-level ratios after each update.
>
> These were the only two heuristics we tried — we did not sweep over $\mathcal{D}\_{\text{partial}}$ configurations and select the best. Moreover, practitioner intuition playing a role in setting things is not unlike other areas of data curation, where frameworks are principled but their instantiation is inherently domain- and evaluation-specific. We view developing more automated guidelines as an important direction for future work.
>
> **W3/Q3 BPB vs Accuracy**: We ran accuracy-based evaluations to validate that BPB tracks downstream performance. The table below compares BPB (lower is better) with accuracy metrics across all three task families (as in Table 8). For math and code, accuracy is measured as pass@16. For QA, accuracy is measured as the rank classification (RC) score. We compare the natural distribution versus full mixture reuse (“ours”) from Section 4.5.
> | Category | Metric | Natural | Ours |
> |-|-|-|-|
> | Math | BPB ↓ | 0.679 | 0.561 |
> | Math | Accuracy ↑ | 13.8 | 21.3 |
> | Code | BPB ↓ | 0.554 | 0.458 |
> | Code | Accuracy ↑ | 12.1 | 26.1 |
> | QA | BPB ↓ | 0.943 | 0.976 |
> | QA | Accuracy ↑ | 54.9 | 52.7 |
>
> BPB and accuracy move in the same direction: improvements in BPB correspond to gains in accuracy (math, code), and the modest QA regression is reflected in both metrics.
>
> **Q1 LightGBM Behavior**: Figure 15 (top right) presents regression fit versus swarm size for m=6 domains from DCLM. Since sample complexity is relative to the number of domains, we can see that for m=6, the regression fit for lightGBM plateaus to roughly the same performance at log-linear at K=63 (9(m+1)), and then slightly exceeds log-linear (by 0.01) once it reaches 105 (15(m+1)). We do not see lightGBM unlocking a fundamentally better regression fit in the limit; moreover, log-linear attains the same performance in 3x fewer proxy models.
>
> **Q2 Spearman Correlation**: Thank you for the suggestion. We ran the experiment corresponding to Figure 4 and reported Spearman correlation in [this plot](https://imgur.com/QBIkSve). Similar trends hold, with log-linear consistently outperforming other models after K=50, BiMix performing the best at K=25, and LightGBM requiring more samples.
>
> While the trends for Spearman and Pearson correlation are similar, we chose to use Pearson correlation because our downstream use of the regression model is to minimize the average predicted BPB. This requires the regression model to capture the magnitude of performance differences between mixtures, not just their rankings.

---

> > ### Author Rebuttal · Reviewer_KoZP · 2026-04-03
> >
> > The issues have now been satisfactorily resolved. I believe this is a technically strong paper that makes a meaningful contribution to data mixing. Therefore, I intend to maintain my current positive rating.

---

### Official Review · Reviewer_yGmN · 2026-03-11

**Soundness:** 4
**Presentation:** 3
**Significance:** 4
**Originality:** 3
**Overall Recommendation:** 5
**Confidence:** 4

**Summary:**

This paper conducts a thorough empirical study of the offline function-fitting style data mixing approaches, identifying good design choices  including proxy model size, swarm size, sampling methods for mixtures for training proxy models, fitting methods etc. Building on this comprehensive empirical study, they propose Olmix-base, which is a strong baseline for offline data mixing for LM training. On top of Olmix, the paper also sets up the problem of evolving domains through LM development, and proposes two simple extensions backed by theory---full mixture reuse and partial mixture reuse. Empirical results show that the mixture reuse methods preserve upto 95% of the performance of full mixture recomputation with much lesser compute, firmly establishing the Olmix-base + mixture reuse as a strong compute multiplier for LM pretraining.

**Compliance With Llm Reviewing Policy:**

Affirmed.

**Key Questions For Authors:**

- In the footnote in page 3, there is this point: ”Investigating the public RegMix code, we found their 1M implementation is closer to 15M, which our results suggest is a good proxy size” — what does this mean? Did they misreport in the paper or do you count parameters a different way (incl or excl embedding params)? This statement as written is quite confusing.
- Given the result in sec 3.2.3 that the swarm size scales linearly with m, would this suggest that for mixtures where the number of domains is very small, one should not use log-linear models (given that there is a sharp drop off in their performance at K=25)? While interesting, a more prescriptive takeaway from section 3.2.4 might be more appreciated by practitioners.
- Minor point: in fig 1, there is a Transform operation whereas it seems to be defined as Revise in sec 4.1. Would be good to make it consistent across both the figure and the section.
- Minor point: in figure 6, there seems to be a mismatch between the colouring of full recompilation in the legend and the actual plot markers (probably an alpha transparency issue).

**Limitations:**

Yes

**Strengths And Weaknesses:**

Strengths:

- The paper tackles an extremely important problem faced by data practitioners for foundation model training.
- The presentation of the data mixing problem makes for a very accessible reading of the paper even for folks not directly working on data mixing as a problem. The paper does a very convincing job in discussing the importance of the problem.
- The empirical study of various design choices is very thorough (so much so that some important aspects like RQ6 are even relegated to the appendix)
- The proposed approach for mixture reuse is well-grounded and motivated by theory; empirical results also clearly showcase the afforded benefits
- Table 2 is a work of art, very well done on this compilation, it will be of great use to practitioners!

Weaknesses:

- [Target model size] All the results in the paper are currently with just the 1B target model size. While I appreciate that running a full sweep for larger target model sizes might be infeasible, it is still important to validate if these methods scale to larger models. An important sub-implication of this comes up for the results of RQ1---could it be plausible that as the target model increases in size, the proxy model also needs to get larger? i.e. is there a maximal-capacity gap that could exist between the proxy model and the target model? It is not necessary to conduct additional experiments, but it would be interesting to add some discussion points / hypotheses around this. This might also lead to interesting connections with the distillation literature where such an optimal capacity gap has also been observed between student and teacher models.

- [Important details missing or not easily accessible] While the paper is generally well presented, I found it difficult to find more details on some important aspects. For example: (i) it seems that both the performance gap and the reuse gap are bounded by the coupling term. How is that computed in practice? Can that be done efficiently without several proxy runs?, and (ii) In PartialMixtureReuse how do you select the \matchcal{D}_{\text{partial}}?

---

> ### Author Rebuttal · Authors · 2026-03-31
>
> Thank you for your thoughtful feedback! We are glad you found the paper accessible and well-motivated. We address your concerns below.
>
> **W1 Target model size** We hypothesize that the proxy model needs to get larger as the target model increases to provide sufficient signal. In our response to G78a (“Q3 Scaling the Rank Correlation”), we observe that scaling law literature [1] demonstrates that the relative ratio of proxy to target model—not the absolute sizes—roughly dictates the prediction error. This is also closely related to the distillation literature [2], like you mentioned.
>
> [1] https://arxiv.org/abs/2410.11840
>
> [2] https://arxiv.org/abs/1902.03393
>
>
> **W2 How to compute the coupling term** We thank the reviewer for this feedback and will improve the presentation of these details in the paper.
>
> (i) The coupling term $\kappa$ is defined in Equation 14 (Appendix E) in terms of $A \in \mathbb{R}^{n \times m'}$, the parameter of the log-linear regression fit over all of $\mathcal{D}'$. Computing $\kappa$ requires training a swarm over the full updated domain set, fitting the regression, and extracting the relevant entries of $A$ — i.e., the very proxy runs that reuse aims to avoid. It should therefore be viewed as a post-hoc theoretical explanation for when and why partial mixture reuse works, rather than a practical input to the method. We hypothesize that embedding-based similarity scores between domains could serve as a cheaper proxy, though such a measure would be imperfect.
>
> (ii) We select $\mathcal{D}\_{\text{partial}}$ based on practitioner knowledge rather than via computing $\kappa$ directly. We found that two simple heuristics sufficed:
> 1. **Recompute domains of the same type as the update.** When adding Stack-Edu (code) to DCLM, we recompute DCLM:software\_development, and validated this post-hoc by confirming its $\kappa$ was substantially higher than other DCLM domains (Appendix F.3).
> 2. **Recompute domains that are "maxed out."** If a domain's current allocation exhausts all available tokens, freezing its ratio during reuse can be suboptimal — adding a new domain will dampen its ratio, leaving high-utility data unused. We observed this in Section 4.5 on the Stack-Edu data source, and therefore always recomputed source-level ratios after each update.
>
> We emphasize that these were the only two heuristics we tried. We did not sweep over $\mathcal{D}\_{\text{partial}}$ configurations and select the best.
>
> More broadly, we introduced PartialMixtureReuse to show that when full mixture reuse falls short, practitioners need not fall back to full recomputation — there exist choices of $\mathcal{D}\_{\text{partial}}$ that recover near-full-recomputation performance at reduced cost. We view developing more systematic guidelines for setting $\mathcal{D}\_{\text{partial}}$ as an important direction for future work.
>
>
> **Q1 RegMix 1M Clarification** We believe that RegMix “1M” refers to the transformer body parameters only (excluding both the embedding and LM head). However, when counting all non-embedding parameters, their model has ~14M parameters (as a quick check, based on their [code](https://github.com/sail-sg/regmix/blob/main/model_training/lit_gpt/config.py#L104), the LM head already adds 50432 x 256 = 12.9M parameters). We will update the footnote and clarify in the paper that all parameter counts we report are non-embedding parameters.
>
> **Q2 Swarm Sizes** When mixing over DCLM topics, the log-linear regression fit lags behind other regression models (e.g., BiMix) for small $K=m+1$, as in Figure 4 and 15. However, when mixing across sources in Figure 16 (Appendix C), the log-linear model outperforms other models at all $K$. This suggests that in the low sample regime, relative regression model fit may be somewhat dependent on the choice of domains. However, all of these figures—across m and domain sets—show that beyond K=3(m+1), log-linear consistently performs the best. We will update section 3.2.4 with the following takeaway: log-linear should be used when K is at least 3(m+1), and in low sample regimes ($K=m+1$), performance can be domain-dependent.
>
> **Q3, Q4** Thank you for these comments; we will incorporate them into our paper.

---

> > ### Author Rebuttal · Reviewer_yGmN · 2026-04-03
> >
> > I thank the authors for the detailed responses. All my concerns are resolved, and I strongly vouch for the paper!

---

### Official Review · Reviewer_G78a · 2026-03-14

**Soundness:** 2
**Presentation:** 3
**Significance:** 3
**Originality:** 3
**Overall Recommendation:** 4
**Confidence:** 4

**Summary:**

This paper introduces Olmix, a comprehensive framework designed to optimize data mixing throughout the language model (LM) development lifecycle. It addresses two primary challenges: configuring offline mixing methods from scratch, and efficiently adapting mixture proportions when the set of training domains evolves over time (e.g., adding, removing, or modifying datasets).

First, the authors conduct extensive empirical ablations to establish an optimized baseline schema, _OlmixBase_. They identify key best practices, such as utilizing proxy models of at least 15M parameters, scaling the proxy swarm size linearly with the number of domains ($\mathcal{O}(m)$), and employing log-linear regression for performance prediction. Second, to handle evolving domains without prohibitive computational costs, the authors propose a novel "Mixture Reuse" mechanism. This technique groups unchanged domains into a single "Virtual Domain" to freeze their relative sampling ratios, effectively reducing the dimensionality of the search space for new mixtures. Empirical results demonstrate that across a sequence of domain updates simulating real-world LM development, Mixture Reuse achieves over 95% of the performance gains of a full recomputation approach while reducing proxy training costs by 74%.

**Compliance With Llm Reviewing Policy:**

Affirmed.

**Key Questions For Authors:**

1. **Proxy Capacity Bias**: How do you prevent the extremely small proxy swarm (>15M parameters) from down-weighting complex reasoning domains (like deep mathematics or algorithmic code) simply because these domains exceed the learning capacity of a 15M model, thus potentially misguiding the target model?

2. **Static vs. Dynamic Curriculum**: Olmix locks in a static data mixture before the target model begins training. Given that frontier models typically exhibit changing data appetites across their training lifecycle (e.g., requiring more logic-heavy data in later stages), how might the Olmix framework be adapted to support dynamic curriculum learning, or is it fundamentally constrained to static mixing?

3. **Scaling the Rank Correlation**: Your empirical study shows that proxy models >15M achieve strong rank correlation with 1B target models. As target models scale up to frontier sizes (e.g., 7B or 70B) where emergent reasoning capabilities appear, do you expect this absolute proxy size (>15M) to remain sufficient, or must the proxy scale proportionally with the target model?

4. **Estimating Coupling in Practice**: _Partial Mixture Reuse_ relies heavily on identifying domains with high "coupling." In a fully automated data pipeline, how should a practitioner compute or estimate this coupling term efficiently without running expensive pre-evaluations or feature divergence calculations?

**Limitations:**

The authors have not adequately discussed the severe limitations regarding model scale-up capabilities and the static nature of the framework. While they acknowledge some boundaries of their empirical study, they fail to explicitly address the risks of proxy capacity bias when applied to much larger target models (e.g., 70B+), nor do they discuss the framework's inability to accommodate dynamic curriculum learning.

The authors had better to add a dedicated paragraph in the Limitations section explicitly discussing how the 15M proxy-to-1B target correlation might break down at frontier model scales, and acknowledge that the framework produces a static mixture that may not suit advanced multi-stage training regimens.

**Strengths And Weaknesses:**

**Strength**: The methodology is rigorous and well-supported. The authors do not arbitrarily choose regression models or swarm sizes; instead, they use systematic control variable experiments to justify _OlmixBase_, and they provide theoretical error bounds regarding the "coupling effect" for their Mixture Reuse mechanism.

**Strength**: The paper is highly practical and addresses a critical, under-explored pain point in LM development: data domains are rarely static. Furthermore, the empirical finding that proxy swarm size scales linearly ($\mathcal{O}(m)$) rather than quadratically provides immense computational relief and actionable guidance to practitioners.

**Weakness (Proxy Capacity Bias)**: The framework relies on extremely small proxy models (e.g., 15M parameters) to determine the data preferences of much larger target models. However, a 15M model lacks the capacity to effectively learn complex reasoning domains (like advanced mathematics or deep algorithmic code). Consequently, the regression model might penalize these high-quality, complex domains simply because the small proxy cannot learn from them, risking a suboptimal, "dumbed-down" data mixture for the target model.

**Weakness (Scaling Limitations)**: The target models are evaluated exclusively at the 1B parameter scale (trained for 100B tokens). It remains unverified whether the proxy-to-target rank correlations hold as target models scale to frontier sizes (e.g., 7B to 70B+ parameters), where emergent reasoning capabilities drastically alter data appetites.

**Weakness (Ignorance of Dynamic Curriculum)**: The framework strictly outputs a static data mixture for the entire training run. This conflicts with recent findings in frontier LLM pre-training that emphasize dynamic, stage-aware curriculum learning (e.g., shifting from general web text to dense reasoning/code data in later training stages). Olmix locks in the distribution early on, ignoring the model's evolving data needs.

---

> ### Author Rebuttal · Authors · 2026-03-31
>
> Thank you for your detailed and positive review! We are glad you found the paper rigorous and practical. We address your concerns and will incorporate this feedback, including adding a limitations section.
>
> **Q1 Proxy Capacity Bias** We present two new results suggesting that while 15M proxy models may not themselves excel at math/code, they learn enough from these domains to provide reliable signal for data mixing.
>
> 1. Math/code domains influence math/code evals on the proxy models. The coefficient $A_{ij}$ of the learned log-linear regression model (L230) roughly encodes how much domain $i$ influences task $j$. In this [heatmap](https://imgur.com/6UO8QeE) based on the DCLM topic swarm, the science_math_and_technology and software_development domains show strong negative $A_{ij}$ (i.e., large positive impact) — up to -38.86 and -14.6 on math and coding tasks like Minerva MATH and MT-MBPP, compared to the average of -0.92.
> 2. Proxy-scale signals transfer to the target scale for math/code tasks. The average rank correlation is 91.03 for math and 91.45 for code tasks; this is better than the average across all tasks, which is 89 (Figure 2).
>
> **Q2 Static vs Dynamic Curriculum** We agree that data needs can evolve throughout training. Both parts of Olmix are naturally extensible to dynamic settings.
>
> P1: Dynamic mixing methods face design choices similar to those of the offline mixing schema. First, mid-training methods like CLIMB [1] and MergeMix [2] directly use the offline schema to produce updated mixtures. More broadly, [3] shows that many dynamic methods (Skill-It [4], ODM [5], Aioli [3]) implicitly solve an online optimization problem subject to a “dynamic mixing law”—an assumed relationship between the current step's mixture and the next step's performance, analogous to the offline schema's regression model. A full comparison is in the table below:
>
> |Offline mixing schema (static)|Dynamic mixing methods|
> |-|-|
> |Exploration via proxy model swarm|Exploration via adjusting the mix during training|
> |Final performance ≈ f(static mix)| Performance at t+1 ≈ some f(mix at t)|
> |Directly optimizes predicted performance| Updates the mix towards the predicted optimum|
>
> Therefore, a similar analysis of design choices could be conducted for dynamic mixing methods, focusing again on the three steps we have explored in P1 of our paper.
>
> P2: if we update the mixture ratios throughout training and the domain set is updated, we now have two forms of dynamics: over the ratios themselves, and over the *support* of the ratios. However, these are largely decoupled. If at step t the current mixture is $p^t$ and a domain update occurs, we can apply mixture reuse to obtain $q^t$, and then use the dynamic mixing algorithm to update from $q^t$ to $q^{t+1}$.
>
> [1] https://arxiv.org/abs/2504.13161
> [2] https://arxiv.org/abs/2601.17858
> [3] https://arxiv.org/abs/2411.05735
> [4] https://arxiv.org/abs/2307.14430
> [5] https://arxiv.org/abs/2312.02406
>
> **Q3 Scaling the Rank Correlation** We agree that measuring proxy-target rank correlation at larger target sizes is ideal. Unfortunately, this requires training many models (we trained 40 per size). For 7B models with 5x Chinchilla scaling, this requires over 350K H100-hours, which is computationally challenging for us to complete for the rebuttal.
>
> We hypothesize that the required proxy size scales with the target size. [6] suggests that scaling law prediction error is governed by the ratio of proxy to target model size. In Figure 6 of [6], achieving a given error threshold requires approximately the same proxy-to-target ratio across model families differing by over 10x in absolute size (12B Pythia vs 175B OPT). This suggests that transfer for Chinchilla scaling laws requires scaling the proxy proportionally with the target, and therefore the same conclusion may hold for transfer in the data mixing setting.
>
> [6] https://arxiv.org/abs/2410.11840
>
> **Q4 Estimating Coupling in Practice** We thank the reviewer for this question. First, we want to clarify that we introduced partial mixture reuse to show that when full mixture reuse falls short, practitioners need not fall back to full recomputation — there exist choices of $\mathcal{D}\_{\text{partial}}$ that recover near-full-recomputation performance at reduced cost. The coupling term $\kappa$ theoretically explains when and why this works, but we set $\mathcal{D}_{\text{partial}}$ based on practitioner knowledge rather than via computing $\kappa$ directly in an automated pipeline — in our experiments, two simple heuristics sufficed (see our response “W2 Partial Mixture Reuse” to KoZP).
>
> To the specific question of computing $\kappa$: the coupling term can be computed from a fitted regression model on the full domain set $\mathcal{D}'$; however, this defeats the purpose of mixture reuse. We hypothesize that embedding-based similarity scores between domains could serve as a cheaper proxy, though such a measure would be imperfect.

---

### Decision · Program_Chairs · 2026-04-30

**Decision:**

Accept (regular)

**Comment:**

Summary: This paper presents OLMIX, a framework for data mixing throughout LM development that addresses two problems: the mixing configuration problem including proxy model size, swarm size, regression model family, and the evolving domain problem for adding new domains. Through a systematic empirical study resolving six research questions, the authors define proxy model, swarm size, and log-linear regression for estimating mix. These findings inform OlmixBase, a strong baseline configuration. For the second problem, the authors propose Mixture Reuse, which aggregates unchanged domains into a "virtual domain", achieving 95% of full performance while requiring fewer proxy training runs.

Strengths: All four reviewers praised the paper's systematic rigor and practical relevance. The empirical study of design choices is thorough and yields actionable recommendations for practitioners. The evolving domain problem setting is novel and addresses an important problem as data domains are rarely static during LM development, yet prior mixing methods assume they are. The Mixture Reuse mechanism is interesting, shown to work empirically, and theoretically grounded. Accuracy evaluations from the rebuttal also confirm the method provides downstream improvements.

Weaknesses: All four reviewers noted that validation is limited to 1B target models. This is a strong limitation, but consistent with other scaling works. It is likely that the proxy model size will need to increase with the target model size, which imposes some constraint but appears reasonable. This should be discussed in limitations. Two reviewers also commented on confusion on computing the coupling in practice, which authors can clarify in the final version.

**Final Recommendation: Accept**

Justification: The paper makes a clear, and practical contribution to scaling LM pretraining. The systematic configuration study and reuse mechanism are both important, and directly applicable to most pretraining pipelines. Although the reviewers all had outstanding concerns on model sizes, which were not resolved, the study already presents clear benefit, and is results are expected to persist at large scales.